## nature
## ecology & evolution
## OPEN

# Comparative genome anatomy reveals evolutionary insights into a unique amphitriploid fish

Yang Wang [1,6], Xi-Yin Li [1,6], Wen-Jie Xu [2,6], Kun Wang [2,6], Bin Wu [3,6], Meng Xu [3,4,6], Yan Chen [1,6], Li-Jun Miao [1], Zhong-Wei Wang [1], Zhi Li [1], Xiao-Juan Zhang [1], Zhan Yin [1], Bo-Tong Zhou [2], Yu-Lan Yang [3], Cheng-Long Zhu [2], Ming-Liang Hu [2], Jiang-Ming Zheng [2], Chen-Guang Feng [2], Qiang Qiu [2], Le-Tian Tian [1], Meng Lu [1], Fang Peng [1], Wei-Jia Lu [1], Jin-Feng Tong [1], Jin-Gou Tong [1], Bei-De Fu [1], Peng Yu [1], Miao Ding [1], Rui-Hai Gan [1], Qin-Qin Zhang [1], Jian-Bo Jian [3], Chi Zhang [3], Wei-Ming He [3], Wei Yang [3], Zi-Cheng Zhao [3], Qian-Qian Zhang [3], Qiang Gao [3], Jun-Yang Xu [3], Ming-Zhou Bai [3], Ya-Ping Zhang [5], Huan-Ming Yang [3], Xiao-Dong Fang [3 ✉], Wen Wang [2 ✉], Li Zhou [1 ✉] and Jian-Fang Gui [1 ✉]

**Triploids are rare in nature because of difficulties in meiotic and gametogenic processes, especially in vertebrates. The *Carassius* complex of cyprinid teleosts contains sexual tetraploid crucian carp/goldfish (*C. auratus*) and unisexual hexaploid gibel carp/Prussian carp (*C. gibelio*) lineages, providing a valuable model for studying the evolution and maintenance mechanism of unisexual polyploids in vertebrates. Here we sequence the genomes of the two species and assemble their haplotypes, which contain two subgenomes (A and B), to the chromosome level. Sequencing coverage analysis reveals that *C. gibelio* is an amphitriploid (AAABBB) with two triploid sets of chromosomes; each set is derived from a different ancestor. Resequencing data from different strains of *C. gibelio* show that unisexual reproduction has been maintained for over 0.82 million years. Comparative genomics show intensive expansion and alterations of meiotic cell cycle-related genes and an oocyte-specific histone variant. Cytological assays indicate that *C. gibelio* produces unreduced oocytes by an alternative ameiotic pathway; however, sporadic homologous recombination and a high rate of gene conversion also exist in *C. gibelio*. These genomic changes might have facilitated purging deleterious mutations and maintaining genome stability in this unisexual amphitriploid fish. Overall, the current results provide novel insights into the evolutionary mechanisms of the reproductive success in unisexual polyploid vertebrates.**

The genus *Carassius* are very important aquaculture fish and a rare group of vertebrates with different ploidies, including tetraploids and hexaploids[1–3]. Previous studies revealed that the chromosomes of *C. gibelio* have undergone a two-step evolutionary process[4]. Approximately 10 million years ago (Mya), an ancient hybridization of two distant species in the family Cyprinidae led to the origin of the common ancestor of *Carassius*, *Cyprinus* and *Sinocyclocheilus*. Both ancestral parents had 50 chromosomes ($2n = 2× = 50$); thus, the allotetraploidy resulted in a doubling of the chromosome number to 100 ($2n = 4× = 100$) (refs. [3,5,6]). Then, *C. gibelio* experienced subsequent autotriploidy and possessed approximately 150 chromosomes ($3n = 6× ≈ 150$) (refs. [4,7–9]). Therefore, the hexaploid *C. gibelio* could also be considered a triploid.

Triploids are generally considered an evolutionary 'dead end' because of two major challenges to become true 'species'[10]. First, triploid organisms usually cannot produce gametes because pairing and equal segregation of three homologous chromosomes in meiotic and gametogenic processes are insurmountable. Second, the ability of recombination to purge deleterious mutations and generate new traits is reduced without sexual reproduction[11,12]. Unisexual organisms are thought to have high intra-individual genetic diversity (Meselson effect) and accumulation of deleterious mutations (Muller's ratchet) because of the lack of meiotic recombination[11,13–15]. However, triploids are commonly found in some polyploid complex species, including the *Loxopholis* complex[16], *Misgurnus* complex[17], *Poecilia* complex[18] and *Carassius* complex[1,19]. Interestingly, triploid *C. gibelio* overcomes reproductive obstacles via unisexual gynogenesis, where the eggs are activated by the sperm of sympatric sexual species to initiate embryogenesis, such as by kleptospermy in the Amazon molly[20,21], and occupies a wider range of habitats and possesses higher genetic diversity than related sexual species[1,19,22,23]. However, the evolutionary mechanisms underpinning the unisexual reproduction of *C. gibelio* remain unknown.

In this study, we sequenced the genomes of the *Carassius* polyploid complex, including *C. gibelio* and its close relative *C. auratus*, and assembled their two high-quality subgenomes (A and B) that were created during the allotetraploidy event. Combined with resequencing data from different strains, we found that the investigated

[1]State Key Laboratory of Freshwater Ecology and Biotechnology, Hubei Hongshan Laboratory, The Innovation Academy of Seed Design, University of Chinese Academy of Sciences, Institute of Hydrobiology, Chinese Academy of Sciences, Wuhan, China. [2]School of Ecology and Environment, Northwestern Polytechnical University, Xi'an, China. [3]BGI Genomics, BGI-Shenzhen, Shenzhen, China. [4]Department of Ecology, Jinan University, Guangzhou, China. [5]State Key Laboratory of Genetic Resources and Evolution, Kunming Institute of Zoology, Chinese Academy of Sciences, Kunming, China. [6]These authors contributed equally: Yang Wang, Xi-Yin Li, Wen-Jie Xu, Kun Wang, Bin Wu, Meng Xu, Yan Chen. ✉e-mail: fangxd@bgi.com; wenwang@nwpu.edu.cn; zhouli@ihb.ac.cn; jfgui@ihb.ac.cn

*C. gibelio* descended from an autotriploidy event hundreds of thousands of years ago. Comparative genome analysis and cytological observations revealed that some meiotic cell cycle-related genes and an oocyte-specific histone variant have intensively expanded and changed, which provided the genomic variation evidence that facilitates gynogenetic oogenesis in *C. gibelio*. Moreover, unexpected sporadic homologous recombination and a high level of gene conversion among homologues may be the main driver to purge deleterious mutations in *C. gibelio*. Overall, these novel discoveries provide unprecedented insights into a rare reproductive mode in nature and the underlying genomic evolution mechanism. Additionally, the newly sequenced genomes are valuable resources for precise genetic breeding of *Carassius* species in aquaculture.

## Results

**C. gibelio and C. auratus genome sequencing and assembly.** PacBio, Illumina and Hi-C sequencing technologies were applied to generate a high-quality genome assembly for *C. gibelio* and *C. auratus* (Supplementary Tables 1–4 and Supplementary Fig. 1). The Illumina short reads were first used to investigate the polyploidy through Smudgeplot analysis (Supplementary Note 1)[24]. In *C. auratus*, 58% of heterozygous *k*-mer pairs (with only one nucleotide difference and presented as *x* and *x′*) are bivalent (*xx′*) and 33% of heterozygous *k*-mer pairs are tetravalent (*xxx′x′* and *xxxx′*) (Extended Data Fig. 1a). This pattern is consistent with amphidiploid (a synonym of allotetraploid[25], AABB) characteristics, where two subgenomes are quite divergent but still homologous. In contrast, *C. gibelio* had mostly heterozygous *k*-mer pairs with the structure *xxx′* (72%), followed by heterozygous *k*-mer pairs with the structure *xxxx′x′* (23%) (Extended Data Fig. 1b), which fits the AAABBB genotype. The estimated haplotype genome size of *C. gibelio* ranged from 1.49 to 1.56 Gb in *k*-mer analysis, which is approximately one-third of the genome content (4.70–5.38 pg) estimated by flow cytometric analysis[26,27] and similar to the estimated haploid genome size of *C. auratus* (Supplementary Table 5 and Supplementary Note 1). These results indicate that both of the species have the same amphihaploid content (AB).

The haplotype genome of *C. gibelio* comprised 2,804 contigs, with a length of 1.59 GB and contig N50 of 1.71 Mb (Supplementary Table 6). In total, 2,063 contigs were anchored into 50 chromosomes with a total length of 1,502.18 Mb using the Hi-C data (Fig. 1a, Supplementary Table 7 and Supplementary Fig. 2). The assembly contained 98.16% of complete benchmarking universal single-copy orthologs (BUSCO) genes, 45,249 protein-coding genes and 728.98 Mb (45.85%) of repeat contents (Supplementary Tables 8–13 and Supplementary Note 2). The *C. auratus* genome was also assembled with a size of 1.52 Gb and contig N50 of 3.89 Mb, and anchored to 50 chromosomes (Fig. 1a, Supplementary Fig. 3 and Supplementary Tables 6 and 7). The 50 chromosomes of the both fish were divided into two subgenomes, each of which included 25 chromosomes (Fig. 1b), based on the annotation of gene and repeat content. The partition of subgenomes was observed to be consistent with previously published domestic goldfish and common carp genomes through synteny analysis (Supplementary Figs. 4 and 5).

Because both the *k*-mer estimated and assembled genome sizes of *C. gibelio* were approximately one-third of the genome content, it was evident that the genome assembly included only AB subgenomes; this was the same as the genome assembly of *C. auratus*. To validate this inference, we made the following two comparisons. First, we performed synteny analysis between *C. gibelio* and *C. auratus*, and found that each of their chromosomes aligned well without obvious chromosomal fission or fusion events (Fig. 1a). Second, the reads of each species were mapped back to corresponding genome assemblies to evaluate the allele frequencies and read depths. The minor allele frequencies of most chromosomes were found to be ~0.33 in *C. gibelio* and ~0.50 in *C. auratus* (Fig. 1c). The read depths

across the genome were also approximately three times that of the single haplotype in *C. gibelio* and two times that of the single haplotype in *C. auratus* (Fig. 1d).

Moreover, to provide more evidence at the genomic block and gene levels, we performed an allelic analysis by BAC phasing and polymerase chain reaction (PCR) verification. We found that most of the phased blocks indeed had three homologous alleles for both A and B subgenomes in *C. gibelio* (Supplementary Fig. 6), and the functionally investigated *foxl2* and *viperin* were also demonstrated to contain three highly identical alleles[28,29]. These results clearly show that both the genome assemblies of *C. gibelio* and *C. auratus* comprise one haplotype of the AB subgenomes, but *C. gibelio* has three haplotypes for most chromosomes (this will be discussed in a later section) and *C. auratus* has two haplotypes for all chromosomes (Fig. 1e). Following the nomenclature of amphidiploid, we called *C. gibelio* an amphitriploid (AAABBB) with two triploid sets of chromosomes, each of which was derived from a different ancestor.

**Allotetraploidy and genomic variations of Carassius.** The phylogenetic relationship was reconstructed using both concatenated and coalescent methods (Fig. 2 and Supplementary Fig. 7). Consistent with previous studies[5,30], subgenome B had a closer relationship to the diploid mud carp (*Cirrhinus molitorella*) and Yunnan Wenkong Barbinae fish (*Poropuntius huangchuchieni*) than subgenome A. It could be inferred that: (1) the progenitor-like genomes (ancestors of subgenomes A and B) diverged around 19.50 Mya (T1) (Fig. 2); (2) the allotetraploidy event (the hybridization of subgenomes A and B) occurred between 10.17 and 12.87 Mya (T2), based on the divergence times of common carp (*Cyprinus carpio*) versus *Carassius*, and versus *P. huangchuchieni*; and (3) the divergence time of *C. gibelio* and *C. auratus* occurred around 0.96 Mya (T3) (Fig. 2). The new estimates of timing were more ancient than previously thought (T1: 13.75 to 15.09 Mya) (ref. [30]) partially because we discarded a suspicious time calibration: the divergence time between Cyprininae and Leuciscinae (~20.5 Mya) (refs. [30,31]). This widely used time calibration was not from fossil records but from estimation based on several nuclear and mitochondrial genes along with the mutation rate of mammals[32]. Compared with previous dating, newly estimated divergence times without this calibration have a better fit to the distribution of synonymous mutations ($K_s$) between species (Supplementary Fig. 8). In addition, we noticed that the phylogenetic position of *Cirrhinus molitorella* and a previous study[30] conflicted with another previous study[33], in which a single gene (*rag2*) tree was constructed and the results showed that *C. molitorella* was an outgroup of both subgenomes A and B. To determine why this inconsistency occurred, we further examined the proportion of topology for each orthologous gene. The results highlighted a high level of phylogeny heterogeneity (Supplementary Table 14), and the topology with the highest proportion was consistent with the current phylogenetic tree.

The evolution of subgenomes of these carps has been widely studied[5,30,31,33–35], and here, the more dominant subgenome B was confirmed (Supplementary Fig. 9 and Supplementary Note 3). Also, we have identified genes that are specifically lost in *Carassius* species (Supplementary Table 15, Supplementary Fig. 10 and Supplementary Note 4).

**Autotriploidy origin and genomic changes of C. gibelio.** Overall, six *C. gibelio* individuals from three strains were used to investigate the origin of this unisexual species, including three individuals for strain A+, two for strain H and one for strain F (Supplementary Table 16). Combined with ten *C. auratus* individuals and one *Cyprinus carpio* individual downloaded from public databases (Supplementary Table 17), 48,843,026 single-nucleotide polymorphisms (SNPs) and 8,431,930 insertions and deletions were called

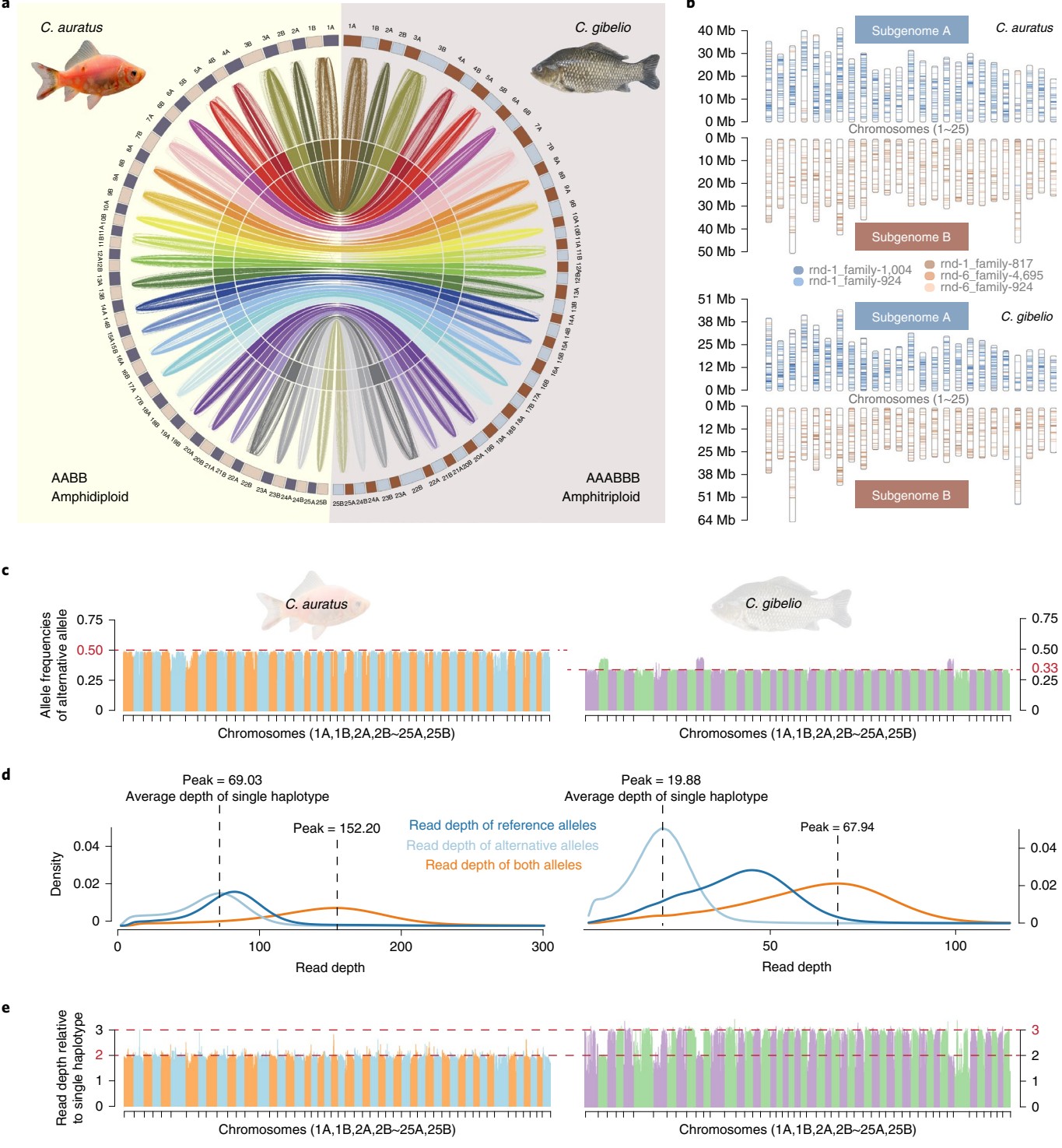

**Fig. 1 | Genome assemblies of *C. gibelio* and *C. auratus*. a**, Syntenic relationship between and within *C. auratus* and *C. gibelio* genomes. The outermost circle shows the 50 chromosomes of *C. auratus* (left) and *C. gibelio* (right); the subsequent coloured lines represent the syntenic relationship between subgenomes A and B in each species; the innermost coloured lines represent the syntenic relationship between the homologous chromosomes of *C. auratus* and *C. gibelio*. **b**, Localization of subgenome-specific repeats in *C. auratus* and *C. gibelio* chromosomes. Two TEs (blue) are mainly distributed in subgenome A, whereas three TEs (tan) are mainly distributed in subgenome B. **c**, Allele frequencies of alternative alleles in each chromosome of *C. auratus* and *C. gibelio*. The average frequencies were close to 0.50 and 0.33 in *C. auratus* and *C. gibelio*, respectively (red dashed line); this indicated two and three haplotypes for their genomes, respectively. The order of chromosomes is consistent with that in **a**. **d**, Density of read depth for reference or alternative alleles in *C. auratus* and *C. gibelio*. The left panel shows a similar read depth of reference and alternative alleles, whereas the right panel shows twice as much coverage of the reference alleles than the alternative alleles. **e**, The read depth relative to the single haplotype in each chromosome of *C. auratus* and *C. gibelio*. The order of chromosomes is consistent with that in **a**.

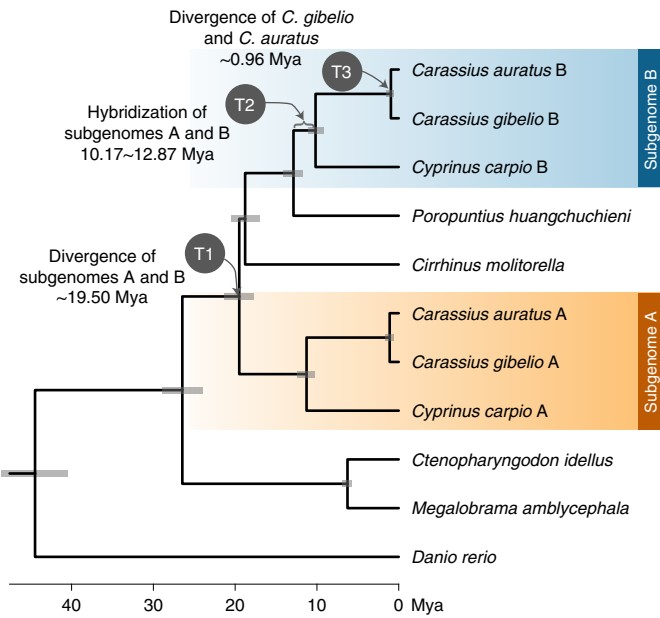

**Fig. 2 | Phylogenetic relationship and divergence time of the two subgenomes of *Carassius*.** The divergence time (T1) and hybridization time (T2) of the two subgenomes are shown at corresponding nodes.

- Divergence of *C. gibelio* and *C. auratus* ~0.96 Mya
- Hybridization of subgenomes A and B 10.17~12.87 Mya
- Divergence of subgenomes A and B ~19.50 Mya
- *Carassius auratus* B
- *Carassius gibelio* B
- *Cyprinus carpio* B
- *Poropuntius huangchuchieni*
- *Cirrhinus molitorella*
- *Carassius auratus* A
- *Carassius gibelio* A
- *Cyprinus carpio* A
- *Ctenopharyngodon idellus*
- *Megalobrama amblycephala*
- *Danio rerio*
- Subgenome B
- Subgenome A
- 40  30  20  10  0 Mya

within *C. gibelio* using the *C. auratus* genome assembly as a reference (Supplementary Table 18). The depth distributions of minor alleles revealed that almost all *C. gibelio* individuals had three alleles for each chromosome, whereas all *C. auratus* individuals had two alleles for each chromosome (Extended Data Fig. 2); this further confirmed that *C. auratus* and *C. gibelio* are amphidiploid and amphitriploid, respectively.

Principal component (PC) analysis was used to examine the phylogenetic relationships among different strains of *C. gibelio* and *C. auratus*. The first component explained 18.62% of the genetic variance and showed a clear split between *C. gibelio* and *C. auratus*, whereas the second component explained 13.28% of the genetic variance and showed clear distance among the three strains of *C. gibelio* that could be associated with the lack of gene flow due to unisexual reproduction (Fig. 3a). The maximum likelihood tree yielded similar results (Fig. 3b). Moreover, 4,400 non-coding elements were found to be shared by all *C. gibelio* individuals (Supplementary Fig. 11 and Supplementary Note 5) but were absent in *C. auratus*, *Cyprinus carpio* and *S. graham*, indicating that they are newly evolved elements in *C. gibelio*. Taken together, these results suggest that the investigated *C. gibelio* might have a common origin.

The divergence time of the three *C. gibelio* strains was estimated to be approximately 0.82 Mya (T4) using four degenerated sites (Supplementary Fig. 12). Therefore, all *C. gibelio* lines probably originated from an amphidiploid ancestor that experienced an autotriploidy event at approximately 0.82–0.96 Mya (Fig. 3c). This also means that the unisexual reproduction of *C. gibelio* has been maintained for a long time.

We also noticed that some chromosomes in the individuals, including *C. gibelio* (*Cg*)-F1, *Cg*-A1, *Cg*-A2 and *Cg*-A3, exhibited unusual alterations of allele frequencies and read depths (Supplementary Fig. 13). Compared with other chromosomes, these unusual chromosomes from different individuals had allele frequencies of approximately 0.50, which is very close to that of *C. auratus* chromosomes, and had approximately 2/3 or 4/3 the read depths of other *C. gibelio* chromosomes (Supplementary Fig. 13). These data indicate that these chromosomes have lost or obtained one haplotype. In addition, we estimated the expression ratios of the individual

*Cg*-F for each chromosome compared with the corresponding *C. auratus* genes. In a global analysis that combined seven tissues to determine the average expression levels of orthologous genes between *C. auratus* and *C. gibelio*, the three unusual chromosomes displayed clear decreases in average gene expression ratio ($P = 6.86 \times 10^{-7}$, $6.24 \times 10^{-8}$ and $2.21 \times 10^{-9}$, *t*-test), and were only approximately 2/3 that of other chromosomes (Supplementary Fig. 14).

**Expansion of meiosis-related genes in the *C. gibelio* genome.** In triploids, the three homologous chromosomes cannot pair correctly or segregate equally during meiosis I, which causes failure of gametogenesis[36]. To understand what happens in *C. gibelio* oogenesis, we first measured the DNA content during oocyte development. The DNA content of *C. gibelio* oocytes at early prophase was approximately 1.67 times that of corresponding *C. auratus* oocytes (Fig. 4a), whereas the DNA content of *C. gibelio* mature oocytes was approximately 3 times that of *C. auratus* mature oocytes (Fig. 4a); this indicates formation of unreduced eggs in *C. gibelio* compared with formation of reduced eggs in *C. auratus*. Additionally, compared with 50 bivalents in *C. auratus*, an average of more than 130 univalents was counted in germinal vesicle breakdown oocytes of *C. gibelio* (Fig. 4b); these findings suggest that chiasmata, which physically connect homologous chromosomes, were largely missing. Therefore, meiosis I was suppressed during oogenesis in *C. gibelio* (Fig. 4c).

To explore the genomic clues concerning the unreduced eggs in *C. gibelio*, we performed an in-depth comparative genomic analysis and found a total of 13 gene families that have more copies in all *C. gibelio* individuals compared with *C. auratus* and *Cyprinus carpio* (Fig. 4d and Supplementary Table 19). Interestingly, nine of the expanded gene families have important roles in oocyte development, especially in meiosis and spindle organization. The most expanded gene is a histone variant, *h2af1al*, of which the B homeologue has expanded to 11 copies in the *C. gibelio* assembly (Fig. 4e). Five of the expanded copies (B1–B5) were found to be specifically expressed in the ovary (Fig. 4e). Further, transcriptomic analyses of the isolated oocytes and embryos indicated that these histone variants are maternal factors with high expression in pre-vitellogenic oocytes (POs) and vitellogenic oocytes (VOs), which correspond to pre- and post-diplotene stages of meiosis prophase I, respectively. Histone variants can replace canonical histones to remodel chromatin and affect histone post-translational modifications[37], and H2af1al has the ability to modify nucleosome properties during oogenesis in *C. gibelio*[38].

Importantly, all of the expanded meiosis-related genes, including two cell cycle-related genes (*fbxo5* and *ccna2*), three spindle organization genes (*rhoA*, *incenp* and *nusap1*) and three nuclear envelope-related genes (*lem4*, *lap2* and *bmb*), were assigned to the common meiosis pathway of oocyte development (Fig. 4f). Most of them (22 of the 26 extra copies of the eight expanded genes) were expressed in the ovary, POs or VOs (RPKM >1) (Supplementary Fig. 15), indicating that they have roles in oocyte development of *C. gibelio*. We also noticed that most of the new expanded copies were distributed far from the parental copies in genome, with only three exceptions (Extended Data Fig. 3a,b and Supplementary Table 19). In particular, all of the extra copies of *h2af1al* (11 extra copies) and *faap24* (two extra copies) were adjacent to a *C. gibelio*-specific repeat unit (Extended Data Fig. 3c), indicating that the expansions of these genes might have been mediated by repetitive sequences. The above data suggest that an alternative oogenic pathway to produce chromosome number-unreduced eggs is probably related to intensive expansion of meiosis-related genes in *C. gibelio*.

**Gene conversion and sporadic homologous recombination.** It is usually believed that unisexual organisms cannot purge deleterious mutations because no homologous recombination exists during

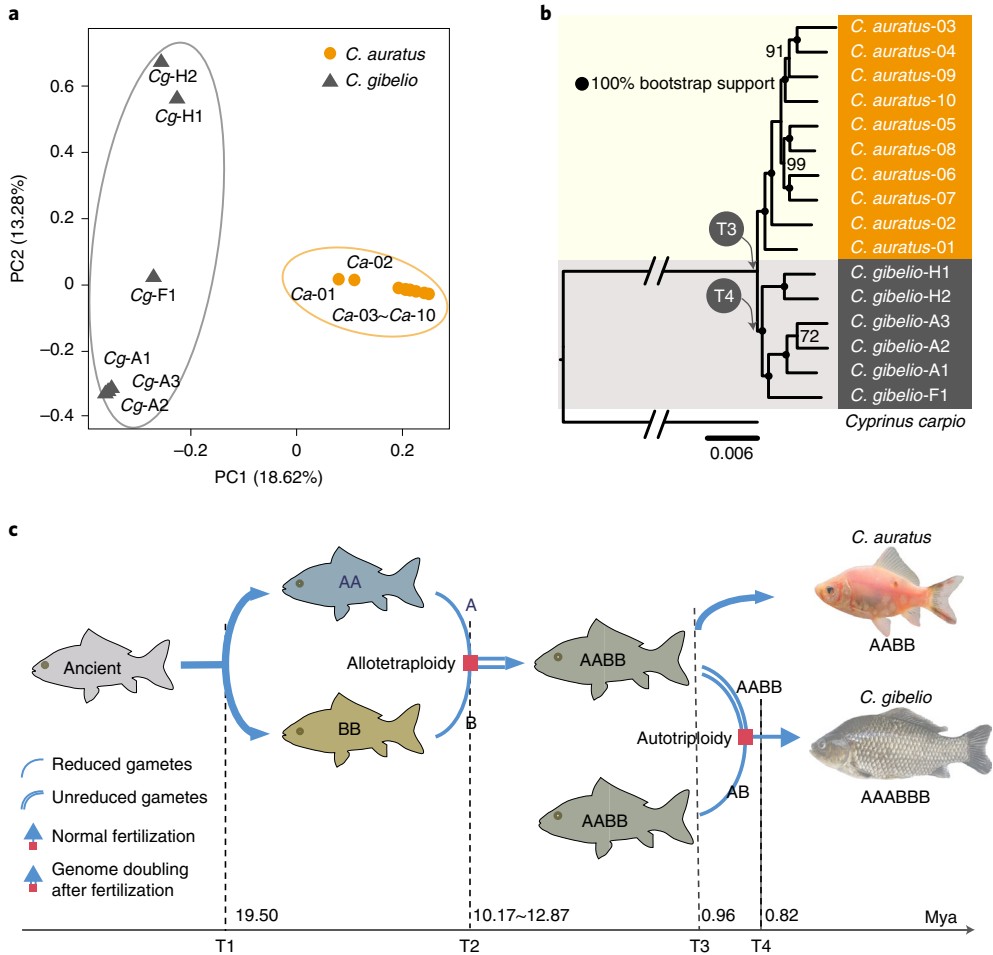

**Fig. 3 | Ancient autotriploidy origin of *C. gibelio*. a**, PC analysis plots of the first two components. The fractions of the explained variance are 18.62% for PC1 and 13.28% for PC2. **b**, Phylogenetic tree constructed using the nuclear genome. The nodes with solid black circles are 100% bootstrap supported. The divergence times of *C. gibelio* and *C. auratus* (T3) and of three *C. gibelio* strains (T4) are shown at corresponding nodes. **c**, Different epochs of *C. gibelio* and *C. auratus* evolution with times. Single lines indicate haploid gametes, and double lines indicate diploid gametes. Red squares with a single-line arrow and double-line arrow represent normal fertilization and genome doubling after fertilization, respectively.

gametogenesis. To study whether deleterious mutations accumulate in *C. gibelio*, we first compared the genomic heterozygosity between the two *Carassius* species. The percentage of heterozygous sites is approximately two times higher in *C. gibelio* than in *C. auratus* (Fig. 5a). As *C. gibelio* has three haplotypes per chromosome, this difference is not surprising. We then investigated the number of loss-of-function mutations, non-synonymous substitutions and synonymous substitutions in the two *Carassius* species using *Cyprinus carpio* as a reference. Interestingly, there was no notable difference between the two species and all three types of mutations exhibited similar distribution patterns (Fig. 5b and Supplementary Fig. 16). These results indicate that *C. gibelio* is likely to have the ability to purge mutations, including deleterious mutations, even though it reproduces unisexually.

To evaluate the ability of *C. gibelio* to purge mutations, we conducted a four-generation breeding experiment for 5 years and tested whether loss of heterozygosity (LOH) occurred in the laboratory environment. LOH is a common form of allelic imbalance by which a heterozygous allele becomes homozygous by deleting one homologue or gene conversion, a unidirectional modification of the DNA sequence between similar sequences (Extended Data Fig. 4). Using 11 individuals from the offspring of the gynogenetic line (Supplementary Table 20), we identified 805 LOH regions across 46 chromosomes (Fig. 5c). Most LOH regions were shared by

many individuals and thus were probably inherited from ancestors; however, a few were unique, which means they should be newly occurring in individuals (Fig. 5c). PCR and Sanger sequencing validated 97 out of 101 arbitrarily selected LOH loci (Supplementary Fig. 17). The rate of LOH was estimated to be $1.49 \times 10^{-4}$ per heterozygous site per generation (Supplementary Table 21), which was much higher than the base-substitution mutation rate of $8.88 \times 10^{-9}$ (Methods). The rate of homologous gene conversion was $1.42 \times 10^{-4}$ per heterozygous site per generation (Supplementary Table 22), which indicated that gene conversion is responsible for the vast majority of LOH. The gene conversion rate of *C. gibelio* is two orders of magnitude higher than that of the reported unisexual species[39,40] and nearly reaches the reported range of some sexual species[41,42], which have an efficient deleterious mutation purging mechanism through recombination in normal meiosis.

Gene conversion has been revealed to be able to compensate for the lack of meiotic recombination in diploid asexual/unisexual organisms[43]. When an LOH event occurs in a genomic region of diploid species, a variant may be cleared or spread, both at a ratio of 50% (Fig. 5d, top). However, there are six possible scenarios of gene conversion in triploid species (Fig. 5d, bottom). In two of the scenarios, the newly occurring mutation was eliminated; in two other scenarios, the proportion of this mutation did not change; and in the last two scenarios, this mutation expanded to more alleles.

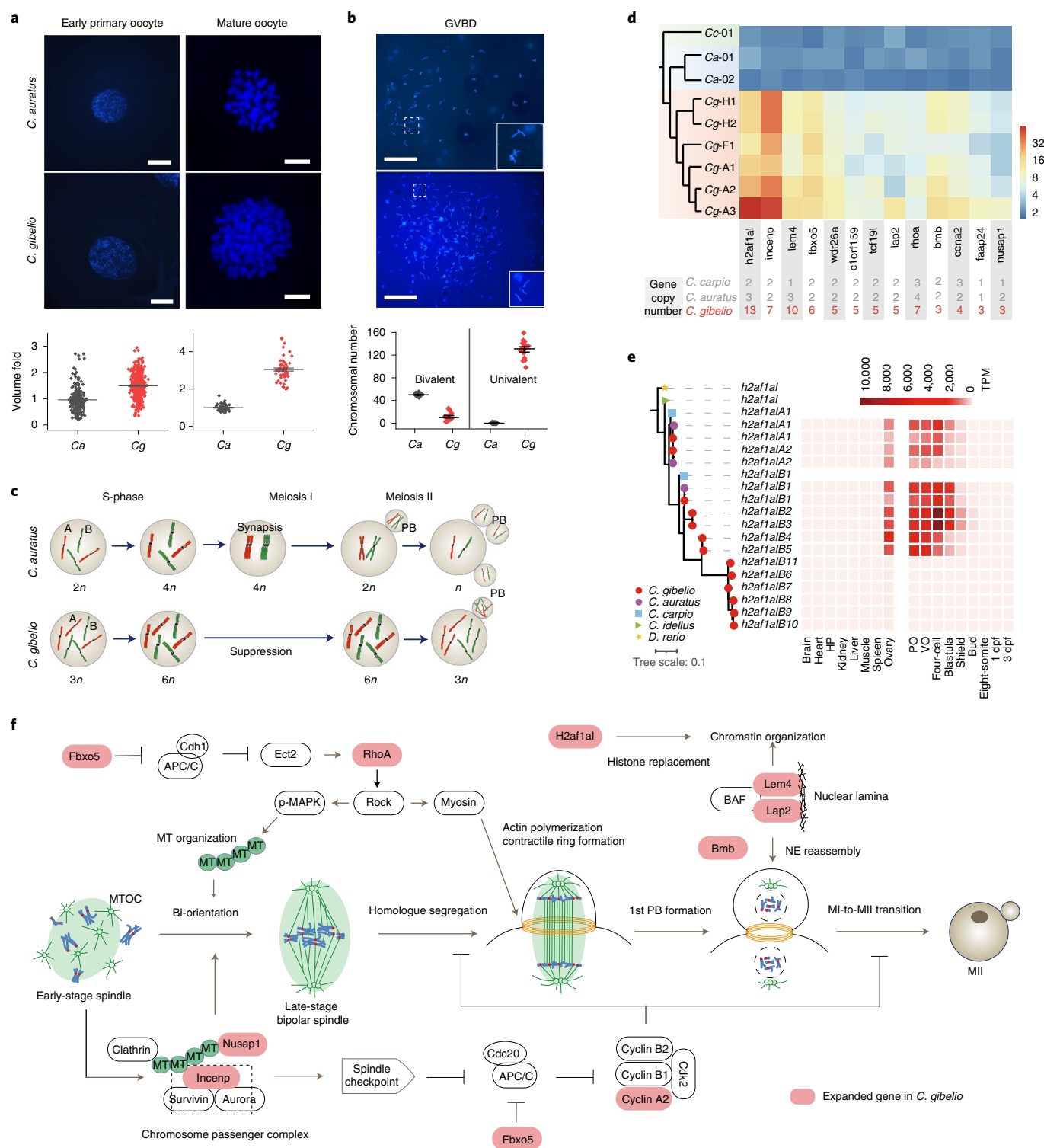

**Fig. 4 | Alteration of oogenesis and expansion of meiosis-related genes in *C. gibelio*. a**, DNA content of early primary oocytes ($n = 318$ for *C. auratus* and $n = 434$ for *C. gibelio*; scale bar, 50 μm) and mature oocytes ($n = 56$ for *C. auratus* and $n = 54$ for *C. gibelio*; scale bar, 5 μm). Data are presented as mean ± standard error of the mean (s.e.m.). **b**, Bivalent and univalent numbers of oocytes at germinal vesicle breakdown (GVBD) ($n = 19$ for *C. auratus* and $n = 20$ for *C. gibelio*; scale bar, 50 μm). Data are presented as mean ± s.e.m.. **c**, Schematic of *C. auratus* and *C. gibelio* oocyte maturation division. Normal meiosis produces oocytes with a halved genome in *C. auratus*, whereas the first meiotic division is suppressed in *C. gibelio*. PB, polar body. **d**, Linkage-specific gene expansion shared by the three investigated *C. gibelio* strains. The numbers below are the gene copy numbers in the assemblies of *Cyprinus carpio* (GCA_000951615.2), *C. auratus* and *C. gibelio* (this study). The heat map shows the relative depth of each gene in the resequencing samples. **e**, Expanded histone variant *h2af1al* and their expression in *C. gibelio*. Left: maximum likelihood gene tree of *h2af1al*. Right: heat map of *h2af1al* expression levels in different tissues, oocytes and embryos. HP, hypothalamus and pituitary. **f**, Nine lineage-specific expanded genes of *C. gibelio* in the regular oocyte meiosis pathway. MT, microtubule; MTOC, microtubule-organizing centre; 1st PB, the first polar body; NE, nuclear envelope; MI, meiosis I; MII, meiosis II.

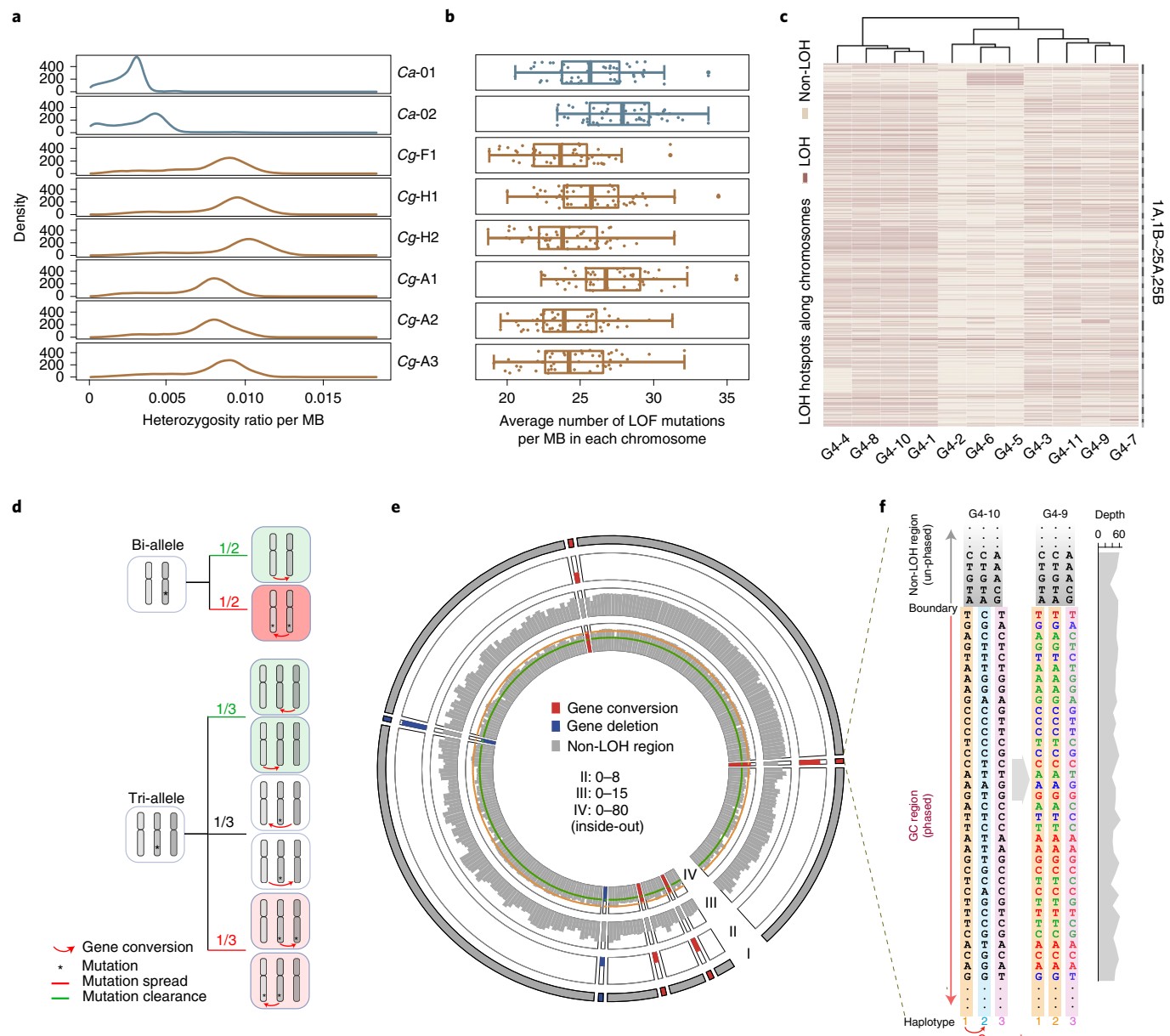

**Fig. 5 | Gene conversion evidence in *C. gibelio*. a**, Heterozygosity ratio per megabase (MB) in different individuals. **b**, Distribution of loss-of-function (LOF) mutations per MB. For each individual there are 50 points, where each point represents the average number of mutations per 1 MB region on each chromosome. The line in the middle of each box plot represents the median of the dataset; the upper and lower edges of box plot indicate the third quartile and first quartile, respectively; and the line extending from the edge is 1.5 times the interquartile range. **c**, Heat map showing LOH regions in 11 individuals of the gynogenetic line. Individuals are clustered on the basis of the LOH regions along chromosomes. **d**, Mutation fates after gene conversion in unisexual organisms. **e**, LOH regions in chromosome 22B of individual G4-9. I, LOH region. II, LOH site number (log$_2$). III, SNP number (log$_2$). IV, read depth of SNP sites. The window size in II–IV is 200 kb. Red and blue bricks indicate gene conversion and gene deletion, respectively. The orange and green lines in IV indicate sequencing depths of 50× and 33×, respectively. **f**, The phased haplotype blocks at the boundary of the gene conversion from haplotype 1 (donor) to haplotype 2 (recipient) in **e**. The same region of individual G4-10 is shown as a control without gene conversion. Red bases, homozygous converted sites result in LOH (LOH sites). Green bases, heterozygous converted sites where the donor allele is minor allele (non-LOH site with changed genotype). Blue bases, heterozygous sites where the donor allele had the same base as the recipient (non-LOH site with unchanged genotype). Black bases with grey shadowed, heterozygous sites outside the gene conversion region. Read depths of SNP sites in individual G4-9 are shown on the right panel.

Therefore, gene conversion can purge mutations and increase diversity among offspring in a more complex manner for triploids.

To understand this from a detailed perspective, we presented two candidate gene conversion regions (Fig. 5e and Extended Data Fig. 5). According to the read coverage of SNP sites between the individuals from the gynogenetic *C. gibelio* pedigree that did or did not experience gene conversion (Supplementary Fig. 18), the

haplotype blocks of gene conversion could be inferred (see the detailed description in Supplementary Note 6). As shown in Fig. 5f, after gene conversion from haplotype 1 to haplotype 2, 12 out of 35 SNP sites (~1/3) became homozygous, which resulted in LOH; the other sites were still heterozygous, among which 14 SNP sites were clearly converted, and nine SNP sites looked unchanged because their haplotypes 1 and 2 had the same bases before conversion.

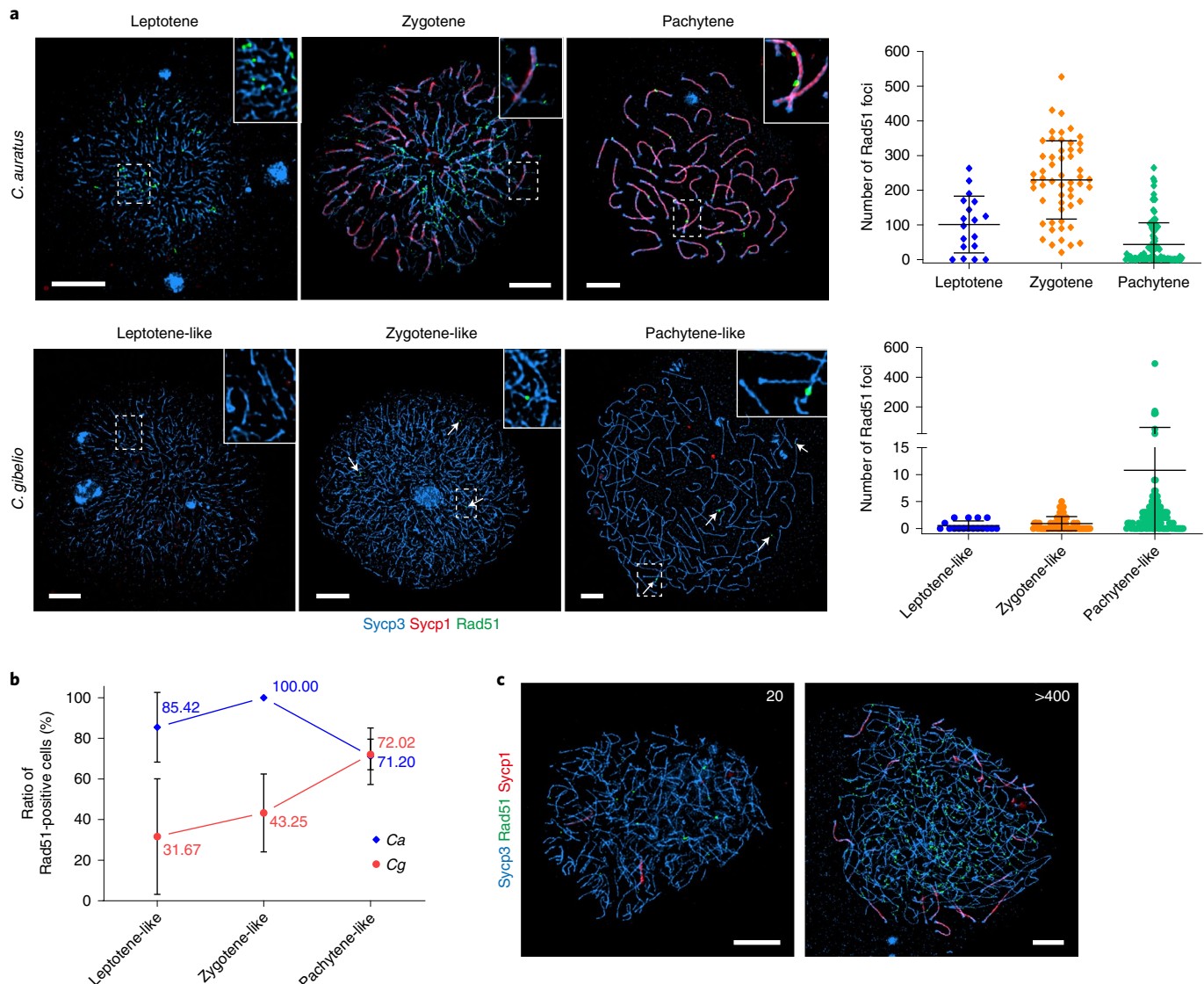

**Fig. 6 | Comparative cytological observation and sporadic homologous recombination during oogenic division between *C. gibelio* and *C. auratus*.** **a**, Chromosomal spreads of different stage primary oocytes of *C. auratus* and *C. gibelio* co-immunostained by anti-Sycp1 (red), anti-Sycp3 (blue) and anti-Rad51 (green) antibodies. Insets show an enlarged view. Arrows mark Rad51 foci. Scale bar, 5 μm. Statistical results are presented as mean ± standard deviation (s.d.) in the right panel. *n* = 174 for *C. auratus* and *n* = 203 for *C. gibelio*. Scale bar, 5 μm. **b**, Percentage of Rad51-positive oocytes. Data are presented as mean ± s.d. *n* = 174 for *C. auratus* and *n* = 203 for *C. gibelio*. **c**, Varying levels of Rad51-stained foci and synaptonemal bivalents in different pachytene-like oocytes of *C. gibelio*. Number of Rad51-stained foci is indicated at the top right. Scale bar, 5 μm.

Therefore, high gene conversion might render *C. gibelio* capable of purging deleterious mutations and may be associated with the alternative ameiotic oogenic mechanism.

Consequently, we comparatively explored chromatin behaviour and recombination occurrence during oogenesis of sexual *C. auratus* and unisexual *C. gibelio* through co-immunostaining with anti-antibodies for synaptonemal complex (SC) transverse element (Sycp1), lateral element (Sycp3) and recombinase Rad51 (refs. [44,45]). Typical SC formation and homologous recombination were observed in *C. auratus*, in which 50 synaptonemal bivalents and numerous recombinase Rad51-stained foci were visible, and the highest number of foci was reached (over ~200 per cell on average) at zygotene (Fig. 6a). In contrast, only Sycp3-stained univalents appeared in most oocytes of *C. gibelio* (Fig. 6a), which indicated that SC did not assemble within these oocytes. Homologous recombination indicated by Rad51 signals was also largely suppressed, but

sporadic Rad51-stained foci were observed in oocytes of *C. gibelio* (Fig. 6a). Importantly, the ratio of the Rad51-positive oocytes was found to have an increasing trend along with the progress of oocyte development (Fig. 6b), in which some oocytes (~2.5%) even showed high levels of Rad51-stained foci (over 400) and synaptonemal bivalents (over 20) (Fig. 6c). The different levels of homologous recombination revealed in different oocytes of *C. gibelio* are consistent with the large variations of gene conversion rates observed among different gynogenetic individuals (Supplementary Table 22), indicating an association between them because non-crossover homologous recombination usually results in gene conversion[46].

## Discussion

The genomic anatomy of polyploids has been broadly determined in plants and animals, such as in a tetraploid frog (LLSS)[47], hexaploid wheat (AABBDD)[48] and octoploid strawberry (AABBCCDD)[49].

However, these dissected polyploid genomes actually represent diploid genomes that contain two or multiple subgenomes. Here, we provide an assembly of an amphitriploid genome (AAABBB), where most genes commonly have two divergent homeologues and each homeologue possesses three highly similar alleles. Although phasing is not complete because of the recent autotriploidy event and the limitation of error-prone long reads, we revealed important genomic changes based on this assembly, including intensive expansion of many meiosis-related genes and a high rate of gene conversion.

Recently, Hojsgaard and Schartl proposed that a genomic assemblage and an alternative reproductive module might be required for the formation of a functioning asexual/unisexual genome[50]. Intriguingly, the unique amphitriploid genome just represents a non-recombinant genomic assemblage, with intensive expansion and alterations of meiotic cell cycle-related genes and an oocyte-specific histone variant (Fig. 4d,e and Supplementary Fig. 15). These genomic alterations might act as a complementary reproductive module to skip meiosis using an alternative ameiotic pathway to develop into unreduced eggs, and may be essential for the success of unisexual gynogenesis in *C. gibelio*.

It has been argued that asexual/unisexual lineages should go extinct quickly because they have a reduced ability to purge deleterious mutations and generate high levels of heterozygosity[51,52]. Similar to *C. gibelio*, some extant asexual lineages do not exhibit such genomic decays[40,53,54]. Ameiotic homologous recombination that results in gene conversion has been proposed to be the mechanism to conquer these hindrances for the evolutionary longevity of asexual/unisexual lineages[14,40,43]. Interestingly, we observed sporadic homologous recombination during oocyte development, and the high rate of gene conversion in *C. gibelio* is even two orders of magnitude higher than the famous unisexual Amazon molly[40], indicating that *C. gibelio* might have an efficient way to increase genetic diversity and purge deleterious mutations. Besides high gene conversion rate, in sharp contrast to other unisexual vertebrates, rare and variable proportions of males (1.2–26.5%) have been found in wild populations of *C. gibelio*[55]. Previous studies revealed that the male-specific supernumerary microchromosomes may be the main driving forces for the occurrence of genotypic males[56,57] and could result in the creation of beneficial genetic diversity[58,59]. Therefore, gene conversion and sex might play a key role in fine-tuning the efficiency of gynogenesis[60] and contribute to the long evolutionary existence of *C. gibelio*. However, after initial attempts, we were unfortunately not able to detect substantial mutations around the potential master sex gene *amh*[61] between *C. auratus* and *C. gibelio* (Supplementary Fig. 19). Additionally, we failed to obtain any informative male-specific supernumerary sequences from one male individual of *C. gibelio* (Supplementary Note 7). A high-quality male genome assembly for *C. gibelio* will be required to uncover the mechanisms underlying male determination[62] and gene conversion in the future.

In addition to the genetic importance of our results, the current genomic anatomy in the *Carassius* complex is also of biological value for genetic breeding to improve aquaculture strains because *C. gibelio* is one of the most important aquaculture species in China, with approximately 3 million tons of annual production capacity. In the past decades, several new varieties, including allogynogenetic gibel carp[63], high dorsal gibel carp[64], gibel carp 'CAS III' (ref. [65]), gibel carp 'CAS V' (refs. [66,67]) and 'Changfeng' gibel carp[68,69], have been successfully bred and have made important contributions to Chinese aquaculture[70,71]. Thus, the genomic data of amphitriploid *C. gibelio* will provide a valuable resource for accelerating the genetic analysis of economic traits and the precise breeding of new varieties.

Overall, our data and analyses have provided important insight into the genome structure, evolutionary history and genetic maintenance mechanism of the unique amphitriploid *C. gibelio*.

Nevertheless, it is noteworthy that better genome assemblies with all chromosomes phased, which requires very advanced sequencing technology, may be able to provide more comprehensive genetic data to infer the complete picture of the evolution and maintenance of the rare amphitriploid genome of *C. gibelio*.

## Methods

**Experimental fish.** All individuals were maintained and sampled from the National Aquatic Biological Resource Center. Animal experiment was approved by the Animal Care and Use Committee of the Institute of Hydrobiology (IHB), Chinese Academy of Sciences (CAS) (approval ID keshuizhuan 0829).

**Genome and transcriptome sequencing.** Genomic DNA was extracted from the blood cells of a female adult individual from strain F of *C. gibelio* and of an adult female from *C. auratus*, separately. The short reads were sequenced for the two species using Illumina Hiseq2000 with PE 100 bp and PE 49 bp respectively for short (170, 250, 500 and 800 bp) and long (2, 5, 10, 20 and 40 kb) insert size libraries. BAC libraries with an insert fragment size of 120 kb in length were constructed only for *C. gibelio*. A total of 95,492 BAC clones (~6.4×) were randomly selected to extract plasmids. For each clone, unique index primer and adapter index were linked to the fragment end, and a 500 bp insert size library was constructed and used for Illumina sequencing with PE 100 bp to a coverage depth of ~100×. The single-molecule long reads were sequenced for both species using Pacific Biosciences Sequel instrument with libraries with a 20-kb average DNA insert size.

For Hi-C sequencing, blood cells were fixed with 2% formaldehyde for each species independently. The cross-linked DNA was digested with MboI, and the sticky ends were biotinylated by incubating with biotin-14-dATP and Klenow enzyme. After DNA purification and removal of biotin from unligated ends, Hi-C products were enriched and physically sheared to fragment sizes of 200–300 bp. The biotin-tagged Hi-C DNA was pulled down and processed into paired-end sequencing libraries that were sequenced PE 100 bp on the Illumina Hi-Seq2000 platform. At last, 440 Gb and 231 Gb Hi-C data were obtained from *C. gibelio* and *C. auratus*, respectively.

RNA was extracted from samples of *C. gibelio* and *C. auratus*, including eight adult tissues (heart, liver, kidney, muscle, ovary, hypothalamus, pituitary and other brain), POs and VOs[72], and embryos at seven developmental stages (four-cell, blastula, gastrula, bud, eight-somite, 1 day post-fertilization (dpf) and 3 dpf). Three biological replicates were analysed per sample. In total, 102 RNA-seq libraries were constructed and sequenced on Illumina Hiseq 2000 platform.

**Genome assembly and chromosome anchoring.** Pacbio long reads were used for de novo assembly by NextDenovo (https://github.com/Nextomics/NextDenovo) software (v2.3.1). Then the Pacbio long reads and all Illumina reads were used to correct raw de novo assembly by Nextpolish software (https://github.com/Nextomics/NextPolish) (v1.3.1, with parameter task=best). Subsequently, Hi-C sequencing data were used to improve the draft genome, and the Hi-C data were mapped to the polished assembly genome with Juicer (v 1.6) (ref. [73]). Next, a chromosome-length assembly was generated by the 3D-DNA software (v180922 with default parameters)[74]. To further improve the chromosome-scale assembly and quality control, manual review and refinement of the candidate assembly were performed by Juicebox Assembly Tools[74]. The haplotigs and overlapping sequence in the assemblies were removed by using Purge_dups (https://github.com/dfguan/purge_dups) software (v1.0.1).

**Genome annotation.** The repetitive sequences were annotated using both homology-based and de novo predictions. First, the long terminal repeats and tandem repeats were identified using LTR FINDER (v1.0.5) and TRF (v4.07b)[75]. Second, the transposable elements (TEs) were identified using RepeatMasker (v4.0.5) (ref. [76]) and RepeatProteinMask (v1.36) with the Repbase TE library. Finally, RepeatModeler (v1.0.8) (ref. [77]) was used to construct a de novo TE library, which was then used to predict repeats with RepeatMasker (v4.0.5).

To comprehensively annotate genes, we integrated different evidence. For de novo prediction, AUGUSTUS (v3.2.1) (ref. [78]) was used to predict coding genes with the repeat-masked genome. For the homologue-based approach, protein-coding sequences from three different species, *Danio rerio* (GRCz11), *Oryzias latipes* (GAculeatus_UGA_version5) and *Gasterosteus aculeatus* (ASM223467v1), were mapped against the repeat-masked genome using tBLASTN[79] with an *E*-value cut-off of 10$^{-5}$. Then, GeneWise (v2.2.0) (ref. [80]) was used to predict gene models with the aligned sequences as well as the corresponding query proteins. Additionally, Illumina RNA-seq data of *C. gibelio* and *C. auratus* were mapped to genome of *C. gibelio* and *C. auratus*, respectively, using HISAT2 (v2.1.0) (ref. [81]) and were assembled to transcripts using StringTie (2.1.4) (ref. [82]) software. In addition, we generated whole-genome alignments to project the Ensembl gene annotation for *D. rerio* by TOGA (https://github.com/hillerlab/TOGA). Finally, EVM (v1.1.1) (ref. [83]) was used to integrate all evidence to produce the final gene sets.

Gene functions were assigned according to the best match of the alignment to the public databases, including Swiss-Prot (release-2017_09), TrEmBLE (release-2017_09) (ref. [84]), KEGG (v84.0) (ref. [85]), COG[86] and NCBI NR (v20170924) protein databases. The motifs and domains in protein sequences were annotated using InterProScan (InterProscan-5.16-55.0) (ref. [87]) by searching publicly available databases, including Pfam, PRINTS, PANTHER, ProDom, SMART, ProSiteProfiles and appl ProSitePatterns. The actinopterygii_odb10 lineage dataset was selected to measure the completeness of the geneset using the BUSCO method[88].

**Subgenome-specific repeats and subgenome distinction.** Firstly, we classified the TEs into clusters according to the target sequences in the Repbase or de novo consensus library. Then we analysed the distribution of each cluster in the chromosomes. For each homoeologous chromosome pairs of subgenomes A and B (LG1 versus LG2, LG3 versus LG4, …), we found some clusters with a notable difference in the homoeologous pairs. If one cluster is an alternative in all the 25 homoeologous chromosomes pairs, it should and could be a specific marker to classify the two subgenomes, which originated from two distinct progenitor species. Finally, we identified the A-subgenome specific TEs in *C. gibelio* that targeted two consensuses from de novo library, and identified the B-subgenome specific TEs that targeted three de novo sequences. The same pattern of subgenome-specific repeats was also found in *C. auratus*. The subgenome distinction was also validated by comparing with previous studies[5,30,31,33–35] by synteny alignment.

In addition, we used MCScan[89] to identify syntenic blocks between *C. gibelio* genome and *C. auratus* genome, between subgenomes A and B of *C. auratus*, between subgenomes A and B of *C. gibelio*, and with other published genomes with the parameters of -a -e 1e-5 -u 1 -s 5. Firstly, we conducted an all-vs-all BLASTP to align proteins of the two genesets with the *E*-value parameters '1e-5'. The alignments were then subjected to MCScan to determine syntenic blocks, which were visualized by using CIRCOS software[90].

**Resequencing-based ploidy analysis.** BWA (Version 0.7.12-r1039) (ref. [91]) was used to map the Illumina reads of the two *C. auratus* and six *C. gibelio* generated in this study (Supplementary Table 16) to their respective genomes and subsequently sorted by SAMtools (Version 1.4) (ref. [92]) to obtain the bam files. The SNPs were called by FreeBayes (v0.9.10-3-g47a713e)[93] and filtered by following four thresholds: (1) ratio of two alleles depth between 1:9 and 9:1 for *Cg* and between 1:6 and 6:1 for *C. auratus* (*Ca*); (2) the highest sequencing depth of SNP position <200× for *Cg* and <400× for *Ca*; (3) the lowest sequencing depth for each allele ≥5; (4) the minimum distance for adjacent SNPs ≥5 bp. Then, the density distribution of the three alleles (reference, alternative and both) of all SNPs was counted, where the smallest peak of the distribution was defined as the depth of single haplotype. The genomic ploidy (*n*) was evaluated through a 1 Mb non-overlapping sliding window by the following equation:

$$1/n = \frac{\sum_1^k (\text{depth of alternative alleles})}{\sum_1^k (\text{depth of both alleles})};$$

$$n = \frac{\frac{1}{k} \sum_1^k (\text{depth of both alleles})}{(\text{depth of single haplotype})};$$

*k* is the number of SNPs in a window.

In addition, the distribution of heterozygosity was estimated using 500 kb non-overlapping sliding windows for each individual. The potential effects of these SNPs were evaluated by SnpEff[94] with default parameters.

**BAC-based ploidy analysis.** We split each BAC library data by index sequences, filtered and assembled each BAC clone in SOAPdenovoso2-r244 software[95]. The haplotype sequences were phased using pairs of adjacent tri- or bi-allelic SNPs that could be spanned by a single Illumina read (SNP pair). The BAC sequences that could be well phased and contain at least four genes were selected for further PCR validation and plotting.

**Phylogenetic analysis of *C. gibelio* and *C. auratus*.** To understand the evolution of the subgenomes A and B of *C. gibelio* and *C. auratus*, genomes of six Cyprinidae fishes were retrieved from public database: *Cirrhinus molitorella* (GCA_004028445.1), *Megalobrama amblycephala* (http://gigadb.org/), *D. rerio* (Ensembl GRCz11), *Ctenopharyngodon idellus* (http://bioinfo.ihb.ac.cn/gcgd/php/index.php), *Poropuntius huangchuchieni* (Datadryad, https://doi.org/10.5061/dryad.crjdfn32p) and *Cyprinus carpio* (GCA_018340385.1). The 11 peptide sequence sets from five genomes (*C. molitorella*, *M. amblycephala*, *D. rerio*, *C. idellus* and *P. huangchuchieni*) and six subgenomes (subgenome A of *C. gibelio*, *C. auratus* and *Cyprinus carpio*, subgenome B of *C. gibelio*, *C. auratus*, and *Cyprinus carpio*) were subjected to DIAMOND[96] to conduct all-to-all blast to identify the potential homologous sequences with an *E*-value <10⁻⁵.

The protein sequences of the 1:1:1 orthologous genes were aligned using MUSCLE (v3.8.425) (ref. [97]) with the default parameters. These alignments were

subsequently converted into coding sequence alignment by tracing the coding relationship using pal2nal.v14 (ref. [98]). Gblocks (v0.91b) (ref. [99]) was employed to conduct further checks (trim) on the coding sequence alignments with parameters '-t = c'. The 4d sites were extracted from the gene sequences retained in the last step. The divergence times between individual species (subgenomes) were estimated using MCMCTree[100] by using the 4d sites and species tree from ASTRAL[101] analysis. Time calibration consults fossil record information: 40.4–48.6 Mya for the time of the most recent common ancestor of *D. rerio* and *C. auratus*[102–105].

On the basis of DIAMOND[96] blast results, we selected the reciprocal optimal gene pairs for each species (subgenome) and *C. auratus* subgenome B. These pairs were aligned by MUSCLE[97] and the *K*$_s$ values were calculated by KaKs_Calculator2.0 (ref. [106]) with the default parameters. Correlation between divergence times of species pairs from various studies and peak values of *K*$_s$ distribution was assessed by least-squares-based regression analysis.

**Phylogenetic analysis of six *C. gibelio* individuals.** BWA (Version 0.7.12-r1039) (ref. [91]) was used to map the Illumina reads of the ten *C. auratus*, six *C. gibelio* and one *Cyprinus carpio* (Supplementary Tables 16 and 17) to the *C. auratus* genomes, and subsequently sorted by SAMtools (Version 1.4) (ref. [92]) to obtain the bam files. The SNPs were called by FreeBayes (v0.9.10-3-g47a713e)[93] with parameters '–gvcf–min-coverage 5–limit-coverage 200'. Subsequently, PLINK v1.90b6.6 (ref. [107]) was used to conduct PC analysis. Moreover, the 4d sites were extracted on the basis of the 'GFF' file of the *C. auratus* genome and the obtained SNPs. The evolutionary relationships of all resequenced individuals were then constructed by RAxML-8.2.12 (ref. [108]) under settings '-m GTRGAMMA -x 12345 -N 100 -p 12345'. The divergence times between individuals were estimated by MCMCTree[100] along the newly obtained evolutionary tree. The time calibration points refer to the previously obtained time settings for *Cyprinus carpio*–*C. auratus* (9.216–11.11 Mya) and *C. auratus*–*C. gibelio* (0.86–1.051 Mya) (Fig. 2).

**Lineage-specific gene expansion in *C. gibelio*.** The Illumina reads of the two *C. auratus*, six *C. gibelio* and one *Cyprinus carpio* (Supplementary Table 16) to the *C. auratus* genome using BWA (Version 0.7.12-r1039)[91]. We first identified the homologous sites whose minimum value of reads depth of all *C. gibelio* individuals were greater than twice the maximum value of the individuals of other species in the whole genome. Then, the genes whose coding sequence contains more than 60% of such sites were selected as genes that are potentially expanded in *C. gibelio*. For each of such genes, we examined its copy number in the genome assemblies of *C. gibelio*, *C. auratus* and *Cyprinus carpio* combined with given gene annotation file and manual annotation with GeneWise[80] using default settings.

**LOH analysis.** For LOH analysis, one female individual of G$_4$ generation of clone F (ref. [66]) was selected to construct a *C. gibelio* clonal line by reproducing successive four generations via gynogenesis. We sequenced 11 individuals (~48× depth for each sample) from the offspring of the gynogenetic line and called SNPs of each individual as the method in 'Resequencing-based ploidy analysis'. After multi-step filtering, we obtained 64,246 LOH sites, in which 101 LOH sites were randomly selected for PCR validation. The contiguous tracts of LOH sites were also extracted and classified into two types: caused by gene deletion or by gene conversion. Finally, the rates of LOH, gene deletion and gene conversion were calculated respectively. The details of the above processes are documented in Supplementary Note 4.

**Base-substitution mutation analysis.** On the basis of the SNPs obtained in the 'LOH analysis' step, we analysed each line for base-substitution mutations and calculated the mutation rate. We analysed mutation sites using the following criteria: (1) The non-triploid chromosome was filtered for each line separately. (2) The minimum coverage was 20× and maximum coverage 80×, on average. (3) Sites directly adjacent to small insertion–deletion mutations were filtered to avoid false-positive inferences created by misalignment. (4) For each SNP site of one line, the coverage depth of minor allele ≥6× was considered as heterozygous site of the line, and ≤2× was considered as homozygous site. (5) Ambiguous SNPs with coverage depth of minor allele >2× and <6× were filtered. Mutation sites were called only when they arose at highly credible ancestrally homozygous sites, and generated unambiguous heterozygous genotype in only one line. We calculated the mutation rate by the mutation sites of G4-4, G4-7, G4-8 and G4-9 using the equation µbs = *m*/(3*nT*) (ref. [109]). Where µbs is the base-substitution rate per site per generation, *m* is the observed number of base substitutions, 3*n* are the total number of analysed sites and *T* is the number of generations. Finally, the base-substitution mutation rate of *C. gibelio* is 8.88 × 10⁻⁹ per site per generation, a little higher than the rate of *C. auratus*.

**Antibody preparation, chromosome spreading and immunofluorescence.** The sequence (5–150 amino acids) of *C. gibelio* Sycp3 was cloned to produce His-tag fusion protein. A peptide (848–864 amino acids) of *C. gibelio* Sycp1 was synthesized and coupled to KLH protein. Polyclonal antibodies were raised in rabbits (ABclonal Biotechnology). Oocyte chromosome spreads were performed as described previously[110] with minor modifications. In brief, four to six ovaries (80–120 dpf) were dissected using a 20 ml injector 15–20 times and pipetted up

and down for 2 min in DMEM. After filtering with a 120-mesh cell strainer, cells were washed with PBS and suspended in 80–120 μl 0.1 M sucrose (pH ~8). Then, 20–25 μl cell suspension was vertically dropped to the centre of the slides that has been covered with 100 μl 1% paraformaldehyde. After drying, slides were rinsed in H₂O and in 1:250 Photo-Flo 200 and ready for immunofluorescence.

The slides of chromosome spreads were repaired in boiled citrate–EDTA antigen retrieval buffers for 20 min, permeabilized with 0.1% Tween 20 and 0.1% Triton X-100 in PBS for 10 min, and blocked for 10 min with 10% ADB (10% goat serum, 3% BSA and 0.05% Triton X-100 in PBS) at room temperature. Then, the slides were incubated overnight at 4 °C with primary antibodies (anti-Sycp3 [1:150]; anti-Sycp1 (1:100); anti-hRad51 (1:50; Abcam)). After washing with PBS three times, slides were incubated for 1 h in the dark at 37 °C with secondary antibodies (1:500 Alexa Fluor 546 goat anti-rabbit, Invitrogen, 1:500 Alexa Fluor 488 goat anti-mouse Invitrogen and 5 μg ml⁻¹ DAPI, Sigma). After incubation, slides were washed for 10 min each in PBS containing 0.04% Photo-Flo 200 and 0.03% Triton X-100. Finally, the samples were mounted with VECTASHIELD Antifade Mounting Medium (Vector Labs) and photographed using the Leica SP8 STED (Analytical & Testing Center, IHB, CAS).

**Reporting summary.** Further information on research design is available in the Nature Research Reporting Summary linked to this article.

## Data availability

The whole genome assembly and the raw resequencing data of *C. gibelio* are deposited into GenBank under BioProject ID PRJNA546443. The whole genome assembly and the raw resequencing data of *C. auratus* are deposited into GenBank under BioProject ID PRJNA546444. The transcriptome data of *C. gibelio* and *C. auratus* are available in the GenBank (PRJNA836313, PRJNA834570, PRJNA833164, PRJNA837728, PRJNA833750, and PRJNA833167). The gene alignments and trees of specific lost and expanded genes are available at figshare database (https://doi.org/10.6084/m9.figshare.19674843.v1). Source data are provided with this paper.

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

## Acknowledgements

We thank J. Luo for providing the genome of goldfish; I. Seim and G. Zhang for helpful discussion and M. Eckstut (Edanz, www.liwenbianji.cn) for assistance in editing this manuscript. The research was supported by Analytical & Testing Center and Supercomputing Centre, CAS, China. This work was supported by the Strategic Priority Research Program of the CAS (XDA024030104, XDB31000000), the Key Program of Frontier Sciences of the CAS (QYZDY-SSW-SMC025), the National Key Research and Development Program of China (2018YFD0900204, 2021YFD1200804), the Earmarked Fund for Modern Agro-industry Technology Research System (NYCYTX-49), the National Natural Science Foundation of China (31772839) and the Autonomous Project of the State Key Laboratory of Freshwater Ecology and Biotechnology (2019FBZ04).

## Author contributions

J.-F.G. and L.Z. designed the study. W.W., Y.W., X.-D.F. and L.Z. supervised the study. B.-T.Z., B.W., M.X., Y.-L.Y. and Y.W. performed genomic assemblies, validation, and karyotype analysis. B.W., Y.-L.Y., Y.W. and X.-Y.L. assembled and annotated RNA-seq data. W.-J.X., B.W., M.X., Y.W., J.-B.J., C.Z., W.-M.H., J.-G.T and B.-D.F. performed the comparative analysis of the two genomes assembly. B.-T.Z., M.X., Y.-L.Y., Y.W., W.Y., Z.-C.Z. and Q.-Qia Z. performed gene annotation and TE analysis. W.-J.X., B.W., Y.W., Q.G., J.-Y.X. and M.-Z.B. performed genome evolution and gene family analysis. W.-J.X, C.-L.Z, M.-L.H., J.-M.Z. and C.-G.F. performed comparative genomic and transcriptome analysis. X.-Y.L., Z.-W.W., Z.L., X.-J.Z., W.-J.L., R.-H.G., J.-F.T. and Q.-Qin Z. provided samples for sequencing and analysis. Y.C., Y.W., L.-J.M. and Z.L. performed chromosome spread and DNA content assays. X.-Y.L., L.-T.T., Z.-W.W., Z.L. X.-J.Z., L.-J.M., Y.C., P.Y., M.L., F.P. and M.D. performed other related experimental work. Y.W., K.W., X.-Y.L.,
X.-D.F., L.Z. and J.-F.G. wrote the manuscript with input from all other authors. J.-F.G., W.W., Z.Y., Y.-P.Z., Q.Q. and H.-M.Y. revised the manuscript.

## Competing interests

The authors declare no competing interests.

## Additional information

**Extended data** is available for this paper at https://doi.org/10.1038/s41559-022-01813-z.

**Correspondence and requests for materials** should be addressed to Xiao-Dong Fang, Wen Wang, Li Zhou or Jian-Fang Gui.

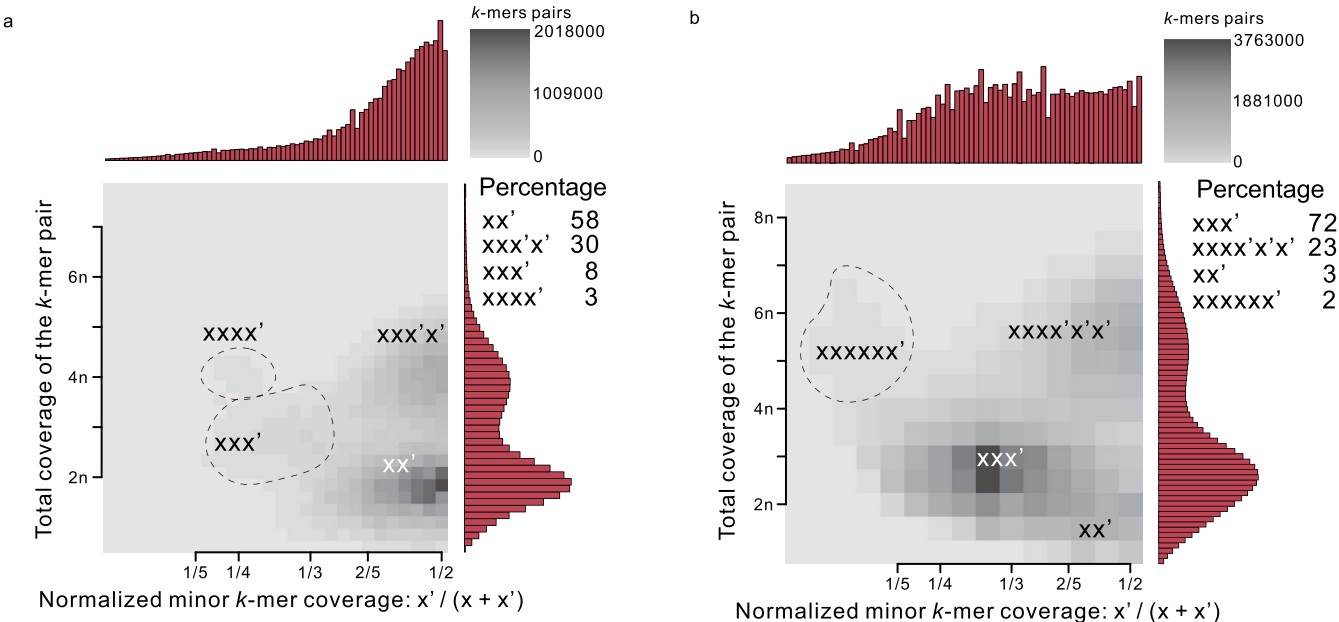

**Extended Data Fig. 1 | Haplotype structures of *C. auratus* (a) and *C. gibelio* (b) shown by Smudgeplots.** x and x' represent a pair of heterozygous *k*-mers with only one SNP difference. The darkness of each smudge is determined by the number of heterozygous *k*-mer pairs that fall within it. The percentage of each genotype is presented in the middle-right. In *C. auratus*, xx' indicates sequences that are consistent with the pattern of diploid species, whereas xxx'x' and xxxx' indicate that these sequences are consistent with the pattern of tetraploid species. In *C. gibelio*, the *k*-mers with the highest percentage (72%) are xxx', representing they belong to the regions with three haplotypes, and the *k*-mers with the second highest percentage (23%) are xxxx'x'x', representing they belong to the regions with six haplotypes.

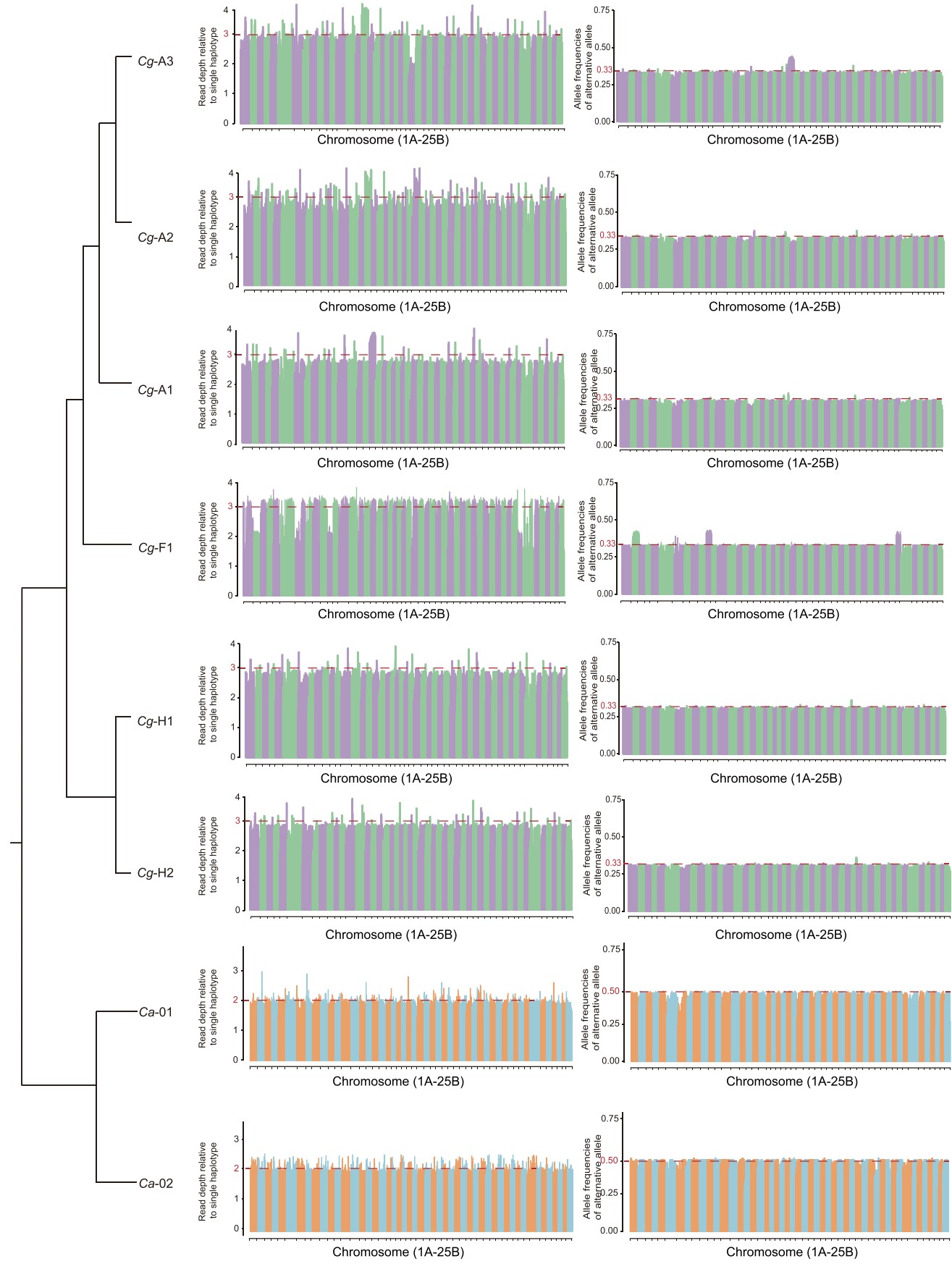

**Extended Data Fig. 2 | See next page for caption.**

**Extended Data Fig. 2 | Chromosomal ploidy of *C. gibelio* and *C. auratus*.** The read depth relative to single haplotype in each chromosome (left) and the allele frequencies of alternative alleles in each chromosome (right). Each color block represents a chromosome. The *C. gibelio* individuals usually have three times of read depth relative to single haplotype and the allele frequencies of alternative alleles in each chromosome is about 0.33, confirming that most of the chromosomes have three haplotypes. The *C. auratus* individuals have two times of read depth relative to single haplotype and the allele frequencies of alternative alleles in each chromosome is about 0.5, confirming that these chromosomes have two haplotypes.

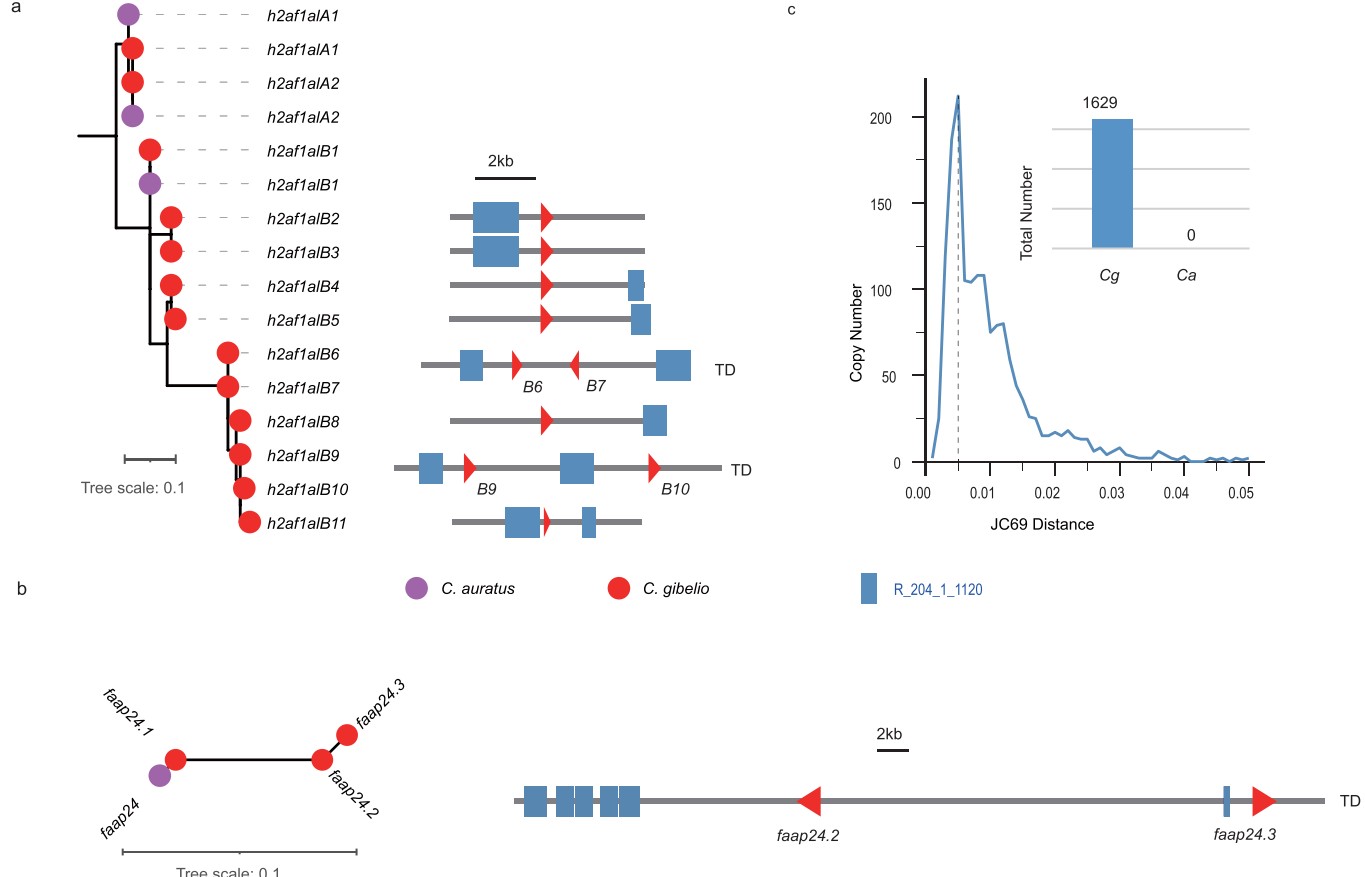

**Extended Data Fig. 3 | Lineage-specific repeats near the expanded *h2af1al* and *faap24* genes in *C. gibelio*. a**, Adjacent specific repeats of expanded *h2af1al* genes. Left panel, *h2af1al* gene tree. Right panel, the location of lineage-specific repeats relative to the *h2af1al* genes. The rectangle and triangle represent repeats and genes, respectively, and the direction of triangles represents the direction of the gene. **b**, The same analyses of adjacent specific repeats of expanded *faap24* genes. **c**, Copy number and distribution of JC69 distance for R_204_1_1120, a lineage-specific repeat near both *h2af1al* and *faap24* in *C. gibelio*. TD: tandem duplications.

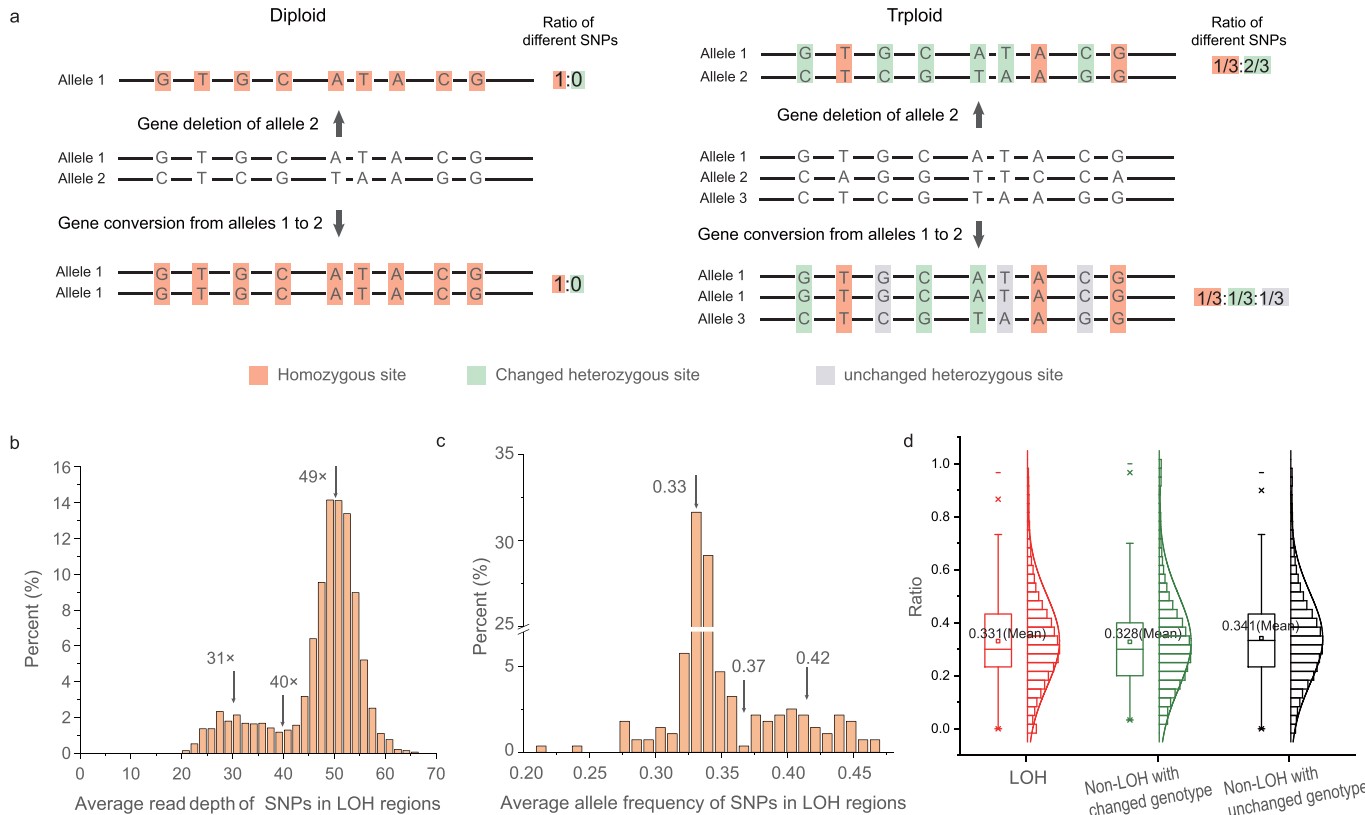

**Extended Data Fig. 4 | Analysis of the LOH regions detected in the gynogenetic pedigree of *C. gibelio*. a**, Schematic diagram of a LOH region in diploid (left panel) or triploid (right panel). Allele 2 was deleted in a gene deletion event, whereas a gene conversion occurred from Allele 1 to Allele 2. Theoretical ratio of different types of SNPs in a LOH region is shown in the right side of each panel. **b**, Distribution of average depths of SNP sites in LOH regions. **c**, Distribution of average minor allele frequencies in LOH regions. **d**, Ratio distributions of the three types of SNP sites in gene conversion blocks in a 30-SNP sliding window and 1-SNP step. The line in the middle of each boxplot represents the median of the dataset; the upper and lower edges of boxplot indicate the third quartile and first quartile, respectively; and the line extending from the edge is 1.5 times the interquartile range. Small dots indicate outliers. n = 10 individuals.

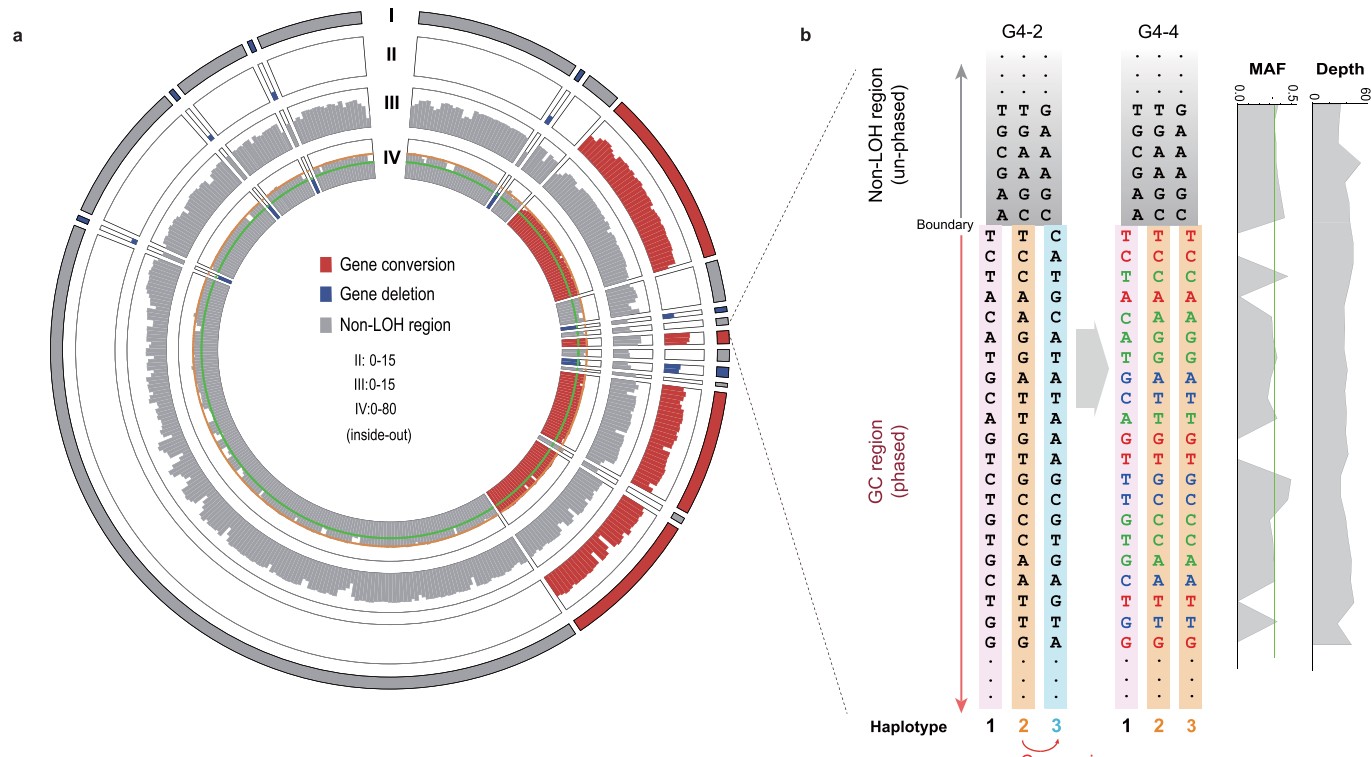

**Extended Data Fig. 5 | A gene conversion in chromosome 2 A of individual G4-4. a**, Circos map showing all LOH regions in chromosome 2 A of individual G4-4. I, LOH region. II, LOH site number (Log2). III, SNP number (Log2). IV, read depth of SNP sites. The window size in II-IV is 200 kb. Red and blue bricks indicate gene conversion and gene deletion respectively. The orange and green lines in IV indicate the sequencing depth of 50× and 33× respectively. **b**, Phased haplotype blocks at the gene conversion boundary in **a**. The same region of individual G4-2 is shown as a control without gene conversion. Red bases, homozygous converted sites which resulted in LOH in this region. Green bases, heterozygous converted sites where the donor allele was minor allele. Blue bases, heterozygous sites where the donor allele had the same base as the recipient allele. Black bases with grey shadowed, heterozygous sites outside the converted region. Minor allele read frequencies (MAF) and depths of SNP sites in individual G4-4 are shown on the right panel. The total reads depth of this LOH region is consistent with non-LOH region, which is different from the LOH region caused by gene deletion. MAF of LOH sites equals to zero while others close to 1/3.

# Reporting Summary

## Statistics

For all statistical analyses, confirm that the following items are present in the figure legend, table legend, main text, or Methods section.

| n/a | Confirmed | |
|---|---|---|
| ☐ | ☒ | The exact sample size (*n*) for each experimental group/condition, given as a discrete number and unit of measurement |
| ☐ | ☒ | A statement on whether measurements were taken from distinct samples or whether the same sample was measured repeatedly |
| ☐ | ☒ | The statistical test(s) used AND whether they are one- or two-sided<br>*Only common tests should be described solely by name; describe more complex techniques in the Methods section.* |
| ☐ | ☒ | A description of all covariates tested |
| ☐ | ☒ | A description of any assumptions or corrections, such as tests of normality and adjustment for multiple comparisons |
| ☐ | ☒ | A full description of the statistical parameters including central tendency (e.g. means) or other basic estimates (e.g. regression coefficient) AND variation (e.g. standard deviation) or associated estimates of uncertainty (e.g. confidence intervals) |
| ☐ | ☒ | For null hypothesis testing, the test statistic (e.g. *F*, *t*, *r*) with confidence intervals, effect sizes, degrees of freedom and *P* value noted<br>*Give P values as exact values whenever suitable.* |
| ☐ | ☒ | For Bayesian analysis, information on the choice of priors and Markov chain Monte Carlo settings |
| ☐ | ☒ | For hierarchical and complex designs, identification of the appropriate level for tests and full reporting of outcomes |
| ☐ | ☒ | Estimates of effect sizes (e.g. Cohen's *d*, Pearson's *r*), indicating how they were calculated |

*Our web collection on statistics for biologists contains articles on many of the points above.*

## Software and code

Policy information about availability of computer code

| Data collection | Software was only used for data analyses. |
|---|---|
| Data analysis | All the softwares used for analysis have been described in the Online Methods as well as Supplementary Methods. All software used in this study included: NextDenovo v2.3.1, Nextpolish v1.3.1, Juicer v 1.6, 3D-DNA v180922, Juicebox Assembly Tools 1.9.8, Purge_dups v1.0.1, BWA v0.7.17-r1198-dirty, SAMtools v1.12, BEDTools, BUSCO v5.2.2, LTR FINDER v1.0.5, TRF v4.07b, RepeatMasker v4.0.5, RepeatProteinMask v1.36, RepeatModeler v1.0.8, AUGUSTUS v3.2.1, BLAST 2.10.1, GeneWise v2.2.0, HISAT2 v2.1.0, StringTie v2.1.4, Cufflinks v5.5.0, InterProscan-5.16-55.0, SOAPdenovoso2-r244, MCScan v0.8, FreeBayes v0.9.10-3-g47a713e, MUSCLE v3.8.42, pal2nal.v14, Gblocks v0.91b, ASTRAL v5.7.1, DensiTree, PAML v4.9h, ASTRAL v5.7.1, KaKs_Calculator2.0, Phybase v.1.5, RAxML-8.2.12, SnpEff v4.3t, MEGA v7.0.26, FigTree v1.4.4, mafft v7.471. |

For manuscripts utilizing custom algorithms or software that are central to the research but not yet described in published literature, software must be made available to editors and reviewers. We strongly encourage code deposition in a community repository (e.g. GitHub). See the Nature Portfolio guidelines for submitting code & software for further information.

## Data

Policy information about availability of data

All manuscripts must include a data availability statement. This statement should provide the following information, where applicable:
- Accession codes, unique identifiers, or web links for publicly available datasets
- A description of any restrictions on data availability
- For clinical datasets or third party data, please ensure that the statement adheres to our policy

The whole genome assembly and the raw resequencing data of C. gibelio are deposited into GenBank under BioProject ID PRJNA546443. The whole genome

# Field-specific reporting

Please select the one below that is the best fit for your research. If you are not sure, read the appropriate sections before making your selection.

☐ Life sciences     ☐ Behavioural & social sciences     ☒ Ecological, evolutionary & environmental sciences

For a reference copy of the document with all sections, see nature.com/documents/nr-reporting-summary-flat.pdf

# Ecological, evolutionary & environmental sciences study design

All studies must disclose on these points even when the disclosure is negative.

| | |
|---|---|
| Study description | Triploids are generally considered as an evolutionary "dead-end" because of two major biological difficulties for meiotic pairing and segregation of three homologous chromosomes and for purging deleterious mutations due to the lack of meiotic recombination. However, triploids are found in some polyploid complex (coexistence of diploid, triploid, tetraploid, or hexaploid) species, which usually have overcome the reproductive obstacles via unisexual production. However, the evolutionary mechanisms underpinning the gynogenetic system have largely remained unknown. Here, we used Carassius complex of cyprinid teleost to illustrate this enigma, which contains sexual tetraploid C. auratus and unisexual hexaploid C. gibelio. |
| | Firstly, we sequenced the genomes of these two species, and managed to assemble their haplotypes that contain two subgenomes (AB) to the chromosome level. Sequencing coverage analysis reveals that C. gibelio is an amphitriploid (AAABBB) with two triploid sets of chromosomes; each set is derived from a different ancestor. And the evolution process of ancient polyploidy has been characterized in the Carassius complex. Resequencing data from different strains of C. gibelio show that unisexual reproduction has been maintained for over 0.82 million years. Secondly, comparative genomic results reveal significant genomic changes specific to unisexual reproduction success, in which many meiotic cell cycle-related genes and an oocyte-specific histone variant gene family are largely expanded in the amphitriploid C. gibelio. This study provides the first genomic evidence for the novel hypothesis that a genomic assemblage and an alternative reproductive module might be required for the formation of a functioning asexual/unisexual genome as suggested by Hojsgaard and Schartl (2021). Thirdly, cytological assays indicate that C. gibelio produces unreduced oocytes by an alternative ameiotic pathway; however, sporadic homologous recombination and a high rate of gene conversion also exist in C. gibelio. These genomic changes might have facilitated purging deleterious mutations and maintaining genome stability in this unisexual amphitriploid fish. Our findings shed novel lights onto the evolutionary mechanisms underpinning the reproduction success in unisexual polyploid vertebrates. |
| Research sample | A female adult individual from strain F of Carassius gibelio and a female adult individual of Carassius auratus were used for de nova genome assembly. Six female C. gibelio individuals from three strains in total were used to study the origin of this unisexual species, including three female individuals of strain A+, two female individuals of strain H, and one female individual of strain F. Eleven female individuals from the offspring of the fourth generation belonging to a gynogenetic line of strain F of C. gibelio were used to perform analysis of gene conversion. |
| Sampling strategy | Blood cells from all samples were used for genome sequencing. |
| Data collection | All data were collected in the author's laboratory or downloaded from NCBI Genbank and the literature described in the manuscript. |
| Timing and spatial scale | from Mar, 2018 to Nov 2020 |
| Data exclusions | No data collected were excluded from analyses. |
| Reproducibility | All attempts to repeat the experiment were successful. |
| Randomization | Samples were randomly selected from healthy adults. |
| Blinding | No blinding was required. |

Did the study involve field work?     ☐ Yes     ☒ No

# Reporting for specific materials, systems and methods

We require information from authors about some types of materials, experimental systems and methods used in many studies. Here, indicate whether each material, system or method listed is relevant to your study. If you are not sure if a list item applies to your research, read the appropriate section before selecting a response.

## Materials & experimental systems

| n/a | Involved in the study |
|-----|----------------------|
| ☒ | ☐ Antibodies |
| ☒ | ☐ Eukaryotic cell lines |
| ☒ | ☐ Palaeontology and archaeology |
| ☐ | ☒ Animals and other organisms |
| ☒ | ☐ Human research participants |
| ☒ | ☐ Clinical data |
| ☒ | ☐ Dual use research of concern |

## Methods

| n/a | Involved in the study |
|-----|----------------------|
| ☒ | ☐ ChIP-seq |
| ☒ | ☐ Flow cytometry |
| ☒ | ☐ MRI-based neuroimaging |

## Animals and other organisms

Policy information about studies involving animals; ARRIVE guidelines recommended for reporting animal research

| | |
|---|---|
| Laboratory animals | A female adult individual from strain F of Carassius gibelio (two years old) and a female adult individual of Carassius auratus (two years old) were used for de nova genome assembly. Six female C. gibelio individuals from three strains in total were used to study the origin of this unisexual species, including three female individuals of strain A+( two years old), two female individuals of strain H (two years old), and one female individual of strain F (two years old). Eleven female individuals from the offspring of the fourth generation belonging to a gynogenetic line of strain F of C. gibelio (two years old) were used to perform analysis of gene conversion. |
| Wild animals | Our study did not sample wild animals. |
| Field-collected samples | Our study did not include field-collected samples. |
| Ethics oversight | No ethical approval was required. |

Note that full information on the approval of the study protocol must also be provided in the manuscript.

