## [Peer Review File · Nature Ecology & Evolution]

Peer Review Information

Journal: Nature Ecology & Evolution

Manuscript Title: NATECOLEVOL-211115122B

Corresponding author name(s): Xiao-Dong Fang, Wen Wang, Li Zhou, Jian-Fang Gui

Editorial Notes:

Reviewer Comments & Decisions:

Decision Letter, initial version:

30th December 2021

Dear Dr. Gui,

Your Article, "Comparative genome anatomy reveals evolutionary insights into a unique amphitriploid fish" has now been seen by three reviewers. You will see from their comments copied below that while they find your work of considerable potential interest, they have raised quite substantial concerns that must be addressed. In light of these comments, we cannot accept the manuscript for publication, but would be very interested in considering a revised version that addresses these serious concerns.

We hope you will find the reviewers' comments useful as you decide how to proceed. If you wish to submit a substantially revised manuscript, please bear in mind that we will be reluctant to approach the reviewers again in the absence of major revisions.

In particular, in addition to addressing all of the technical concerns of the reviewers, I should stress that we need to see more data, as suggested by the reviewers, to provide mechanistic insights into the reproductive mode and evolutionary success of this species. We agree with reviewer 3 that more functional insights are needed to bring this study to the level we look for in Nature Ecology & Evolution

.

If you choose to revise your manuscript taking into account all reviewer and editor comments, please

highlight all changes in the manuscript text file in Microsoft Word format.

* Include a "Response to reviewers" document detailing, point-by-point, how you addressed each referee comment. If no action was taken to address a point, you must provide a compelling argument. This response will be sent back to the referees along with the revised manuscript.

* If you have not done so already we suggest that you begin to revise your manuscript so that it conforms to our Article format instructions at <http://www.nature.com/natecolevol/info/final-submission>. Refer also to any guidelines provided in this letter.

[REDACTED]

If you wish to submit a suitably revised manuscript we would hope to receive it within 6 months. If you cannot send it within this time, please let us know. We will be happy to consider your revision so long as nothing similar has been accepted for publication at Nature Ecology & Evolution or published elsewhere.

Nature Ecology & Evolution is committed to improving transparency in authorship. As part of our efforts in this direction, we are now requesting that all authors identified as 'corresponding author' on published papers create and link their Open Researcher and Contributor Identifier (ORCID) with their account on the Manuscript Tracking System (MTS), prior to acceptance. This applies to primary research papers only. ORCID helps the scientific community achieve unambiguous attribution of all scholarly contributions. You can create and link your ORCID from the home page of the MTS by clicking on 'Modify my Springer Nature account'. For more information please visit www.springernature.com/orcid.

Thank you for the opportunity to review your work.

[REDACTED]

Reviewer expertise:

Reviewer #1: genome sequencing, vertebrate polyploid genomes

Reviewer #2: vertebrate genome sequencing and evolution

Reviewer #3: genome sequencing, fish genomics and evolution

Reviewers' comments:

Reviewer #1 (Remarks to the Author):

I am impressed by the results in the manuscript considering the rarity of hexaploids in vertebrates and very high sequence identities between the subgenomes. Wang et al. revealed that *C. auratus* is amphidiploid (AABB) and *C. gibelio* is amphitriploid (AAABBB) and observed that many gene families related to meiotic cell cycle and oocyte expanded in *C. gibelio*. I believe, although there still exists room to improve, current results will contribute to our understanding of the evolution of polyploid vertebrates. Before publication, some suggestions would be beneficial for them to improve the analysis and reveal more true biology facts.

Major concerns:

1. The estimated genome size is much affected by k-mer size in the k-mer analysis. They used 17-mer to estimate the genome size of *C. auratus* and *C. gibelio*. The author should perform the k-mer size simulation to estimate the genome sizes.
2. The assembly size of *C. auratus* in this study is smaller than that in other previous reports (Chen et al., *Science Advances*, 2019; Luo et al., *Science Advances*, 2020; Chen et al., *PNAS*, 2020), possibly resulting in the artificial gene losses. Thus, the authors should perform comprehensively comparisons of all available *C. auratus* assemblies, including RNA-seq alignment, genome read alignment, and EST/cDNA alignment, to demonstrate the quality of this new assembled genome of *C. auratus*. Were the predicted genes in other assemblies covered in this new assembly? The alignment ratio of the BAC sequences to the *C. gibelio* genome is a good indicator of genome assembly quality.
3. In Figure 2B, *Cirrhinus molitorella* was clustered with the B subgenomes of Common carp, *C. auratus*, and *C. gibelio*. Therefore, they stated that the subgenome B had a closer relationship to the diploid species *Cirrhinus molitorella* and *Poropuntius huangchuchieni* than subgenome A. However, Xu et al. (*Nature Communication*, 2019, Figure 2a) showed that *Cirrhinus molitorella* was the outgroup of subgenomes A and B. The authors should carefully examine the phylogenetic location of *Cirrhinus molitorella*.

34. Besides scanning the subgenome-specific TEs for subgenome division, the author should further validate the accuracy of subgenome division according to Chen et al.'s method (Chen et al., PNAS, 2020), which utilized the alignment read number from close diploid fish, for instance, *Poropuntius huangchuchieni*, to two subgenomes.

5. In line 184 of Supplementary file, to understand the evolution of the subgenomes A and B of *C. gibelio* and *C. auratus*, the authors aligned the 1:1:1 orthologous genes among *Cirrhinus molitorella*, *Megalobrama amblycephala*, *D. rerio*, *Ctenopharyngodon idellus*, *Poropuntius huangchuchieni*, *Cyprinus carpio*, *C. gibelio*, and *C. auratus*. I suggest that they should select the 1:1:1:1:1:2:2:2 gene families having a single gene in each diploid, 2 common carp genes, 2 *C. auratus* genes, and 2 *C. gibelio* genes. How many such gene families did they find?

6. Figure 2B exhibits signal of gene exchange between two subgenomes with a frequency of 1.86%. Was this frequency significantly different from the null hypothesis? To demonstrate the real signals rather than the phylogenetic tree error, the author should validate these exchanges by comparing the aligned *Poropuntius huangchuchieni* read numbers on the homoeologues from two subgenomes.

7. In Figure 2C and D, the author should define the error bars and analyze whether there existed statistical significances.

8. In Supplementary file, I did not see the genome sequencing method information on *C. auratus*. The author should rewrite it.

9. I am curious about which assembly of common carp genome was used in Extended data Table 7. The ENSEMBLE transcript annotations were based on the assembly of GCA_000951615.2, rather than GCA_018340385.1 that they stated in line 181 in Supplementary file.

10. The nomenclatures in Extended data Table 6 were confusing. What are 'CaiJi_A', 'CaiJi_B', 'YinJi_A', and 'YinJi_B'?

Reviewer #2 (Remarks to the Author):

This manuscript by Wang et al. involves a comparative study of two *Carassius* fish genomes, one of them (*C. gibelio*) is an unusual unisexually hexaploid fish, being an exciting model to study unisexual polyploids in vertebrates. The subgenome A and B have been assembled at the chromosome-level, though unfortunately the two parental genomes (AA/A and BB/B) prior to triploidization were not resolved, likely due to the recent triploidization event and the limitation of error-prone long reads. Many meiotic cell-cycle and oocyte-specific histone variant genes were found to be specifically expanded in *C. gibelio*, but its implications for unisexual triploid adaptation are rather speculative and obscure (functional experiments are lacking). The data for analysing the gene conversion rate is useful, but I doubt whether the results are solid (see more comments below).

4Major concerns

1. The extremely large variation of gene conversion rate (almost 200 fold difference among individuals) is concerning - I wonder this is the true nature of this species or it reflects some artifacts in individuals with particular high gene conversion rates.

Figure 5E shows one example of gene conversion where some changed genotypes (in red) were interpreted as the outcome of gene conversion while some (in green) were not. This is quite difficult to understand - wouldn't be gene conversion affect the entire region? is there another mechanism underlying the changed genotype (green) other than gene conversion? Can authors show the read alignment for this region so that the three karyotypes can be confirmed?

For some LOH regions, deletions were to blame where 2/3 coverage is shown; and for regions with normal coverage it was assumed gene conversion caused LOH. This is quite a crude identification of gene conversion, though the authors explained in the methods that the coverage patterns were stable. This does not seem to be the case as one can spot a lot of coverage variation in figure 5D. It could be possible that the regions defined as gene conversion actually lost one allele (deletions) but somehow show higher coverage than 2/3 due to for example differential GC content? More evidence is needed to define gene conversion, for install, showing three PHASED haplotype blocks at the gene conversion boundary containing non-converted and converted sites.

Furthermore, I wonder how elevated gene conversion rate is necessarily adaptive - this is not well explained in the manuscript. It is also not clear whether elevated gene conversion rate is a derived feature for *C. gibelio*. It would be ideal to compare the gene conversion rate between *C. auratus* and *C. gibelio* to confirm that *C. auratus* does not have a high gene conversion rate.

2. The authors used gene phylogeny to test whether there are gene exchanges between A and B subgenomes. The fact that only 13 genes show the alternative topology gave me a pause - is the topology of all 13 gene trees well supported? Apparently there could be other explanations including ancestral gene flow or incomplete lineage sorting which were not assessed.

Minor concerns

L52, also popular in some other countries.

L95, Hi-C data was not shown in the Extended Table 1. Also, I believe for the Illumina table the labels "Illumina insert size" and "Read length" were swapped.

L98, awkward sentence. In general, English language needs to be polished throughout the manuscript.

L105, unclear what is a hierarchical strategy - The extended fig.2 only show a workflow for long-read genome assembly and annotation.

L117, Fig. 1B shows five TEs, where are the nine TEs?

L119, perhaps use different colors for A and B subgenome, making it clear that A and B assemblies have 1:1 correspondence.

L122, unclear how the level of redundancy was measured.

L186, are we sure this is the only reason for the disparity? Otherwise one needs to add “partially”.

L192, more details are needed in the legend.

L202-203, why a lower Ka/Ks rate is consistent with dominant B? An explanation is needed. Similarly, please help readers understand why the dominant subgenome should increase gene expression in later stages.

L207, it's strange why Fig. 4C is mentioned here.

L225, this is confusing. The authors identified shared gene loss by *C. auratus* and *C. gibelio*, and came to the suggestion that the gene loss events are associated with ploidy alternation. But the fact is only *C. gibelio* has experienced one additional polyploidization event.

L243, should be three alleles.

L254-255, confusing, did that authors try to say different populations have diverged.

L255, looking at the methods, here the conserved non-coding elements should rather be species-specific elements.

L256, Fig. 3C seems to show only one example?

L257, this is misleading - the elements could have an ancestral origin, but only remain conserved in *C. gibelio*.

L261, in extended figure 14, the dating of T3 is 0.962. Also, the time points were already calibrated with “0.86 – 1.051” (Method L256), making the time estimation not independent. So I wouldn't trust the claim at line 264-265.

L281, where is the data for the lower fitness?

L297, so what is the case in *C. gibelio*? Please briefly describe the meiosis process in *C. gibelio*.

L301-302, that number is average number across individuals? Please confirm.

L310, please specify what special meiosis.

L338, The extended table 10 shows 3 copies instead of 4, please confirm.

L340, is it inferred that *C. gibelio* has altered cell division of meiosis I, or this is already observed?

L355, because the haploid genome was assembled for *C. gebelio*, I wonder whether the loss-of-function mutations occurred in all three alleles? And if the authors wanted to investigate the

accumulations of deleterious mutations, they need also to look at the slightly deleterious mutations. Besides the density of mutations, the authors should also test for an excess of non-synonymous substitutions.

L366, I assumed the parental fish was also sequenced? But the Extend Table 11 only shows info about the 11 offspring.

L416, please remind the readers what are the significant genomic changes?

Methods L71, isn't Illumina reads were sequenced? Why BGI short reads?

Methods L94, I believe StringTie is a assembler, not a read mapper. Please correct.

Reviewer #3 (Remarks to the Author):

The past few years have seen a lot of attention to cyprinid genomes and the analysis of evolutionary genetic patterns following the carp genome duplication, an allopolyploidization event in the ancestor of common carp, goldfish and others (see Chen et al. 2019, *Sci Adv*; Kon et al. 2020, *Curr Biol*; Chen et al. 2020, *PNAS*; Li et al. 2021, *Nat Gen*). The current study by Wang et al. presents yet another high-quality genome assembly for the allotetraploid goldfish, confirming some of the subgenome dominance patterns observed previously. More importantly, the present study focuses on the investigation of the Prussian carp (*C. gibelio*), an allo-auto-hexaploid and asexually reproducing species with yet another amplification of the genome from the ancestral carp situation. The study is well written and touches upon some very interesting questions and first insights regarding the long-term sustainability of this hexaploid lineage despite its unisexual reproductive mode. This asexual reproductive mode is reminiscent of the situation in the gynogenetic Amazon molly, albeit the latter not being polyploid.

While the current work is certainly an important first step for our understanding of the genetic regulation of the unique genomic architecture and reproductive mode of this species, I find that the manuscript remains rather descriptive, and unfortunately does not provide some more mechanistic explanations for the evolutionary survival of such unique genomic constitution.

Major points:

1) While *C. gibelio* is hexaploid (AAABBB), only the A and B subgenomes from the pan-carp allopolyploidization could be separately assembled (but maybe I am misunderstanding). It needs to be discussed why additional phasing of the different A and B chromosomes is not possible. The study introduces an XX' nomenclature which is confusing as it is unclear whether it refers to alleles (?) or ohnologs (?) or both (?). While in goldfish, for many/most genes or loci 4 different versions would be expected (2 alleles per A and B subgenome), in *C. gibelio* one should be able to find 6 (3 alleles of A and B subgenome). Has this not been observed?

72) It is interesting that apparently a substantial fraction of genes implicated in meiosis and oocyte formation have been amplified specifically in *C. gibelio*. However, besides the provided discussion of the role of some of these individual genes, one would like to know how these extra copies have been generated (tandem duplication, en bloc duplication, retrotransposition, etc.), what their expression profiles look like (beyond the case of the histone gene), how many of these extra copies may be functional, etc.

3) Given the unique reproductive mode of *C. gibelio*, I would have expected an analysis also of the genomic region that corresponds to the sex determination region of *C. auratus* that has been discussed in previous studies. The occurrence of male *C. gibelio* is mentioned, and the same group has just recently provided an in-depth investigation of supernumerary chromosomes in male *C. gibelio*, but I would have expected to see a male genome assembly in the current study for comparison as well.

4) The section "Allotetraploidy and genetic variations in the common ancestor of *Carassius*" mostly confirms findings of the many other recent carp and goldfish genome articles and should thus be shortened and partially moved to the supplements.

Other specific points:

- Check if the use of references in the abstract is ok.
- Amphidi/triploid is not a common term, please define.
- It would be good to also provide common names of species at first mentioning.
- I.58: I don't think it is correct to say that the hybridization increased the chromosome number, because most hybridization events per se do not lead to increased ploidy. It is the allo-polyploidization that leads from $2n=50$ to $2n=100$.
- I.73-74: Is this kleptospermy like in the Amazon molly? If so, please also use the term kleptospermy for comparability.
- I.98-99: Please define the $x \times x'$ terminology more clearly.
- Why is about 50% of the goldfish genome not incorporated into the assembly (based on estimated genome size)?
- I.105-123: The goldfish genome sequence has now been published multiple times, so this part should be further shortened.
- I.130: Why is the genome assembly size roughly the same length as in goldfish despite the higher ploidy?
- I.185-188: Why is *Sinocyclohelius* not included in the phylogeny and the divergence time estimates? Genomes are certainly available.
- I.193-198: Homeologous gene exchange has also been shown for the common carp (Li et al. 2021, Nat Gen), so this is not total news. The published results in carp should be mentioned and discussed.
- I.199-201: Despite the many recent publications on carp and goldfish genomes, only one is mentioned here (ref. 27). Please pay full reference to these other studies as well.
- I.203: 23 genes are commonly lost in *C. gibelio* and *C. auratus*. This is not a strikingly high number, easily just by chance. Is this statistically remarkably high? What are the number of shared gene losses between *C. auratus* and *C. carpio* and *C. gibelio* and *C. carpio*? How have these gene losses been

8confirmed – are there reads bridging the loss?

- l. 224-225 “but the genetic changes in these cell division genes in *Carassius* may be associated with subsequent more dramatic ploidy alternation.” This highly speculative statement is not further substantiated and therefore should be removed.

- l.361: It is more important to know how many generations (rather than years) the breeding experiment ran.

Author Rebuttal to Initial comments

Reviewers' comments:

Reviewer #1 (Remarks to the Author):

I am impressed by the results in the manuscript considering the rarity of hexaploids in vertebrates and very high sequence identities between the subgenomes. Wang et al. revealed that *C. auratus* is amphidiploid (AABB) and *C. gibelio* is amphitriploid (AAABBB) and observed that many gene families related to meiotic cell cycle and oocyte expanded in *C. gibelio*. I believe, although there still exists room to improve, current results will contribute to our understanding of the evolution of polyploid vertebrates. Before publication, some suggestions would be beneficial for them to improve the analysis and reveal more true biology facts.

Response: Thank you very much for your positive comments on this work.

Major concerns:

1. The estimated genome size is much affected by k -mer size in the k -mer analysis. They used 17-mer to estimate the genome size of *C. auratus* and *C. gibelio*. The author should perform the k -mer size simulation to estimate the genome sizes.

Response: Thank you for this suggestion. Accordingly, we further applied five different k -mer sizes (17, 19, 21, 23, 25) to estimate the genome size using a more widely used software program: GenomeScope 2.0 (Ranallo-Benavidez et al., Nat Commun, 2020). The estimated genome size for *C. auratus* ranged from 1.325 to 1.360 Gb (line 109 in the revised manuscript),

9whereas that for *C. gibelio* ranged from 1.411 to 1.424 Gb (line 134 in the revised manuscript). These estimated sizes have very small variances and are slightly smaller than previous estimations. We updated Extended Data Table 5 with the new results.

2. The assembly size of *C. auratus* in this study is smaller than that in other previous reports (Chen et al., Science Advances, 2019; Luo et al., Science Advances, 2020; Chen et al., PNAS, 2020), possibly resulting in the artificial gene losses. Thus, the authors should perform comprehensively comparisons of all available *C. auratus* assemblies, including RNA-seq alignment, genome read alignment, and EST/cDNA alignment, to demonstrate the quality of this new assembled genome of *C. auratus*. Were the predicted genes in other assemblies covered in this new assembly? The alignment ratio of the BAC sequences to the *C. gibelio* genome is a good indicator of genome assembly quality.

Response: Thank you for this suggestion. Accordingly, we performed several comparisons with the available *C. auratus* assemblies.

First, the genomic and RNA-seq reads were mapped to the genome assembly. The mapping rate of genomic reads was 99.85% and the mapping rate of RNA-seq reads ranged from 85.6% to 89.9%. These are comparable to 98.23% and 62.1%–90.4% in the genome version published in BMC Genomics (GCA_013115835.1), which has a genome size of 2.198 Gb (Extended Data Table 8). We were not able to calculate the mapping rate of the other three published genome assemblies (GCF_003368295.1 in Chen et al., Sci Adv, 2019; GCA_014332655.1 in Chen et al., Proc Natl Acad Sci U S A, 2020; GWHAAIA000000000 in Luo et al., Sci Adv, 2020) because their short reads were not available.

Second, we examined the extent of redundancy in different genome assemblies using the software `purge_dups`, including in our genome assembly (1.522 Gb), GCA_014332655.1 (1.740

Gb) using nanopore long reads (Chen et al., Proc Natl Acad Sci U S A, 2020), and GCA_013115835.1 (2.198 Gb) (Wen et al., BMC Genomics, 2020). The results showed that both previously published genomes had more obvious heterozygosity peaks than our assembly (newly included Extended Data Fig. 3). This indicates more redundant assembly in heterozygous regions, which would inflate the assembled size. We then purged redundancy and extracted the haploid sequences from the two previous assemblies using `purge_dups` and obtained similar genome sizes (1.528 Gb for GCA_014332655.1 and 1.407 for GCA_013115835.1) (Extended Data Table 8). Therefore, these results indicate that the larger genome size in previous assemblies was mainly caused by redundancy in haploid sequences. We added these results to lines 90-106 of the revised Supplementary Methods and lines 113–116 of the revised manuscript: “*We noticed that our assembly size was smaller than previously reported C. auratus assemblies, and this was mainly caused by the redundancy in the heterozygous regions (Extended Data Fig. 3 and Extended Data Table 8).*”

Moreover, we also manually checked the lost genes identified by our study in four other published assemblies and confirmed that they did not exist in any of the other assemblies, which confirmed that they are lost genes. This information is provided on lines 296–298 of the revised Supplementary Methods.

Extended Data Fig. 3. Assessment of heterozygous sequences in different versions of *C. auratus* genomes using `purge_dups`. These sequences were largely resulted from assembly redundancy. The horizontal coordinate refers to the read depth, and the vertical coordinate refers to the read occurrence frequency.

3. In Figure 2B, *Cirrhinus molitorella* was clustered with the B subgenomes of common carp, *C. auratus*, and *C. gibelio*. Therefore, they stated that the subgenome B had a closer relationship to the diploid species *Cirrhinus molitorella* and *Poropuntius huangchuchieni* than subgenome A. However, Xu et al. (Nature Communication, 2019, Figure 2a) showed that *Cirrhinus molitorella* was the outgroup of subgenomes A and B. The authors should carefully examine the phylogenetic location of *Cirrhinus molitorella*.

Response: Thank you for pointing out the inconsistency in phylogenetic relationships. However, we noticed that Xu et al. (Xu et al., Nat Commun, 2019, Figure 2a) only used one gene (*rag2*) to construct their phylogenetic tree, and this gene has three copies in the genome of *Sinocyclocheilus grahami*, two copies (one copy is a pseudogene) in some assemblies of *Cyprinus carpio* (German_Mirror_carp_1.0, Hebao_red_carp_1.0, Hunaghe_carp_2.0, GCA_018340385), and one copy in another assembly of *Cyprinus carpio* (GCA_000951615) and all assemblies of *C. auratus* and *C. gibelio*. Therefore, this gene is not appropriate for inferring phylogenetic relationships between these subgenomes, and a phylogenetic tree based on genome-wide data should be more reliable.

4. Besides scanning the subgenome-specific TEs for subgenome division, the author should further validate the accuracy of subgenome division according to Chen et al.'s method (Chen et al., PNAS, 2020), which utilized the alignment read number from close diploid fish, for instance, *Poropuntius huangchuchieni*, to two subgenomes.

Response: Thank you for this suggestion. We added validation of the accuracy of subgenome division by comparison with the goldfish genome (Luo et al., Sci Adv, 2020) and the common carp genome (Li et al., Nat Genet, 2021). Both genomes showed excellent subgenomic correspondence to the *C. gibelio* assembly. We added this new result to the main text (line 122-124) and newly included Extended Data Fig. 7.

Extended Data Fig. 7. Percentage of reciprocal best hit orthologs for each pair of chromosomes of *C. gibelio* to *C. auratus* (this study), goldfish (GWHAIA00000000 from Luo et al., Sci Adv, 2020), and common carp (GCA_018340385.1 from Li et al., Nat Genet, 2021). Red to white indicate high to low similarity values, respectively.

5. In line 184 of Supplementary file, to understand the evolution of the subgenomes A and B of *C. gibelio* and *C. auratus*, the authors aligned the 1:1:1 orthologous genes among *Cirrhinus molitorella*, *Megalobrama amblycephala*, *D. rerio*, *Ctenopharyngodon idellus*, *Poropuntius huangchuchieni*, *Cyprinus carpio*, *C. gibelio*, and *C. auratus*. I suggest that they should select the 1:1:1:1:1:2:2:2 gene families having a single gene in each diploid, 2 common carp genes, 2 *C. auratus* genes, and 2 *C. gibelio* genes. How many such gene families did they find?

14

Response: We apologize for the ambiguous description. Our method is actually consistent with your suggestion because the subgenomes A and B of *Cyprinus carpio*, *C. auratus*, and *C. gibelio* were treated independently. Therefore, 11 gene sets (three sets for subgenome A, three sets for subgenome B, and five sets for other species) were used to identify all reciprocal best hits (as shown in Fig. 2a). Finally, we obtained 3,950 gene pairs, and they were equivalent to 1:1:1:1:1:2:2:2 orthologs in the eight species.

We rephrased the corresponding methods as follows: “*The 11 peptide sequence sets from five genomes (C. molitorella, M. amblycephala, D. rerio, C. idellus, P. huangchuchieni) and six subgenomes (subgenome A of C. gibelio, C. auratus, and Cyprinus carpio; subgenome B of C. gibelio, C. auratus, and Cyprinus carpio) were subjected to Diamond²⁸ to conduct all-to-all blast to identify the potential homologous sequences with an E-value < 1e-5.*” (lines 222–225 in the Supplementary Methods).

6. Figure 2B exhibits signal of gene exchange between two subgenomes with a frequency of 1.86%. Was this frequency significantly different from the null hypothesis? To demonstrate the real signals rather than the phylogenetic tree error, the author should validate these exchanges by comparing the aligned *Poropuntius huangchuchieni* read numbers on the homoeologues from two subgenomes.

Response: Thank you for this suggestion. In the previous version, we performed an incomplete lineage sorting simulation and found that the observed value of 1.86% was much higher than that of the null hypothesis (0.28%). Following your suggestion, we further aligned the reads of *Poropuntius huangchuchieni* to *C. auratus*. However, we noticed that only five genes (marked with yellow color in the following table) exhibited an even read depth distribution between subgenomes A and B. Although this value is higher than the background, it no longer significantly supported this conclusion. Therefore, we removed this part from the revised manuscript.

15

Gene ID		Mean depth per base	
Subgenome A	Subgenome B	Subgenome A	Subgenome B
evm.model.chr13A.786	evm.model.chr13B.826	22.42	31.85
evm.model.chr14A.114	evm.model.chr14B.114	24.84	32.19
evm.model.chr14A.80	evm.model.chr14B.81	2.54	50.46
evm.model.chr16A.339	evm.model.chr16B.362	9.39	43.17
evm.model.chr18A.32	evm.model.chr18B.28	28.91	23.64
evm.model.chr19A.698	evm.model.chr19B.769	8.83	54.66
evm.model.chr1A.272	evm.model.chr1B.915	12.68	48.43
evm.model.chr21A.73	evm.model.chr21B.76	22.58	33.28
evm.model.chr3A.783	evm.model.chr3B.1091	17.7	43.34
evm.model.chr7A.799	evm.model.chr7B.781	14.38	47.06
evm.model.chr9A.102	evm.model.chr9B.160	8.65	47.01
evm.model.chr9A.156	evm.model.chr9B.78	11.87	39.41
evm.model.chr9A.160	evm.model.chr9B.74	25.34	31.24

7. In Figure 2C and D, the author should define the error bars and analyze whether there existed statistical significances.

Response: Thank you for this suggestion. Accordingly, we added information on statistical significance and the meaning of error bars in the figures and figure legends as follows: “*The line in the middle of each boxplot represents the median of the dataset; the upper and lower edges of boxplot indicate the third quartile and the first quartile, respectively; and the line extending from the edge is 1.5 times the interquartile range. Small dots indicate outliers.*” In addition, these figures have been moved to supplementary information (Extended Data Fig. 13) because, as pointed out by reviewer #3, subgenome B genes tend to have lower Ka/Ks ratios and slightly higher expression levels have been previously reported (Li et al., Nat Genet, 2021, Luo et al., Sci Adv, 2020, Chen et al., Sci Adv, 2019, Kon et al., Curr Biol, 2020, Chen et al., Proc Natl Acad Sci U S A, 2020), and this is a very minor point in our paper.

8. In Supplementary file, I did not see the genome sequencing method information on *C. auratus*. The author should rewrite it.

Response: We apologize that this was not clearly described in the previous version. We used the same sequencing strategy for genome sequencing of both species and added this information in the revised Methods section (lines 5, 25, and 30 in the Supplementary Methods).

9. I am curious about which assembly of common carp genome was used in Extended data Table 7. The ENSEMBLE transcript annotations were based on the assembly of GCA_000951615.2, rather than GCA_018340385.1 that they stated in line 181 in Supplementary file.

Response: Thank you for pointing out this issue. We used two different common carp genomes for different analyses. To reconstruct the phylogenetic tree with subgenomes, we used version GCA_018340385.1 for analysis, which has well-distinguished subgenomes A and B. To scan for genes lost in *Carassius*, we used version GCA_000951615.2, which was annotated by ENSEMBL and has abundant gene annotation information. We more clearly indicated which

data we used at corresponding locations in the text to avoid confusion (lines 221 and 286 in the Supplementary Methods).

10. The nomenclatures in Extended data Table 6 were confusing. What are ‘CaiJi_A’, ‘CaiJi_B’, ‘YinJi_A’, and ‘YinJi_B’?

Response: We apologize for the non-standard nomenclature. This table was removed because the results were not significant based on the newly conducted read mapping analysis of *Poropuntius huangchuchieni*, as you suggested in the sixth point.

Reviewer #2 (Remarks to the Author):

This manuscript by Wang et al. involves a comparative study of two *Carassius* fish genomes, one of them (*C. gibelio*) is an unusual unisexually hexaploid fish, being an exciting model to study unisexual polyploids in vertebrates. The subgenome A and B have been assembled at the chromosome-level, though unfortunately the two parental genomes (AA/A and BB/B) prior to triploidization were not resolved, likely due to the recent triploidization event and the limitation of error-prone long reads. Many meiotic cell-cycle and oocyte-specific histone variant genes were found to be specifically expanded in *C. gibelio*, but its implications for unisexual triploid adaptation are rather speculative and obscure (functional experiments are lacking). The data for analysing the gene conversion rate is useful, but I doubt whether the results are solid (see more comments below).

Response: Thank you for your comments and suggestions. As you mentioned, we could not resolve all three haplotypes when using the sequencing mode of PacBio CLR in this study because of the recent autotriploidization event. Nevertheless, in the revision, we conducted

additional experiments and sequence analyses that provided more solid evidence for both unisexual triploid reproduction and gene conversion in this unisexual triploid. The reference haploid genome assembly together with our pedigree re-sequencing data can largely allow inference of linkage relationships among a stretch of SNPs (i.e., phased haplotypes in the conversion regions) and thus identification of candidate conversion regions.

To further reveal the implications of expansion of meiosis-related genes for unisexual reproduction, we carried out the following experiments. 1. DNA content and chromosome spread assays were performed to illustrate that there is suppression of the first meiotic division, which is necessary for unisexually producing eggs in *C. gibelio* (newly included Fig. 4a and 4b). 2. Meiotic synapsis and recombination between homologs were visualized using Sycp1, Sycp3, and Rad51 antibodies (newly included Fig. 6). Based on the visualization, both meiotic synapsis and recombination were observed to be seriously impaired but homologous recombination could occur to some extent in *C. gibelio* oocytes. This provides the cytological basis both for the suppression of the first division and for gene conversion. 3. The expression of these expanded genes was examined and most of the extra copies of *C. gibelio* were expressed during oogenesis, especially during the corresponding stages of normal meiosis prophase I (newly included Extended Data Fig. 19).

We also further checked loss of heterozygosity (LOH) regions and identified possible gene conversion events based on the haplotype information (SNP-converted pattern in the LOH regions) in our gynogenetic *C. gibelio* pedigree. Although some gene conversion blocks were filtered, the conclusions regarding loss of heterozygosity and gene conversion remained unchanged.

We provide more detailed responses to your specific comments below.

Major concerns

1. The extremely large variation of gene conversion rate (almost 200 fold difference among individuals) is concerning – I wonder this is the true nature of this species or it reflects some artifacts in individuals with particular high gene conversion rates.

Response: To test why this gene conversion pattern exists, we examined homologous recombination during the corresponding meiotic stages in *C. gibelio*, which is the main cause of gene conversion. Homologous recombination (indicated by the recombinase Rad51 signals, which is located at the position of DNA double-strand breaks and demonstrates homologous recombination) was suppressed in the early prophase I oocyte of *C. gibelio*, but we observed sporadic homologous recombination in oocytes (Fig. 6a). Importantly, the ratio of the Rad51-positive oocytes was found to have an increasing trend along with the progress of oocyte development (Fig. 6b), and some oocytes (~2.5%) even showed a high level of Rad51-stained foci (over 400) (Fig. 6c). The different levels of homologous recombination occurrence revealed in different oocytes of *C. gibelio* were consistent with the large variations of gene conversion rates observed among different gynogenetic individuals (Extended Data Table 22). Because chiasmata that results from crossover were largely missing in *C. gibelio* oocytes (Fig. 4b), the observed homologous recombination would result in gene conversion through a non-crossover pathway. We added these new results to clearly explain this large variation (lines 406-425 in the revised manuscript): “... *the ratio of the Rad51-positive oocytes was found to have an increasing trend along with the progress of oocyte development (Fig. 6b), in which some oocytes (~2.5%) even showed high levels of Rad51-stained foci (over 400) and synaptonemal bivalents (over 20) (Fig. 6c). The different levels of homologous recombination revealed in different oocytes of C. gibelio are consistent with the large variations of gene conversion rates observed among*

different gynogenetic individuals (Extended Data Table 22), indicating an association between them because non-crossover homologous recombination usually results in gene conversion.”

Fig. 6. Comparative cytological observation and sporadic homologous recombination during oogenic division between *C. gibelio* and *C. auratus*.

a, Chromosomal spreads of different stage primary oocytes of *C. auratus* and *C. gibelio* co-immunostained by anti-Sycp1 (red), anti-Sycp3 (blue), and anti-Rad51 (green) antibodies. Insets show an enlarged view. Arrows mark Rad51-stained foci. Bar=5 μ m. Statistical results are presented as mean \pm SEM in the right panel. n=174 for *C. auratus* and n=203 for *C. gibelio*. Data are presented as mean \pm

21SEM. Bar=5 μ m. **b**, Percentage of Rad51-positive oocytes. **c**, Varying levels of Rad51-stained foci and synaptonemal bivalents in different pachytene-like oocytes of *C. gibelio*. Number of Rad51-stained foci is indicated at the top right. Bar=5 μ m.

Figure 5E shows one example of gene conversion where some changed genotypes (in red) were interpreted as the outcome of gene conversion while some (in green) were not. This is quite difficult to understand - wouldn't be gene conversion affect the entire region? is there another mechanism underlying the changed genotype (green) other than gene conversion? Can authors show the read alignment for this region so that the three karyotypes can be confirmed?

Response: We apologize for the ambiguous expression and description of the previous Fig. 5E. We redrew Figure 5E (Figure 5f in the revised manuscript) and included a new supplementary figure (Extended Data Fig. 24) to more clearly demonstrate this. In these figures, we showed the three haplotype blocks and read alignments for the exemplified regions. In addition, we rephrased this section and the legend of Figure 5f in the revised paper. For example, the following was modified in the main text: *“According to the read coverage of SNP sites between the individuals from the gynogenetic C. gibelio pedigree that did or did not experience gene conversion (Extended Data Fig. 24), the haplotype blocks of gene conversion could be inferred (see the detailed description in Supplementary Methods 14). As shown in Fig. 5f, after gene conversion from haplotype 1 to haplotype 2, 12 out of 35 SNP sites (~1/3) became homozygous, which resulted in LOH; the other sites were still heterozygous, among which 14 SNP sites were clearly converted, and nine SNP sites looked unchanged because their haplotypes 1 and 2 had the same bases before conversion.”* (lines 395–403 in the revised manuscript).

Fig. 5. e, Circos map showing all LOH regions in chromosome 22B of individual G4-9. I, LOH region. II, LOH site number (Log2). III, SNP number (Log2). IV, read depth of SNP sites. The window size in II–IV is 200 kb. Red and blue bricks indicate gene conversion and gene deletion, respectively. The orange and green lines in IV indicate sequencing depths of 50× and 33×, respectively. **f**, One example of the phased haplotype blocks at the gene conversion boundary in **e**. The same region of individual G4-10 is shown as a control without gene conversion. This gene conversion occurred from haplotype 1 (donor) to haplotype 2 (recipient). Red bases, homozygous converted sites which resulted in LOH in this region (LOH sites). Green bases, heterozygous converted sites where the donor allele was minor allele (non-LOH site with changed genotype after conversion). Blue bases, heterozygous sites where the donor allele had the same base as the recipient allele (non-LOH site with unchanged genotype after conversion).

Black bases with grey shadowed, heterozygous sites outside the gene conversion region. Read mapping depths of SNP sites in individual G4-9 are shown on the right panel.

Extended Data Fig. 24. Read coverage for a block in the gene conversion region shown in Figure 5e. The same positions of G4-9, G4-10, and G4-11 individuals are visualized using IGV software. In each sample, the top panel shows the read coverage based on the aligned reads presented in the bottom panel. Four bases (ATCG) at SNP sites are presented with different colors. *, LOH site. Δ, Non-LOH site with changed genotype after conversion. #, Non-LOH site with unchanged genotype after conversion.

For some LOH regions, deletions were to blame where 2/3 coverage is shown; and for regions with normal coverage it was assumed gene conversion caused LOH. This is quite a crude identification of gene conversion, though the authors explained in the methods that the coverage patterns were stable. This does not seem to be the case as one can spot a lot of coverage variation in figure 5D. It could be possible that the regions defined as gene conversion actually lost one allele (deletions) but somehow show higher coverage than 2/3 due to for example differential GC content? More evidence is needed to define gene conversion, for install, showing three PHASED haplotype blocks at the gene conversion boundary containing non-converted and converted sites.

Response: We apologize for not clearly describing the method of LOH analysis and the shown phased haplotype blocks. Thank you for the suggestions. According to these comments, we further analyzed gene conversion regions by taking more criteria into account and carefully rephrased the method, as explained in details below.

First, we used the stricter criteria to process SNPs and identified LOH sites in the SNP set identified in the 11 offsprings from the gynogenetic pedigree: (1) Filter the non-triploid chromosomes (Chr1B, 6A and 22A of Cg-F1 in Extended Data Fig.18). (2) Trimorphic SNP sites that have three different bases account for only 0.26% (34,916) of the total SNP sites, and the rest are dimorphic, of which the depth of one type of base is usually twice of the other type

25(referred to as the minor allele), indicating most SNPs were singletons (minor allele) in the first parental mother of the gynogenetic pedigree. Therefore, we used these dimorphic SNPs for following analyses. (3) SNPs with a minimum average of 20× coverage and a maximum coverage of 80× were maintained. (4) Sites directly adjacent to small insertion-deletion mutations were filtered to avoid false-positive inferences created by misalignment. (5) For each SNP site of one individual, the coverage depth of minor allele $\geq 5\times$ was considered as heterozygous site of the line, and $\leq 1\times$ was considered as homozygous site. (6) If SNPs in any individual with a coverage depth of minor allele $> 1\times$ and $< 5\times$ were considered as ambiguous sites and filtered from the SNP set. (7) LOH sites were only called when they were heterozygous in some individual but became unambiguous homozygous genotype in one or more individual(s), and 64,246 LOH sites were obtained from a total of 9,780,732 SNP sites. (8) 97 of 101 randomly selected LOH sites were verified using PCR and Sanger sequencing that determine SNP genotype.

Unlike in diploid where LOH (homozygous) SNP sites are continuous, every LOH block may contain both LOH SNPs sites and still heterozygous SNPs (referred to as non-LOH) sites in triploid (Extended Data Fig. 22a). Given this discontinuity, we next restricted our search to contiguous tracts of LOH sites. We considered the first LOH site found on a tract to be part of a possible LOH region and iteratively extended the region if a next LOH site was found nearby to the previous LOH site. The tract length of each LOH region was calculated from the interval midpoint between the first LOH site and upstream non-LOH SNP site to interval midpoint between the last LOH site and downstream non-LOH SNP site. In addition, we filtered the LOH regions with only one LOH site presenting in a single individual.

After identifying LOH sites and regions, we then moved to filter deletion regions. (1) We plotted the distribution of the average normalized read depth for SNP sites in each LOH region, and it showed two peaks around 49× (triploid) and 31× (diploid) (Extended Data Fig. 22b). (2)

We plotted the distribution for the average frequency of minor allele in each LOH region, and the results also showed two peaks around $0.33\times$ (triploid) and $0.42\times$ (diploid) (Extended Data Fig. 22c). (3) LOH regions with an average read depth $>40\times$ and average frequency of minor allele <0.37 at the same time were considered as gene conversion regions (triploid) (p -value < 0.05 in one or more lines, binomial test). Eventually, we obtained the candidate gene conversion regions that contained 61,014 LOH sites (95.0% of total LOH sites) in the 11 individuals of the gynogenetic pedigree.

Finally, we analyzed the identified gene conversion regions based on the unique SNP-converted pattern in triploids. As shown in Extended Data Fig. 22a, after a gene deletion, there are two SNP sites: $1/3$ of SNP sites show LOH and the rest remain heterozygous with similar read depth for each allele (like in diploids); however, after a conversion, there are three types of SNP sites: homozygous converted sites which result in LOH in this region, heterozygous converted sites where the donor allele is minor allele before conversion, and converted sites that look unchanged where the donor allele has same base as the recipient allele before conversion but heterozygous with the minor allele, and their ratios should show a pattern of $1/3:1/3:1/3$ if a conversion region is long enough. Accordingly, we calculated the ratios of the three types of SNP sites in the candidate gene conversion regions; exactly, each type approximately occupied $1/3$ of total SNP sites in a conversion region (Extended Data Fig. 22d). Moreover, we phased the blocks by comparing homologous SNP sites between individuals that did or did not experience gene conversion, where SNP genotyping was determined by the read coverages of its two base statuses (Extended Data Fig. 24). Since gene conversion is a unidirectional DNA modification from one haplotype to another, the donor and recipient alleles at each SNP site can be inferred respectively in the gynogenetic pedigree, and thereby the three haplotypes will be phased. As expected, the phasing blocks (Fig. 5f and Extended Data Fig. 23) present a well-defined SNP pattern for gene conversion in triploid (Extended Data Fig. 22a).

Based on these new analyses, we updated Extended Data Table 12 and 13, and carefully rephrased the Methods section (lines 348-417 in the revised Supplementary Methods).

Extended Data Fig. 22. Analysis of the LOH regions detected in the genogenetic pedigree of *C. gibelio*. **a**, Schematic diagram of a LOH region in diploid (left panel) or triploid (right panel). Allele 2 was deleted in a gene deletion event, whereas a gene conversion occurred from Allele 1 to Allele 2. Theoretical ratio of different types of SNPs in a LOH region is shown in the right side of each panel. **b**, Distribution of average depths of SNP sites in LOH regions. **c**, Distribution of average minor allele frequencies in LOH regions. **d**, Ratio distributions of the three types of SNP sites in gene conversion blocks in a 30-SNP sliding window and 1-SNP step. The line in the middle of each boxplot represents the median of the dataset; the upper and lower edges of the

boxplot indicate the third quartile and first quartile, respectively; and the line extending from the edge is 1.5 times the interquartile range.

Furthermore, I wonder how elevated gene conversion rate is necessarily adaptive - this is not well explained in the manuscript. It is also not clear whether elevated gene conversion rate is a derived feature for *C. gibelio*. It would be ideal to compare the gene conversion rate between *C. auratus* and *C. gibelio* to confirm that *C. auratus* does not have a high gene conversion rate.

Response: Thank you for this suggestion. We agree that it is inappropriate to use “adaptive” to describe the gene conversion phenomenon in *C. gibelio* because it is simply a mechanism of purging severely deleterious mutations in oogenesis of this unisexual species. Therefore, this was revised as follows: “*Therefore, high gene conversion might render C. gibelio capable of purging deleterious mutations and may be associated with the alternative ameiotic oogenic mechanism.*” (lines 403-405 in the revised manuscript).

Regarding the comparison of gene conversion rates between *C. auratus* and *C. gibelio*, because we do not have a good pedigree for the sexual *C. auratus*, which shows a normal rate of meiotic homologous recombination similar to other sexual organisms (Fig. 6a), it is difficult to conduct such comparisons. Our results indicate that the observed recombination in *C. gibelio* provides a possible pathway for gene conversion, even in the absence of normal meiosis (Fig. 6a and Fig. 4c). Therefore, the gene conversion rate of *C. gibelio* is high relative to other unisexual species, and we rephrased this sentence as: “*The gene conversion rate of C. gibelio is two orders of magnitude higher than that of the reported unisexual species and nearly reaches the reported range of some sexual species.*” (lines 381-383 in the revised manuscript).

2. The authors used gene phylogeny to test whether there are gene exchanges between A and B subgenomes. The fact that only 13 genes show the alternative topology gave me a pause - is the topology of all 13 gene trees well supported? Apparently there could be other explanations including ancestral gene flow or incomplete lineage sorting which were not assessed.

Response: Thank you for pointing out this problem, which prompted us to carefully examine the homeologous gene exchanges.

This analysis was actually started with 3,950 orthologous genes from 11 protein datasets (five genomes and six subgenomes). These genes were then used to independently construct maximum likelihood trees. Among these trees, 698 of them had a bootstrap value larger than 70% in every node. Additionally, the 13 genes are among the 698 genes (1.86%). We also constructed maximum parsimony trees for these genes and the results were consistent. In addition, if we did not consider the bootstrap value of the trees, 95 trees (2.64%) had a topology that supported homeologous gene exchanges. However, when we mapped the reads of *Poropuntius huangchuchieni* to *C. auratus*, we noticed that only five genes exhibited an even read distribution between subgenomes A and B, which challenges the conclusion of gene exchange between subgenomes A and B.

Furthermore, we entirely agree with you that there could be other explanations for such a pattern, including ancestral gene flow and incomplete lineage sorting. In the previous version, we performed an incomplete lineage sorting simulation, and found that the observed value of 1.86% was much higher than that of the null hypothesis (0.28%). Therefore, to be cautious, we removed the related descriptions about gene exchange between subgenomes A and B in the revised manuscript.

Minor concerns

L52, also popular in some other countries.

Response: Thank you for point this out. We deleted the phrase “in China.”

L95, Hi-C data was not shown in the Extended Table 1. Also, I believe for the Illumina table the labels “Illumina insert size” and “Read length” were swapped.

Response: Thank you for pointing out the problems in Extended Data Table 1. The information in the original Extended Table 1 were instead provided in Extended Data Tables from 1 to 4 in the revised document. We added Hi-C information in Extended Data Table 2 and the amended column headers of Extended Data Table 3 in the revised manuscript.

L98, awkward sentence. In general, English language needs to be polished throughout the manuscript.

Response: We apologize for all of the awkward sentences in the manuscript. We have had the English throughout the manuscript edited by an English editing service. Additionally, this specific sentence was rephrased as follows: “*In C. auratus, 58% of heterozygous k-mer pairs (with only one nucleotide difference and presented as x and x') are bivalent (xx') and 33% of heterozygous k-mer pairs are tetravalent (xxx'x' and xxx'x').*” (lines 104–106 in the revised manuscript).

L105, unclear what is a hierarchical strategy - The extended fig.2 only show a workflow for long-read genome assembly and annotation.

Response: We apologize for this misleading phrase. We deleted the phrase “using a hierarchical strategy.”

L117, Fig. 1B shows five TEs, where are the nine TEs?

Response: We rechecked the original data and determined there were five subgenome-specific TEs used in this study.

L119, perhaps use different colors for A and B subgenome, making it clear that A and B assemblies have 1:1 correspondence.

Response: Thank you for this suggestion. Accordingly, we used different colors for subgenomes A and B in Extended Data Fig. 6.

L122, unclear how the level of redundancy was measured.

Response: We apologize for the ambiguity. We added the assessment of the level of redundancy of different genome versions to the supplementary information (newly included Extended Data Fig. 3 and Table 8). This analysis was performed using the software `purge_dups` (https://github.com/dfguan/purge_dups), which employs the read depth information to identify haplotigs and overlaps in the genome. The results showed that both previously published genomes had more obvious heterozygosity peaks than our assembly (Extended Data Fig. 3), indicating more redundant assembly in heterozygous regions, which would inflate the assembled size. We then purged redundancy and extracted the haploid sequences using `purge_dups` from the two previous assemblies and obtained similar genome sizes (1.528 Gb for GCA_014332655.1 and 1.407 for GCA_013115835.1) (Extended Data Table 8). Therefore, these results indicate that the larger genome size in previous assemblies is mainly caused by the redundancy of haploid sequences, and we added this result to the revised main text: “*We noticed that our assembly size was smaller than previously reported C. auratus assemblies, and this was mainly caused by the*

redundancy in the heterozygous regions (Extended Data Fig. 3 and Extended Data Table 8).”
(lines 113–116 in the revised manuscript).

Extended Data Fig. 3. Assessment of heterozygous sequences in different versions of the *C. auratus* genome using purge_dups. These sequences largely resulted from redundant

assemblies. The horizontal coordinate refers to the read depth, and the vertical coordinate refers to the frequency of read occurrence.

L186, are we sure this is the only reason for the disparity? Otherwise one needs to add “partially”.

Response: Thank you for this suggestion. Accordingly, we added “partially” here (line 197 in the revised manuscript).

L192, more details are needed in the legend.

Response: We added more details in the legend of Extended Data Fig. 12 as follows: *“Extended Data Fig. 12. Correlation between divergence time and Ks from different studies. Each point in the figure represents a differentiated node, where the horizontal coordinate refers to the estimated divergence time in each study (distinguished by color) and the vertical coordinate refers to the Ks value between two branches of this node. When points from different studies had the same Ks value, we put them in the same differentiated node. We applied Pearson’s correlation coefficient analysis to calculate the R² value and p-value for divergence times (from different studies) and Ks distribution.”*

L202-203, why a lower Ka/Ks rate is consistent with dominant B? An explanation is needed. Similarly, please help readers understand why the dominant subgenome should increase gene expression in later stages.

Response: A lower *Ka/Ks* usually indicates a higher level of purifying selection. As this phenomenon is well-known, we moved the figure 2C into the supplementary information (newly included Extended Data Fig. 13a) and added an explanation to the corresponding figure legend:

“A lower K_a/K_s indicates a higher level of purifying selection in subgenome B, which indicates that subgenome B is more likely to be the dominant genome with more important function.”

For the same reason, we also moved the figure 2D into the supplementary information (newly included Extended Data Fig. 13b) and added the following explanation to the corresponding figure legend: *“...indicating that subgenome B should play a more important functional role than subgenome A; thus, the B genes may have higher expression levels.”*

L207, it's strange why Fig. 4C is mentioned here.

Response: We apologize for this mistake. We removed “Fig. 4C” from here.

L225, this is confusing. The authors identified shared gene loss by *C. auratus* and *C. gibelio*, and came to the suggestion that the gene loss events are associated with ploidy alternation. But the fact is only *C. gibelio* has experienced one additional polyploidization event.

Response: We agree with you that this statement was incorrect and very speculative. Therefore, we deleted this statement.

L243, should be three alleles.

Response: We changed “haplotypes” to “alleles” (lines 235 in the revised manuscript).

L254-255, confusing, did that authors try to say different populations have diverged.

Response: We apologize for the confusing expression. Here, we were attempting to say that the investigated *C. gibelio* might evolve independently from a common origination. Therefore, to improve clarity, we rephrased this sentence as: *“these results suggest that the investigated C. gibelio might have a common origin.”*(line 246 in the revised manuscript).

35L255, looking at the methods, here the conserved non-coding elements should rather be species-specific elements.

Response: Thank you for this advice. Accordingly, we defined these non-coding elements as “*C. gibelio*-specific element” (lines 279-280 in the revised manuscript).

L256, Fig. 3C seems to show only one example?

Response: Yes, Fig. 3C shows one *C. gibelio*-specific element in the *cenpw* gene. The other *C. gibelio*-specific elements have similar read depth distributions.

L257, this is misleading - the elements could have an ancestral origin, but only remain conserved in *C. gibelio*.

Response: Thank you for this suggestion. We rephrased the sentence as follows: “*Moreover, 4,400 non-coding elements were found to be shared by all C. gibelio individuals (Fig. 3c) but were absent in C. auratus, Cyprinus carpio, and S. graham, indicating that they are newly evolved elements in C. gibelio.*” (lines 247–249 in the revised manuscript).

L261, in extended figure 13, the dating of T3 is 0.962. Also, the time points were already calibrated with “0.86 – 1.051” (Method L256), making the time estimation not independent. So I wouldn’t trust the claim at line 264-265.

Response: We apologize for these ambiguous descriptions. We independently estimated the divergence times twice. First, we estimated T1 (the divergence time of the ancestors of subgenomes A and B), T2 (time of the allotetraploidy event), and T3 (divergence time between *C. gibelio* and *C. auratus*) using fossil records and 3,950 orthologous genes. Second, we

estimated T4 (divergence time of different *C. gibelio* lineages) using resequencing data and two calibration points (T3 and the divergence time between *Cyprinus carpio*–*C. auratus*) for time calibration.

To make these descriptions clearer, we revised the related sentences as follows: “*It could be inferred that: 1) the progenitor-like genomes (ancestors of subgenomes A and B) diverged around 19.50 Mya (T1) (Fig. 2a); 2) the allotetraploidy event (the hybridization of subgenomes A and B) occurred between 10.17 to 12.87 Mya (T2) based on the divergence times of common carp (Cyprinus carpio) versus Carassius, and versus P. huangchuchieni; and 3) the divergence time of C. gibelio and C. auratus occurred around 0.96 Mya (T3) (Fig. 2a).*” (lines 190–196 in the revised manuscript), and “*The divergence time of the three C. gibelio strains was estimated to be approximately 0.82 Mya (T4) using four degenerated sites (Extended Data Fig. 16).*” (lines 251–253 in the revised manuscript).

L281, where is the data for the lower fitness?

Response: We deleted this statement because there is no solid evidence to support this, as you pointed out.

L297, so what is the case in *C. gibelio*? Please briefly describe the meiosis process in *C. gibelio*.

Response: As described above, we conducted more experiments and added the results in the revised manuscript. Briefly, to understand the process of meiosis in *C. gibelio*, we measured the DNA contents and counted the bivalent/univalent numbers during oogenesis (Fig. 4a and 4b). In addition, suppressed homologous pairing and recombination were also observed in *C. gibelio* (Fig. 6). These new data indicate that meiosis I has been suppressed during oogenesis in *C. gibelio* (Fig. 4c): “*To understand what happens in C. gibelio oogenesis, we first measured the*

DNA content during oocyte development. The DNA content of C. gibelio oocytes at early prophase was approximately 1.67 times that of corresponding C. auratus oocytes (Fig. 4a), whereas the DNA content of C. gibelio mature oocytes was approximately 3 times that of C. auratus mature oocytes (Fig. 4a); this indicates formation of unreduced eggs in C. gibelio compared with formation of reduced eggs in C. auratus. Additionally, compared with 50 bivalents in C. auratus, an average of more than 130 univalents was counted in germinal vesicle breakdown oocytes of C. gibelio (Fig. 4b); this means chiasmata, physically associated structures between homologous chromosomes, were largely missing. Therefore, meiosis I was suppressed during oogenesis in C. gibelio (Fig. 4c).” (lines 288–299 in the revised manuscript).

Fig. 4. Alteration of oogenesis and expansion of meiosis-related genes in *C. gibelio*.

a, DNA content of early primary oocytes ($n=318$ for *C. auratus* and $n=434$ for *C. gibelio*, bar=50 μm) and mature oocytes ($n=56$ for *C. auratus* and $n=54$ for *C. gibelio*, bar=5 μm). Data are presented as mean

39± SEM. **b**, Bivalent and univalent numbers of oocytes at germinal vesicle breakdown (GVBD) (n=19 for *C. auratus* and n=39 for *C. gibelio*, bar=50 µm). Data are presented as mean ± SEM. **c**, Schematic of *C. auratus* and *C. gibelio* oocyte maturation division. Normal meiosis produces oocytes with a halved genome in *C. auratus*, whereas the first meiotic division is suppressed in *C. gibelio*. PB, polar body. **d**, Linkage-specific gene expansion shared by the three investigated *C. gibelio* strains. The numbers below are the gene copy numbers in the assemblies of *Cyprinus carpio* (GCA_000951615.2), *C. auratus*, and *C. gibelio* (this study). The heatmap shows the relative depth of each gene in the resequencing samples. **e**, Expanded histone *h2af1a* gene variants and their expression in *C. gibelio*. Left, maximum likelihood gene tree of *h2af1a*. Right, heatmap of *h2af1a* expression levels in different tissues, oocytes, and embryos. HP, hypothalamus and pituitary; PO, pre-vitellogenic oocytes; VO, vitellogenic oocytes; dpf, days post-fertilization. **f**, Nine lineage-specific expanded genes of *C. gibelio* in the regular oocyte meiosis pathway. MT, microtubule; MTOC, microtubule-organizing center; 1st PB, the first polar body; NE, nuclear envelope; MI, meiosis I; MII, meiosis II.

L301-302, that number is average number across individuals? Please confirm.

Response: The numbers in Fig. 4c in the revised manuscript are the gene copy numbers in the genome assembly of strain F, not an average number across individuals. We specifically added this information to the figure legend: “*The numbers below are the gene copy numbers in the assemblies of Cyprinus carpio (GCA_000951615.2), C. auratus, and C. gibelio (this study).*” (lines 340–341 in the revised manuscript).

L310, please specify what special meiosis.

Response: As described above in response to your question about L297, we added a description of the oogenesis process in *C. gibelio* (Fig. 4b; lines 288–299 in the revised manuscript).

L338, The extended table 10 shows 3 copies instead of 4, please confirm.

Response: We apologize for this mistake. We carefully checked the data and corrected the mistakes in Extended Data Table 10: *C. auratus* has three *h2af1a1* copies and *C. gibelio* has seven *incenp* copies.

L340, is it inferred that *C. gibelio* has altered cell division of meiosis I, or this is already observed?

Response: As described above in response to your question about L297, we conducted several new experiments and revealed a unique mode of oogenesis in *C. gibelio*, as shown in Fig. 4a and 4b (lines 288–299 in the revised manuscript).

L355, because the haploid genome was assembled for *C. gibelio*, I wonder whether the loss-of-function mutations occurred in all three alleles? And if the authors wanted to investigate the accumulations of deleterious mutations, they need also to look at the slightly deleterious mutations. Besides the density of mutations, the authors should also test for an excess of non-synonymous substitutions.

Response: Yes, these loss-of-function mutations occurred in all three alleles for each individual; this was confirmed by checking the sequencing reads. Following your suggestion, we also analyzed the accumulation of two other types of mutations, non-synonymous and synonymous substitutions. The results showed that all three types of mutations exhibited similar distribution pattern (Extended Data Fig. 21), and thus we added the following sentence: “*We then investigated the number of loss-of-function mutations, non-synonymous substitutions, and synonymous substitutions in the two Carassius species using Cyprinus carpio as a reference.*”

41Interestingly, there was no significant difference between the two species and all the three types of mutations exhibited similar distribution patterns (Fig. 5b and Extended Data Fig. 21)” (lines 358–362 in the revised manuscript).

Extended Data Fig. 21. Average numbers of different types of mutations in different individuals. Left, the ID of each individual. The horizontal coordinate refers to the average number of mutations (fixed in each individual) per individual within each megabase (MB). The line in the middle of each boxplot represent the median of the dataset; the upper and lower edges of boxplot indicate the third quartile and first quartile, respectively; and the line extending from the edge is 1.5 times the interquartile range. Small dots indicate outliers.

L366, I assumed the parental fish was also sequenced? But the Extend Table 11 only shows info about the 11 offspring.

Response: Unfortunately, the parental fish of these individuals were too old and had died before

we sampled it. Nevertheless, resequencing analyses were used to infer the pedigree relationship among the gynogenetic offspring.

L416, please remind the readers what are the significant genomic changes?

Response: As suggested, we highlighted the genomic changes as follows: “...including intensive expansion of many meiosis-related genes and a high rate of gene conversion.” (lines 475–476 in the revised manuscript).

Methods L71, isn't Illumina reads were sequenced? Why BGI short reads?

Response: Thank you for pointing out this typo. We corrected this to Illumina reads on line 85 in the Supplementary Methods.

Methods L94, I believe StringTie is an assembler, not a read mapper. Please correct.

Response: Thank you for pointing out this error. We corrected this as follows: “Additionally, Illumina RNA-seq data of all *C. gibelio* and *C. auratus* embryos were mapped to genome of *C. gibelio* and *C. auratus*, respectively, using HISAT2 (v2.1.0) and were assembled to transcripts using StringTie (v2.1.4) software.” (lines 123–126 in the Supplementary Methods).

Reviewer #3 (Remarks to the Author):

The past few years have seen a lot of attention to cyprinid genomes and the analysis of evolutionary genetic patterns following the carp genome duplication, an allopolyploidization event in the ancestor of common carp, goldfish and others (see Chen et al. 2019, Sci Adv; Kon et al. 2020, Curr Biol; Chen et al. 2020, PNAS; Li et al. 2021, Nat Gen). The current study by

43Wang et al. presents yet another high-quality genome assembly for the allotetraploid goldfish, confirming some of the subgenome dominance patterns observed previously. More importantly, the present study focuses on the investigation of the Prussian carp (*C. gibelio*), an allo-auto-hexaploid and asexually reproducing species with yet another amplification of the genome from the ancestral carp situation. The study is well written and touches upon some very interesting questions and first insights regarding the long-term sustainability of this hexaploid lineage despite its unisexual reproductive mode. This asexual reproductive mode is reminiscent of the situation in the gynogenetic Amazon molly, albeit the latter not being polyploid.

While the current work is certainly an important first step for our understanding of the genetic regulation of the unique genomic architecture and reproductive mode of this species, I find that the manuscript remains rather descriptive, and unfortunately does not provide some more mechanistic explanations for the evolutionary survival of such unique genomic constitution.

Response: Thank you very much for your positive comments on this work.

Major points:

1) While *C. gibelio* is hexaploid (AAABBB), only the A and B subgenomes from the pan-carp allopolyploidization could be separately assembled (but maybe I am misunderstanding). It needs to be discussed why additional phasing of the different A and B chromosomes is not possible. The study introduces an XX' nomenclature which is confusing as it is unclear whether it refers to alleles (?) or ohnologs (?) or both (?). While in goldfish, for many/most genes or loci 4 different versions would be expected (2 alleles per A and B subgenome), in *C. gibelio* one should be able to find 6 (3 alleles of A and B subgenome). Has this not been observed?

Response: We apologize for not clearly presenting the related results. To explain why additional haplotype phasing is not feasible at the moment, we added the following: "...because of the

recent autotriploidy event and the limitation of error-prone long reads...” (lines 473–474 in the revised manuscript).

Based on your comment, we also revised the following sentence: “In *C. auratus*, 58% of heterozygous *k*-mer pairs (with only one nucleotide difference and presented as *x* and *x'*) are bivalent (*xx'*) and 33% of heterozygous *k*-mer pairs are tetravalent (*xxx'x'* and *xxxx'*) (Extended Data Fig. 1).” (lines 104–106 in the revised manuscript).

Additionally, details regarding heterozygous *k*-mer pair analysis by Smudgeplots was added to Supplementary Methods 2.2.

Extended Data Fig. 1. Smudgeplots showing the haplotype structure of *C. auratus*.

x and *x'* represent a pair of heterozygous *k*-mers with only one SNP difference. The darkness of each smudge is determined by the number of heterozygous *k*-mer pairs that fall within it. The

45percentage of each genotype is presented at the right. xx' indicates sequences that are consistent with the pattern of diploid species. Therefore, the homologous sequences from subgenomes A and B are different. In contrast, xxx'x' and xxxx' indicate that these sequences are consistent with the pattern of tetraploid species, in which four alleles are observed. Given the proportion of different types of *k*-mers, it was concluded that this is an allotetraploid species, and most of the regions from the two subgenomes are largely diverged.

We apologize for the inaccurate statement. As you mentioned, most genes were observed having three alleles in *C. gibelio*. Furthermore, for some phased regions by BAC sequencing, we were able to discriminate six copies. We showed two phased blocks in this study (Extended Data Fig. 10), and *foxl2* and *viperin* were described in our previous papers (Mou et al., *Front Immunol*, 2021; Gan et al., *Mol Biol Evol*, 2021); most of them have six different copies. Accordingly, we revised this information in the manuscript as follows: “*Moreover, to provide more evidence at the genomic block and gene levels, we performed an allelic analysis by BAC phasing and PCR verification. We found that most of the phased blocks indeed had three homologous alleles for both A and B subgenomes in C. gibelio (Extended Data Fig. 10), and the functionally investigated foxl2 and viperin were also demonstrated to contain three highly identical alleles*^{27,28}.” (lines 156–160 in the revised manuscript).

2) It is interesting that apparently a substantial fraction of genes implicated in meiosis and oocyte formation have been amplified specifically in *C. gibelio*. However, besides the provided discussion of the role of some of these individual genes, one would like to know how these extra copies have been generated (tandem duplication, en bloc duplication, retrotransposition, etc.),

what their expression profiles look like (beyond the case of the histone gene), how many of these extra copies may be functional, etc.

Response: Thank you very much for this comment. We investigated the expression profiles and genomic status of these extra copies. Significantly, all of the expanded meiosis-related genes can be assigned to the common meiosis pathway of oocyte development, including two cell cycle-related genes, *fbxo5* and *ccna2*; three spindle organization genes, *rhoA*, *incenp* and *nusap1*; and three nuclear envelope-related genes, *lem4*, *lap2* and *bmb* (Fig. 4f). Additionally, most of them (22 of 26 extra copies of the eight expanded genes) were expressed in the ovary, pre-vitellogenic oocytes, or vitellogenic oocytes (RPKM > 1) (newly included Extended Data Fig. 19), which indicates that they have roles in oocyte development of *C. gibelio*. In addition, we noticed that most of the newly expanded copies are distributed far from the parental copies, with only three exceptions (newly included Extended Data Fig. 20a and 20b). In particular, all of the extra copies of *h2af1al* (11 extra copies) and *faap24* (two extra copies) were adjacent to a *C. gibelio*-specific repeat unit (newly included Extended Data Fig. 20c), indicating that the expansions of these genes might be mediated by repetitive sequences. The above data suggest that an alternative oogenic pathway to produce chromosome number-unreduced eggs may be related to intensive **expansion of meiosis-related genes in *C. gibelio***. Accordingly, we added the following new results and descriptions: “*Most of them (22 of the 26 extra copies of the eight expanded genes) were expressed in the ovary, pre-vitellogenic oocytes, or vitellogenic oocytes (RPKM > 1) (Extended Data Fig. 19), indicating that they have roles in oocyte development of C. gibelio. We also noticed that most of the new expanded copies were distributed far from the parental copies, with only three exceptions (Extended Data Fig. 20a and 20b). In particular, all of the extra copies of h2af1al (11 extra copies) and faap24 (two extra copies) were adjacent to a C. gibelio-specific repeat unit (Extended Data Fig. 20c), indicating that the expansions of these genes might have been mediated by repetitive sequences. The above data suggest that an alternative*

oogenic pathway to produce chromosome number-unreduced eggs is likely related to intensive expansion of meiosis-related genes in C. gibelio.” (lines 318-329 in the revised manuscript).

Extended Data Fig. 19. Expression pattern of the eight expanded gene families related to oocyte formation in *C. gibelio*. The red lines represent the old genes of the two *Carassius* species. The green lines represent the newly evolved copies in *C. gibelio*. HP, hypothalamus and pituitary; PO, pre-vitellogenic oocytes; VO, vitellogenic oocytes; dpf, days post-fertilization.

Extended Data Fig. 20. Lineage-specific repeats near the expanded *h2af1al* and *faap24* genes in *C. gibelio*. **a**, Adjacent specific repeats of expanded *h2af1al* genes. Left panel, *h2af1al* gene tree. Right panel, the location of lineage-specific repeats relative to the *h2af1al* genes. The rectangle and triangle represent repeats and genes, respectively, and the direction of triangles represents the direction of the gene. **b**, The same analyses of adjacent specific repeats of expanded *faap24* genes. **c**, Copy number and distribution of JC69 distance for R_204_1_1120, a lineage-specific repeat near both *h2af1al* and *faap24* in *C. gibelio*. TD: tandem duplications.

3) Given the unique reproductive mode of *C. gibelio*, I would have expected an analysis also of the genomic region that corresponds to the sex determination region of *C. auratus* that has been discussed in previous studies. The occurrence of male *C. gibelio* is mentioned, and the same

49

group has just recently provided an in-depth investigation of supernumerary chromosomes in male *C. gibelio*, but I would have expected to see a male genome assembly in the current study for comparison as well.

Response: Thank you for highlighting this issue. A previous study (Wen et al., BMC Genomics, 2020) and the results of our group (unpublished data) identified chromosome B22 as the sex chromosome of *C. auratus*, and a potential master sex gene, *amh*, is located in the sex determination region. We performed comparative analysis between B22 chromosomes of *C. auratus* and *C. gibelio* and found that they have conserved gene synteny around the *amh* gene. Unfortunately we unable to detect significant mutations in *amh* between *C. auratus* and *C. gibelio*, as shown in following figure (newly included Extended Data Fig. 25).

Extended Data Fig. 25 Alignments of the sex determination region and candidate sex-determining gene of *C. auratus*. **a**, Synteny alignment of the sex chromosomes between different versions of *C. auratus* genomes and the *C. gibelio* genome. The red bars indicate the

51sex chromosome regions. **b**, Protein sequence alignment of Amh. The results indicate it is largely conserved between *C. auratus* and *C. gibelio*.

It is known that sexual *C. auratus* has an XY sex determination system (Wen et al., BMC Genomics, 2020), whereas unisexual *C. gibelio* occasionally produce males that were observed to have supernumerary microchromosomes (Ding et al., PLoS Genet, 2021). Intrigued by your comments, we sequenced a male individual of *C. gibelio* (F strain) using Illumina sequencing technology. In total, 333 Gb reads were mapped to the reference genome of *C. gibelio* with a mapping rate of 99.23%. The unmapped reads were then used to assemble possible male-specific regions using platanus v1.2.4. Finally, we obtained 33 Kb sequences, with an N50 of 16.6 Kb. Only one gene (*tufm*) was found in the assembled sequence. Unfortunately, we noticed that this gene is highly similar to the copy in *Streptococcus*, which indicates this is most likely from contamination during sampling and sequencing. Moreover, we examined the read mapping to the *amh* gene region, and the *amh* genes from males and females were identical.

Therefore, the current available data are not sufficient to reveal how the unisexual *C. gibelio* can produce males. We added these new data (Extended Data Fig. 25) and discussion (lines 503-509 in the revised manuscript) about this issue, and pointed out that a good *de novo* assembly of a male *C. gibelio* genome with the male-specific supernumerary sequences deciphered is needed to address this issue: “*However, after initial attempts, we were unfortunately not able to detect significant mutations around the potential master sex gene amh between C. auratus and C. gibelio (Extended Data Fig. 25). Additionally, we failed to obtain any informative male-specific supernumerary sequences from one male individual of C. gibelio (Supplementary Methods 2.9). A high-quality male genome assembly for C. gibelio will be required to uncover the mechanisms underlying male determination and gene conversion in the future.*”

4) The section “Allotetraploidy and genetic variations in the common ancestor of *Carassius*” mostly confirms findings of the many other recent carp and goldfish genome articles and should thus be shortened and partially moved to the supplements.

Response: Thank you for this suggestion. According to your suggestion, in the revised version, we only kept the contents associated with the updated divergence time and shared lost genes. The sentences about the dominance of subgenome B, which has been previously reported, were moved to supplementary materials. The sentences about gene exchange between subgenomes A and B were deleted because they were not well supported by our new analysis, as shown in the response to the sixth comment of reviewer #1.

Other specific points:

- Check if the use of references in the abstract is ok.

Response: Abiding by the format of *Nature Ecology and Evolution*, we deleted the references in the abstract.

- Amphidi/triploid is not a common term, please define.

Response: Thank you for your suggestion. We added definitions on line 107: “*amphidiploid (a synonym of allotetraploid) characteristics*²³” and line 164-167: “*Following the nomenclature of amphidiploid, we called *C. gibelio* an amphitriploid (AAABBB) with two triploid sets of chromosomes, each of which was derived from a different ancestor.*”

- It would be good to also provide common names of species at first mentioning.

Response: Thank you for this advice. Accordingly, we added the common names of species at their first mention. For example, “*crucian carp/goldfish (C. auratus)*” (line 32 in the revised manuscript), “*gibel carp/prussian carp (C. gibelio)*” (line 33 in the revised manuscript), “*mud carp (Cirrhinus molitorella)*” (line 189 in the revised manuscript), “*Yunnan Sixupi (Poropuntius huangchuchieni)*” (line 190 in the revised manuscript), “*common carp (Cyprinus carpio)*” (line 194 in the revised manuscript), and “*golden-line barbel (Sinocyclocheilus grahami)*”(line 219 in the revised manuscript).

- 1.58: I don't think it is correct to say that the hybridization increased the chromosome number, because most hybridization events per se do not lead to increased ploidy. It is the allo-polyloidization that leads from $2n=50$ to $2n=100$.

Response: Thank you for pointing out this mistake. We have changed this sentence to: “*Both ancestral parents had 50 chromosomes ($2n=2\times=50$); thus, the allotetraploidy resulted in doubling of chromosome number to 100 ($2n=4\times=100$).*” (lines 60–62 in the revised manuscript).

- 1.73-74: Is this kleptospermy like in the Amazon molly? If so, please also use the term kleptospermy for comparability.

Response: Yes, this is kleptospermy like in the Amazon molly. To clarify this, we revised this sentence as follows: “*...where the eggs are activated by the sperm of sympatric sexual species to initiate embryogenesis, such as by kleptospermy in the Amazon molly...*” (lines 77-79 in the revised manuscript).

- 1.98-99: Please define the x x' terminology more clearly.

Response: Thank you for this suggestion. We revised this sentence as follows: “...58% of heterozygous *k*-mer pairs (with only one nucleotide difference and presented as *x* and *x'*) are bivalent (*xx'*) and 33% of heterozygous *k*-mer pairs are tetravalent (*xxx'x'* and *xxxx'*).” (lines 104–106 in the revised manuscript).

- Why is about 50% of the goldfish genome not incorporated into the assembly (based on estimated genome size)?

Response: We apologize for not clearly explaining this. Approximately 50% of the goldfish genome was not included because *K*-mer analysis estimates the haploid genome size and flow cytometric analysis estimates the total DNA content, and our genome assembly is a haploid genome. To avoid misleading, we rephrased the descriptions as follows: “*The estimated haploid genome size obtained by k-mer analysis ranged from 1.325 to 1.360 Gb (Extended Data Table 5), which was approximately half of the genome content estimated by flow cytometric analysis at 3.00 pg (1 pg is equal to 0.98 Gb).*” (lines 108-111 in the revised manuscript).

- 1.105-123: The goldfish genome sequence has now been published multiple times, so this part should be further shortened.

Response: Thank you for this suggestion. In the revised manuscript, we mainly kept the parts that compared different versions of the goldfish genome. We shortened this part as follows: “*The haplotype genome of C. auratus was assembled (Extended Data Fig. 2), with a size of 1.52 Gb and contig N50 of 3.89 Mb (Extended Data Tables 6–7). We noticed that our assembly size was smaller than previously reported C. auratus assemblies, and this was mainly caused by the redundant assemblies of those heterozygous regions (Extended Data Fig. 3 and Extended Data Table 8). In total, 961 contigs were anchored to 50 chromosomes and had a total length of*

1,476.45 Mb (Fig. 1a and Extended Data Fig. 4) that spanned a genetic map with 3,214 unique markers²⁵ (Extended Data Fig. 5). The 50 chromosomes were divided into two subgenomes (named A and B), each of which included 25 chromosomes (Fig. 1b), based on the annotation of gene and repeat content (Extended Data Tables 9–13 and Supplementary Methods 2.8). The partition of subgenomes was observed to be consistent with previously published domestic goldfish and common carp genomes through synteny analysis (Extended Data Figs. 6 and 7). Compared with the four published domestic goldfish genomes, this new assembly had a higher contig N50, higher chromosome anchoring rate, and much lower level of redundancy, but a high level of completeness (98.30% of complete BUSCO genes) (Extended Data Tables 13 and 14).” (lines 112–128 in the revised manuscript).

- 1.130: Why is the genome assembly size roughly the same length as in goldfish despite the higher ploidy?

Response: We apologize for not clearly elaborating on this issue. Because *C. gibelio* is a real autotriploid that has the same size haploid genome (n=50) as in goldfish, the assembled size of the haplotype genome is similar to that of goldfish. We added the following sentence to lines 136–139 of the revised manuscript: “*The C. gibelio haploid AB subgenome assembly comprised 2,804 contigs, with a length of 1.59 GB and contig N50 of 1.71 Mb (Extended Data Tables 6 and 7), and the size is similar to the C. auratus assembly because both of species have the same amphihaploid content (AB).*”

- 1.185-188: Why is *Sinocyclocheilus* not included in the phylogeny and the divergence time estimates? Genomes are certainly available.

Response: There are four *Sinocyclocheilus* genome assemblies, but none of them have well-distinguished subgenomes A and B. Therefore, we did not use these genomes in this analysis. However, to identify the shared lost genes in the genus *Carassius*, we used the short reads from these genomes.

- 1.193-198: Homeologous gene exchange has also been shown for the common carp (Li et al. 2021, Nat Gen), so this is not total news. The published results in carp should be mentioned and discussed.

Response: Thank you for this comment. In the revised manuscript, we removed related descriptions because we noticed that this conclusion was not well supported by read mapping from *Poropuntius huangchuchieni*. According to the suggestion of Reviewer #1, we aligned the reads of *Poropuntius huangchuchieni* to *C. auratus*. Only five genes (marked with yellow color in the following table) exhibited an even read depth distribution between subgenomes A and B. Although this proportion is higher than the background proportion, this conclusion was no longer significantly supported. Therefore, we removed this part from the revised manuscript.

Gene ID		Mean depth per base	
Subgenome A	Subgenome B	Subgenome A	Subgenome B
evm.model.chr13A.786	evm.model.chr13B.826	22.42	31.85
evm.model.chr14A.114	evm.model.chr14B.114	24.84	32.19
evm.model.chr14A.80	evm.model.chr14B.81	2.54	50.46
evm.model.chr16A.339	evm.model.chr16B.362	9.39	43.17

evm.model.chr18A.32	evm.model.chr18B.28	28.91	23.64
evm.model.chr19A.698	evm.model.chr19B.769	8.83	54.66
evm.model.chr1A.272	evm.model.chr1B.915	12.68	48.43
evm.model.chr21A.73	evm.model.chr21B.76	22.58	33.28
evm.model.chr3A.783	evm.model.chr3B.1091	17.7	43.34
evm.model.chr7A.799	evm.model.chr7B.781	14.38	47.06
evm.model.chr9A.102	evm.model.chr9B.160	8.65	47.01
evm.model.chr9A.156	evm.model.chr9B.78	11.87	39.41
evm.model.chr9A.160	evm.model.chr9B.74	25.34	31.24

- l.199-201: Despite the many recent publications on carp and goldfish genomes, only one is mentioned here (ref. 27). Please pay full reference to these other studies as well.

Response: Thank you for the suggestion. We have added other references about common carp and goldfish genomes to line 204 of the revised manuscript.

- l.203: 23 genes are commonly lost in *C. gibelio* and *C. auratus*. This is not a strikingly high number, easily just by chance. Is this statistically remarkably high? What are the number of shared gene losses between *C. auratus* and *C. carpio* and *C. gibelio* and *C. carpio*? How have these gene losses been confirmed – are there reads bridging the loss?

Response: We apologize for the misunderstanding caused by our wording. It was not our intention to emphasize that 23 is a large number, but to objectively describe the genetic changes

shared by the two species. We rephrased this sentence as follows: “*We mainly analyzed the genetic variations shared by the two Carassius species, and noticed that some of the genes that were lost in both C. gibelio and C. auratus are associated with mitosis/meiosis (Extended Data Table 7).*” (lines 205–208 in the revised manuscript).

Moreover, these genes were confirmed to be present in the four outgroup species, including *Cyprinus carpio*, *S. grahami*, *S. rhinoceros*, and *S. anshuiensis* (we chose these species because that they all experienced allotetraploidy events), which indicated that the genes were conserved. Therefore, the loss of these genes in *C. gibelio* and *C. auratus* may not be the result of neutral events. Indeed, the lost genes were directly identified from read mapping to the genome of *Cyprinus carpio*, and each of them was carefully checked using both read depth analysis and genome annotation. We added the following descriptions in the corresponding Methods section: “*The genes that were detected to be specifically lost in Carassius were further manually checked by: 1) inspecting the read depth across the above individuals, and 2) examining the gene annotation in corresponding genome sequences.*” (lines 293–296 in the Supplementary Methods)

- l. 224-225 “but the genetic changes in these cell division genes in *Carassius* may be associated with subsequent more dramatic ploidy alternation.” This highly speculative statement is not further substantiated and therefore should be removed.

Response: Thank you for this suggestion. We deleted this speculative statement.

- l.361: It is more important to know how many generations (rather than years) the breeding experiment ran.

Response: Thank you for this suggestion. We revised this as follows: “*four-generation breeding experiment*” (lines 365-366 in the revised manuscript).

59References for response letter:

Chen, D. *et al.* The evolutionary origin and domestication history of goldfish (*Carassius auratus*). *Proc. Natl. Acad. Sci. U. S. A.* 202005545 (2020).

Chen, Z. *et al.* De novo assembly of the goldfish (*Carassius auratus*) genome and the evolution of genes after whole-genome duplication. *Sci. Adv.s* **5**, eaav0547 (2019).

Ding, M. *et al.* Genomic anatomy of male-specific microchromosomes in a gynogenetic fish. *PLoS Genet.* **17**, e1009760 (2021).

Gan, R. H. *et al.* Functional divergence of multiple duplicated foxl2 homeologs and alleles in a recurrent polyploid

- fish. *Mol. Biol. Evol.*, **38**, 1995-2013 (2021).
- Kon, T. *et al.* The genetic basis of morphological diversity in domesticated goldfish. *Curr. Biol.* **30**, 1-15 (2020).
- Li, J. T. *et al.* Parallel subgenome structure and divergent expression evolution of allo-tetraploid common carp and goldfish. *Nat. Genet.* **53**, 1493-1503 (2021).
- Luo, J. *et al.* From asymmetrical to balanced genomic diversification during rediploidization: subgenomic evolution in allotetraploid fish. *Sci. Adv.* **6**, eaaz7677 (2020).
- Mou, C. Y. *et al.* Divergent antiviral mechanisms of two *viperin* homeologs in a recurrent polyploid fish. *Front. Immunol.* **12**, 702971 (2021).
- Ranallo-Benavidez, T. R., Jaron, K. S. & Schatz, M. C. GenomeScope 2.0 and Smudgeplot for reference-free profiling of polyploid genomes. *Nat. Commun.* **11**, 1432 (2020).
- Wen, M. *et al.* Sex chromosome and sex locus characterization in goldfish, *Carassius auratus* (Linnaeus, 1758). *BMC Genomics* **21**, 552 (2020).
- Xu, P. *et al.* The allotetraploid origin and asymmetrical genome evolution of common carp *Cyprinus carpio*. *Nat. Commun.* **10**, 4625 (2019).

Decision Letter, first revision:

21st April 2022

Dear Dr. Gui,

Thank you for submitting your revised manuscript "Comparative genome anatomy reveals evolutionary insights into a unique amphitriploid fish" (NATECOLEVOL-211115122A). It has now been seen again by the original reviewers and their comments are below. The reviewers find that the paper has improved in revision, and therefore we'll be happy in principle to publish it in Nature Ecology & Evolution, pending minor revisions to satisfy the reviewers' final requests and to comply with our editorial and formatting guidelines.

[REDACTED]

61Reviewer #1 (Remarks to the Author):

The manuscript was greatly improved and their conclusions are supported with more evidence. The sequencing technologies develop rapidly. It is completely possible that a finished genome of *C. auratus* will be generated in the near future. Many assemblies of goldfish were released but their sizes and the annotated gene numbers much diverged. Therefore, the authors should try their best to ensure the solidness of their conclusions. Before publication, some concerns should be replied.

1. The estimated genome size for *C. auratus* ranged from 1.325 to 1.360 Gb. But the assembled haplotype genome was 1.52 Gb, larger than the estimated size, which suggested the redundancy in the assembly. In most cases, the assembled size is smaller than the estimated one. They should explain it.
2. The authors were not able to calculate the mapping rate of the other three published genome assemblies (GCF_003368295.1 in Chen et al., *Sci Adv*, 2019; GCA_014332655.1 in Chen et al., *Proc Natl Acad Sci U S A*, 2020; GWHAIA000000000 in Luo et al., *Sci Adv*, 2020) because the corresponding short reads were not available. I suggest that they align their own short reads to the other versions and estimate the read coverages.
3. The author manually checked the lost genes identified by their study in four other published assemblies and confirmed that they did not exist in any of the other assemblies, which confirmed that they are lost genes. How did they manually check? Please described the method. A supplementary table summarizing the lost genes and the alignment information to the other assemblies is useful.
4. The author pointed out that *rag2* (in Xu et al., *Nat Commun*, 2019, Figure 2a) was not appropriate for inferring phylogenetic relationships between these subgenomes, and a phylogenetic tree based on genome-wide data should be more reliable. This information should be described in the manuscript to prove that the authors' analysis is solid. A correct phylogenetic tree benefits the community.
5. The authors used two different common carp genomes for different analyses. To avoid misunderstandings, I suggest the author should use only one common carp genome assembly throughout the entire manuscript. The recent version is the best choice. They can download the updated gene annotation of this version in the NCBI assembly database. I checked the ENSEMBL database and it has not provided the gene annotation for GCA_000951615.2.

Reviewer #2 (Remarks to the Author):

The authors have addressed my previous comments properly, and have also performed additional experiments (Fig. 6). The English language has also been significantly improved. Below I have a few minor comments and suggestions.

L115-116, I was wondering whether this is because other assemblies did not have the *halotigs* purged?

L186, this part describes the genomic changes in the common ancestor of *Carassius* - I am wondering what are the implications for the success of *C. gibelio*? If these genomic changes (e.g. loss of *cdk2*) are common to all *Carassius*, they should have been selected at the *Carassius* ancestor. Given that this manuscript mainly deals with *C. gibelio*, this part seems a bit distracting. I would suggest streamlining this part, and perhaps also moving Fig. 2 into supplementary figures.

L247-251, this is a bit out of context, perhaps move it (together with Fig. 3c-d) somewhere else.

L304, in Fig. 3d, I am wondering what was the reference used to estimate copy numbers in re-sequencing data? For instance, *h2af1a1* has 13 copies in the assemblies, presumably all copies are assembled - would it be a good reference to estimate copy numbers for other individuals? I am also wondering how divergent are between different copies, and whether and how it could impact RNA-seq analyses? I am also curious how was the gene duplicated - through tandem duplications?

L375, perhaps use a supplementary figure to show one example?

Reviewer #3 (Remarks to the Author):

The authors have addressed my previous concerns diligently and in much detail and I have no further requests for revision.

In case there are length issues for the final publication, I suggest to further shorten the parts about assembling the *C. auratus* genome and its characterization, which are confirming results that have been repeatedly published before.

Our ref: NATECOLEVOL-211115122A

25th April 2022

Dear Dr. Gui,

Thank you for your patience as we've prepared the guidelines for final submission of your Nature Ecology & Evolution manuscript, "Comparative genome anatomy reveals evolutionary insights into a unique amphitriploid fish" (NATECOLEVOL-211115122A). Please carefully follow the step-by-step

63instructions provided in the attached file, and add a response in each row of the table to indicate the changes that you have made. Please also check and comment on any additional marked-up edits we have proposed within the text. Ensuring that each point is addressed will help to ensure that your revised manuscript can be swiftly handed over to our production team.

****We would like to start working on your revised paper, with all of the requested files and forms, as soon as possible (preferably within two weeks). Please get in contact with us immediately if you anticipate it taking more than two weeks to submit these revised files.****

In recognition of the time and expertise our reviewers provide to Nature Ecology & Evolution's editorial process, we would like to formally acknowledge their contribution to the external peer review of your manuscript entitled "Comparative genome anatomy reveals evolutionary insights into a unique amphitriploid fish". For those reviewers who give their assent, we will be publishing their names alongside the published article.

Nature Ecology & Evolution offers a Transparent Peer Review option for new original research manuscripts submitted after December 1st, 2019. As part of this initiative, we encourage our authors to support increased transparency into the peer review process by agreeing to have the reviewer comments, author rebuttal letters, and editorial decision letters published as a Supplementary item. When you submit your final files please clearly state in your cover letter whether or not you would like to participate in this initiative. Please note that failure to state your preference will result in delays in accepting your manuscript for publication.

Cover suggestions

As you prepare your final files we encourage you to consider whether you have any images or illustrations that may be appropriate for use on the cover of Nature Ecology & Evolution.

64Please submit your suggestions, clearly labeled, along with your final files. We'll be in touch if more information is needed.

Nature Ecology & Evolution has now transitioned to a unified Rights Collection system which will allow our Author Services team to quickly and easily collect the rights and permissions required to publish your work. Approximately 10 days after your paper is formally accepted, you will receive an email in providing you with a link to complete the grant of rights. If your paper is eligible for Open Access, our Author Services team will also be in touch regarding any additional information that may be required to arrange payment for your article.

Please note that *Nature Ecology & Evolution* is a Transformative Journal (TJ). Authors may publish their research with us through the traditional subscription access route or make their paper immediately open access through payment of an article-processing charge (APC). Authors will not be required to make a final decision about access to their article until it has been accepted. [Find out more about Transformative Journals](https://www.springernature.com/gp/open-research/transformative-journals)

Authors may need to take specific actions to achieve [compliance with funder and institutional open access mandates](https://www.springernature.com/gp/open-research/funding/policy-compliance-faqs). If your research is supported by a funder that requires immediate open access (e.g. according to [Plan S principles](https://www.springernature.com/gp/open-research/plan-s-compliance)) then you should select the gold OA route, and we will direct you to the compliant route where possible. For authors selecting the subscription publication route, the journal's standard licensing terms will need to be accepted, including <https://www.nature.com/nature-portfolio/editorial-policies/self-archiving-and-license-to-publish>. Those licensing terms will supersede any other terms that the author or any third party may assert apply to any version of the manuscript.

Please use the following link for uploading these materials:
[REDACTED]

[REDACTED]

Reviewer #1:

Remarks to the Author:

The manuscript was greatly improved and their conclusions are supported with more evidence. The sequencing technologies develop rapidly. It is completely possible that a finished genome of *C. auratus* will be generated in the near future. Many assemblies of goldfish were released but their sizes and the annotated gene numbers much diverged. Therefore, the authors should try their best to ensure the solidness of their conclusions. Before publication, some concerns should be replied.

1. The estimated genome size for *C. auratus* ranged from 1.325 to 1.360 Gb. But the assembled haplotype genome was 1.52 Gb, larger than the estimated size, which suggested the redundancy in the assembly. In most cases, the assembled size is smaller than the estimated one. They should explain it.
2. The authors were not able to calculate the mapping rate of the other three published genome assemblies (GCF_003368295.1 in Chen et al., *Sci Adv*, 2019; GCA_014332655.1 in Chen et al., *Proc Natl Acad Sci U S A*, 2020; GWHAIA000000000 in Luo et al., *Sci Adv*, 2020) because the corresponding short reads were not available. I suggest that they align their own short reads to the other versions and estimate the read coverages.
3. The author manually checked the lost genes identified by their study in four other published assemblies and confirmed that they did not exist in any of the other assemblies, which confirmed that they are lost genes. How did they manually check? Please describe the method. A supplementary table summarizing the lost genes and the alignment information to the other assemblies is useful.
4. The author pointed out that *rag2* (in Xu et al., *Nat Commun*, 2019, Figure 2a) was not appropriate for inferring phylogenetic relationships between these subgenomes, and a phylogenetic tree based on genome-wide data should be more reliable. This information should be described in the manuscript to prove that the authors' analysis is solid. A correct phylogenetic tree benefits the community.
5. The authors used two different common carp genomes for different analyses. To avoid misunderstandings, I suggest the author should use only one common carp genome assembly throughout the entire manuscript. The recent version is the best choice. They can download the updated gene annotation of this version in the NCBI assembly database. I checked the ENSEMBL database and it has not provided the gene annotation for GCA_000951615.2.

Reviewer #2:

Remarks to the Author:

The authors have addressed my previous comments properly, and have also performed additional experiments (Fig. 6). The English language has also been significantly improved. Below I have a few

66minor comments and suggestions.

L115-116, I was wondering whether this is because other assemblies did not have the halotigs purged?

L186, this part describes the genomic changes in the common ancestor of *Carassius* - I am wondering what are the implications for the success of *C. gibelio*? If these genomic changes (e.g. loss of *cdk2*) are common to all *Carassius*, they should have been selected at the *Carassius* ancestor. Given that this manuscript mainly deals with *C. gibelio*, this part seems a bit distractive. I would suggest streamlining this part, and perhaps also moving Fig. 2 into supplementary figures.

L247-251, this is a bit out of context, perhaps move it (together with Fig. 3c-d) somewhere else.

L304, in Fig. 3d, I am wondering what was the reference used to estimate copy numbers in re-sequencing data? For instance, *h2af1a1* has 13 copies in the assemblies, presumably all copies are assembled - would it be a good reference to estimate copy numbers for other individuals? I am also wondering how divergent are between different copies, and whether and how it could impact RNA-seq analyses? I am also curious how was the gene duplicated - through tandem duplications?

L375, perhaps use a supplementary figure to show one example?

Reviewer #3:

Remarks to the Author:

The authors have addressed my previous concerns diligently and in much detail and I have no further requests for revision.

In case there are length issues for the final publication, I suggest to further shorten the parts about assembling the *C. auratus* genome and its characterization, which are confirming results that have been repeatedly published before.

Author Rebuttal, first revision:

Reviewer #1 (Remarks to the Author):

The manuscript was greatly improved and their conclusions are supported with more evidence. The sequencing technologies develop rapidly. It is completely possible that a finished genome of *C. auratus* will be generated in the near future. Many assemblies of goldfish were released but their sizes and the annotated gene numbers much diverged. Therefore, the authors should try their best to ensure the solidness of their conclusions. Before publication, some concerns should be replied.

Reply: Thank you very much for your positive comments.

1. The estimated genome size for *C. auratus* ranged from 1.325 to 1.360 Gb. But the assembled haplotype genome was 1.52 Gb, larger than the estimated size, which suggested the redundancy in the assembly. In most cases, the assembled size is smaller than the estimated one. They should explain it.

Reply: Thank you for the suggestion. We have tried different softwares to estimate the genome sizes, including GCE, GenomeScope and SOAPec. However, the estimated sizes obtained by different softwares and parameters varied greatly (such as 0.66, 1.3, 1.6 and 4.8Gb), perhaps due to the fact that these softwares are deficient in dealing with complex polyploids, such as misjudging the peak of the *k-mer* distribution. Therefore, we re-counted *k-mer* again with the classic jellyfish method (Marçais and Kingsford, Bioinformatics, 2011) and combined various information, such as polyploidy, flow cytometry results, and other published genome assembly results, to determine which peak corresponds to the haplotype size of the genome in the *k-mer* distribution and make an estimate of the genome size. The estimated genome sizes by this approach are about 1.44~1.54 Gb for *C. auratus* and 1.49~1.56 Gb for *C. gibelio*, both of which match well with the final assembled sizes of the two genomes.

We have updated the results in the new table (Supplementary Table S5) and discussed these estimates in Supplementary Note 1 Line 20-29: *“It should be noted that we have also tried different softwares to estimate the genome size, including GCE, GenomeScope and SOAPec, but the estimated sizes from different methods and parameters varied greatly (such as 0.66, 1.3, 1.6 and 4.8Gb). Perhaps this could be attributed to that these softwares are still deficient in dealing with complex polyploids, such as misjudging the peak of the k-mer distribution. Therefore, we combined various information, such as polyploidy, flow cytometry results, and other published genome assembly results, to determine which peak corresponds to the haplotype size of the genome in the k-mer distribution and make an estimate of the genome size.”*

2. The authors were not able to calculate the mapping rate of the other three published genome assemblies (GCF_003368295.1 in Chen et al., Sci Adv, 2019; GCA_014332655.1 in Chen et al., Proc Natl Acad Sci U S A, 2020; GWHA000000000 in Luo et al., Sci Adv, 2020) because the corresponding short reads were not available. I suggest that they align their own short reads to the other versions and estimate the read coverages.

Reply: We appreciate your suggestion and have aligned our short reads to the three published genome assemblies. The overall mapping rates are ranged from 98.90 ~ 99.29%, slightly lower than that of our own genome (99.85%), which is either caused by the divergence of the sequenced individuals or higher quality of our assembly. We have updated the above information to Supplementary Table 23 and Supplementary Fig. 20 in the revised manuscript.

3. The author manually checked the lost genes identified by their study in four other published assemblies and confirmed that they did not exist in any of the other assemblies, which confirmed that they are lost genes. How did they manually check? Please described the method. A supplementary table summarizing the lost genes and the alignment information to the other assemblies is useful.

Reply: Thanks for the suggestion, and we have added the details in the Supplementary information (line 88-101):

“Moreover, to check whether the lost genes identified by our study are present or absent in other four published assemblies, we applied a three-step manual check for each gene. Firstly, the amino acid (aa) sequences of the gene from C. carpio genome (GCA_018340385.1) were aligned to six Carassius genomes’ sequences as well as to Danio rerio and Sinocyclocheilus graham using tblastn (v2.10.1) to obtain the top best hits (identity > 50%). Secondly, the gene structure and aa sequences of the targeted regions were extracted using annotated gff or genewise (v2.2.0). Thirdly, all the aa sequences were aligned together using mafft (v7.471), and a maximum likelihood tree was constructed using RAxML (v8.2.12). Then, the alignment sequences were checked with MEGA (v7.0.26) and the ML tree was checked with figtree (v1.4.4) to determine whether this gene was present in each genome assembly (Supplementary Fig. S21). The gene alignments and trees are available at figshare database (<https://doi.org/10.6084/m9.figshare.19674843.v1>).”

4. The author pointed out that rag2 (in Xu et al., Nat Commun, 2019, Figure 2a) was not appropriate for inferring phylogenetic relationships between these subgenomes, and a phylogenetic tree based on genome-wide data should be more reliable. This information should be described in the manuscript to prove that the authors' analysis is solid. A correct phylogenetic tree benefits the community.

Reply: Thanks for your suggestion. We have added the relevant results and discussion in line 198-205:

“In addition, we noticed that the phylogenetic position of Cirrhinus molitorella from our species tree and a previous study³⁰ conflicted with another previous study³³, in which a single gene (rag2) tree was constructed and the results showed that C. molitorella was an outgroup of both subgenomes A and B. To determine why this inconsistency occurred, we further examined the proportion of topology for each orthologous gene. The results highlighted a high level of phylogeny heterogeneity (Supplementary Table 14), and the topology with the highest proportion was consistent with our species tree.”

5. The authors used two different common carp genomes for different analyses. To avoid misunderstandings, I suggest the author should use only one common carp genome assembly throughout the entire manuscript. The recent version is the best choice. They can download the updated gene

annotation of this version in the NCBI assembly database. I checked the ENSEMBL database and it has not provided the gene annotation for GCA_000951615.2.

Reply: We agree with this suggestion. In the previous manuscript, we used the genome assembly GCA_000951615.2 (the older assembly from ENSEMBL) for analyzing the gene loss in the genus *Carassius*. In the revised manuscript, we have redone this analysis using the genome assembly GCA_018340385.1 (the more recent assembly at NCBI) and updated relevant table (Supplementary Table 15) and figure (Supplementary Fig. 21). The new analysis did not change the conclusion.

Reviewer #2 (Remarks to the Author):

The authors have addressed my previous comments properly, and have also performed additional experiments (Fig. 6). The English language has also been significantly improved. Below I have a few minor comments and suggestions.

Reply: Thank you very much for your positive comments.

L115-116, I was wondering whether this is because other assemblies did not have the halotigs purged?

Reply: As you said, this is indeed the case. We have also purged their assemblies and obtained genome sizes similar to our assembly (Supplementary Table 23).

L186, this part describes the genomic changes in the common ancestor of *Carassius* - I am wondering what are the implications for the success of *C. gibelio*? If these genomic changes (e.g. loss of *cdk2*) are common to all *Carassius*, they should have been selected at the *Carassius* ancestor. Given that this manuscript mainly deals with *C. gibelio*, this part seems a bit distracting. I would suggest streamlining this part, and perhaps also moving Fig. 2 into supplementary figures.

Reply: Thanks for your suggestion, we agree that this part is a bit distracting. In the revised version, we have moved this section into Supplementary Note 4 and Supplementary Fig. 10.

L247-251, this is a bit out of context, perhaps move it (together with Fig. 3c-d) somewhere else.

Reply: Thank you for your suggestion, and these two figures are now moved to the supplementary files and the related sentence was rephrased into “Moreover, 4,400 non-coding elements were found to be shared by all *C. gibelio* individuals (Supplementary Fig. 11 and Supplementary Note 5) but were absent in *C. auratus*, *Cyprinus carpio*, and *S. graham*, indicating that they are newly evolved elements in *C. gibelio*.”

L304, in Fig. 3d, I am wondering what was the reference used to estimate copy numbers in re-sequencing

70data? For instance, h2af1a1 has 13 copies in the assemblies, presumably all copies are assembled - would it be a good reference to estimate copy numbers for other individuals? I am also wondering how divergent are between different copies, and whether and how it could impact RNA-seq analyses? I am also curious how was the gene duplicated - through tandem duplications?

Reply: Thanks for pointing out these issues.

For the first question, in fact, we used the genome assembly of *C. auratus* for reads mapping to initially screening genes that are specifically expanded in *C. gibelio*. These genes were then checked in the genome assembly of *C. gibelio*. Therefore, the specific expansion of these genes did not affect the initial screen.

For the second question, as the expansion events were occurred before the divergence of all the investigated *C. gibelio* individuals, which have been diverged for over 0.82 million years, these expanded copies had accumulated some specific mutations. Many mutations can be observed in the aligned nucleotide acids (average 1% to 5%, and one gene family even with 12% nucleotide acid substitutions between any pairs in the expanded genes). These mutations ensures that the RNA-seq reads could be specifically mapped. We have uploaded the sequence alignments to figshare database (<https://doi.org/10.6084/m9.figshare.19674843.v1>).

For the last question, we have checked the expanded genes and found some of them are results of tandem duplication (Extended Data Fig. 3) and some of them are dispersed in the genome. We have added a column in Supplementary Table 19 to show how each gene duplicated by indicating tandem duplication or dispersed distribution.

L375, perhaps use a supplementary figure to show one example?

Reply: Following your suggestion, we have added a supplementary figure (Supplementary Fig. 17) to show two examples.

Reviewer #3 (Remarks to the Author):

The authors have addressed my previous concerns diligently and in much detail and I have no further requests for revision.

Reply: Thank you very much for your positive comments.

In case there are length issues for the final publication, I suggest to further shorten the parts about assembling the *C. auratus* genome and its characterization, which are confirming results that have been repeatedly published before.

Reply: Thanks for your suggestion, we have streamlined the text here and put the details of genome assembly into Supplementary Note 2.

The corresponding text now reads as follows:

“The C. auratus genome was also assembled with a size of 1.52 Gb and contig N50 of 3.89 Mb, and anchored to 50 chromosomes (Fig. 1a, Supplementary Fig.3, Supplementary Tables 6 and 7, Supplementary Note 2).”

Final Decision Letter:

25th May 2022

Dear Professor Gui,

We are pleased to inform you that your Article entitled "Comparative genome anatomy reveals evolutionary insights into a unique amphitriploid fish", has now been accepted for publication in Nature Ecology & Evolution.

Over the next few weeks, your paper will be copyedited to ensure that it conforms to Nature Ecology and Evolution style. Once your paper is typeset, you will receive an email with a link to choose the appropriate publishing options for your paper and our Author Services team will be in touch regarding any additional information that may be required

You will not receive your proofs until the publishing agreement has been received through our system

Due to the importance of these deadlines, we ask you please us know now whether you will be difficult to contact over the next month. If this is the case, we ask you provide us with the contact information (email, phone and fax) of someone who will be able to check the proofs on your behalf, and who will be available to address any last-minute problems . Once your paper has been scheduled for online publication, the Nature press office will be in touch to confirm the details.

Acceptance of your manuscript is conditional on all authors' agreement with our publication policies (see www.nature.com/authors/policies/index.html). In particular your manuscript must not be published elsewhere and there must be no announcement of the work to any media outlet until the publication date (the day on which it is uploaded onto our web site).

Please note that *Nature Ecology & Evolution* is a Transformative Journal (TJ). Authors may

72publish their research with us through the traditional subscription access route or make their paper immediately open access through payment of an article-processing charge (APC). Authors will not be required to make a final decision about access to their article until it has been accepted. [Find out more about Transformative Journals](https://www.springernature.com/gp/open-research/transformative-journals)

Authors may need to take specific actions to achieve [compliance with funder and institutional open access mandates](https://www.springernature.com/gp/open-research/funding/policy-compliance-faqs). If your research is supported by a funder that requires immediate open access (e.g. according to [Plan S principles](https://www.springernature.com/gp/open-research/plan-s-compliance)) then you should select the gold OA route, and we will direct you to the compliant route where possible. For authors selecting the subscription publication route, the journal's standard licensing terms will need to be accepted, including [self-archiving-and-license-to-publish](https://www.nature.com/nature-portfolio/editorial-policies/self-archiving-and-license-to-publish). Those licensing terms will supersede any other terms that the author or any third party may assert apply to any version of the manuscript.

We welcome the submission of potential cover material (including a short caption of around 40 words) related to your manuscript; suggestions should be sent to Nature Ecology & Evolution as electronic files (the image should be 300 dpi at 210 x 297 mm in either TIFF or JPEG format). Please note that such pictures should be selected more for their aesthetic appeal than for their scientific content, and that colour images work better than black and white or grayscale images. Please do not try to design a cover with the Nature Ecology & Evolution logo etc., and please do not submit composites of images related to your work. I am sure you will understand that we cannot make any promise as to whether any of your suggestions might be selected for the cover of the journal.

You can generate the link yourself when you receive your article DOI by entering it here: http://authors.springernature.com/share.

[REDACTED]

P.S. Click on the following link if you would like to recommend Nature Ecology & Evolution to your librarian <http://www.nature.com/subscriptions/recommend.html#forms>

** Visit the Springer Nature Editorial and Publishing website at www.springernature.com/editorial-and-publishing-jobs for more information about our career opportunities. If you have any questions please click here. **